# A new accurate retrieval algorithm of bromine monoxide columns inside minor volcanic plumes from Sentinel-5 Precursor/TROPOMI observations

Simon Warnach[1,2], Holger Sihler[4], Christian Borger[1], Nicole Bobrowski[2,3], Steffen Beirle[1], Ulrich Platt[2], and Thomas Wagner[1]

[1]Max-Planck-Institut für Chemie (MPIC), Satellite remote sensing group, Mainz, Germany
[2]Institut für Umweltphysik (IUP), Heidelberg University, Heidelberg, Germany
[3]Istituto Nazionale di Geofisica e Vulcanologia - Osservatorio Etneo, Catania, Italy
[4]formerly at: Max-Planck-Institut für Chemie (MPIC), Satellite remote sensing group, Mainz, Germany

**Correspondence:** Simon Warnach (s.warnach@mpic.de)

**Abstract.** Bromine monoxide (BrO) is a key radical in the atmosphere, influencing the chemical state of the atmosphere, most notably the abundance of ozone ($O_3$). $O_3$ depletion caused by the release of bromine has been observed and modeled in polar regions, salt pans, and in particular inside volcanic plumes. Furthermore, the molar ratio of BrO and $SO_2$ – which can be detected simultaneously via spectroscopic measurements using the Differential Optical Absorption Spectroscopy (DOAS) method - is a proxy for the magmatic composition of a volcano and potentially an eruption forecast parameter.

The detection of BrO in volcanic plumes from satellite spectroscopic observations is limited by the precision and sensitivity of the retrieval, which so far only allowed for the detection of BrO during major eruptions. The unprecedented spatial resolution of up to $3.5 \times 5.5\,\mathrm{km}^2$ and high signal-to-noise ratio of the TROPOspheric Monitoring Instrument (TROPOMI) onboard Sentinel-5 Precursor (S-5P) enables to observe and monitor volcanic bromine release globally even for minor eruptions or even quiescent degassing.

In this study, we investigate, how far the BrO retrieval can be improved using TROPOMI data and how well BrO can be detected, even in small eruptions and during quiescent volcanic degassing. There are two steps, for which improvements in accuracy are investigated and applied: the improvement and quantitative determination of (1) the detection limit of the DOAS BrO column retrieval and (2) the correction of non-volcanic background BrO signal. First, the DOAS retrieval settings are varied and their influence on accuracy and precision is investigated with respect to the detection limit and potential systematic influences. Based on these results, we propose a dedicated DOAS evaluation scheme optimized for the detection of BrO in volcanic plumes. For the DOAS retrieval, we propose the use of a large fit window from $323 - 360\,\mathrm{nm}$, yielding a factor of 1.8 lower statistical uncertainty compared to previous BrO DOAS algorithms, while not enhancing systematic influences. Second, the effect of the background BrO is reduced by a latitude dependent empirical correction scheme correlated to cloud information as well as information on the $O_3$ column. Via these improvements, the combined statistical and systematic uncertainties of the resulting BrO vertical column density is in the order of $7 \times 10^{12}\,\mathrm{molecules\,cm}^{-2}$.

We present a new and accurate retrieval algorithm of BrO columns from TROPOMI observations, which allows for the detection of even slightly enhanced BrO amounts inside minor eruptive plumes of bromine-rich volcanoes. While designed specifically for TROPOMI observations, the retrieval algorithm is in general also applicable to other hyperspectral satellite observations. However, some parts might require adaptation.

## 1 Introduction

Bromine monoxide (BrO) has been found to be a catalyst for ozone ($O_3$) destruction (e. g. in the polar regions, see Barrie et al., 1988; Simpson et al., 2007). Also volcanic bromine release is thought to also affect atmospheric $O_3$ chemistry (Bobrowski et al., 2003; von Glasow, 2010; Surl et al., 2021). However, $O_3$ measurements inside volcanic plumes are limited and the strength of the $O_3$ destruction of volcanic bromine is still not fully clear (an estimation can be found in the review by Surl et al., 2015). Recently, the development of chemiluminescence instruments for the detection of $O_3$ inside volcanic plumes shows promising results (Rüth, 2021; Bräutigam, 2022) and could help to better constrain the volcanic $O_3$ destruction, as they allow interference-free measurements of $O_3$ in volcanic plumes, in contrast to the standard Ultra-Violet (UV)-$O_3$ monitors often used today. Strong eruptions ejecting bromine into the tropical stratosphere could potentially impact global stratospheric $O_3$ abundance over a ten-year period (Brenna et al., 2019).

BrO in volcanic plumes was first detected at Soufriere Hills (Bobrowski et al., 2003) using ground-based Differential Optical Absorption Spectroscopic (DOAS, Perner and Platt, 1979) measurements. BrO and sulphur dioxide ($SO_2$), which can be detected by DOAS simultaneously, have been measured over the last 20 years at many volcanoes using ground-based instruments. Ground-based (Bobrowski and Platt, 2007; Gutmann et al., 2018), as well as satellite observations (Hörmann et al., 2013) report inter-volcanic variations of the BrO/$SO_2$ ratio of three orders of magnitude, strongly suggesting a link to differences in the geological settings of the individual volcanoes (Platt and Bobrowski, 2015). Additionally, variations of the BrO/$SO_2$ over time have been attributed as a proxy for changes in the volcanic systems, for instance at Mt. Etna, Italy (Bobrowski and Giuffrida, 2012), Nevado del Ruiz, Colombia (Lübcke et al., 2014), Cotopaxi, Ecuador (Dinger et al., 2018) and Tungurahua, Ecuador (Warnach et al., 2019). These variations are suggested to be linked to differences in the partitioning of bromine and sulphur from the magmatic melt, i. e. that bromine and sulphur are released at different depths below the surface. However, the interpretation remains difficult, as the partitioning of bromine from the melt is not yet well-constrained.

After the unsuccessful attempt to detect BrO in volcanic plumes using satellite data from the Global Ozone Monitoring Instrument (GOME) and SCanning Imaging Absorption Spectrometer for Atmospheric Chartography (SCIAMACHY) (Afe et al., 2004), volcanic BrO was first detected from GOME-2 in the plume of Kasatochi in 2008 by Theys et al. (2009). A global detection algorithm of BrO and the BrO/$SO_2$ ratios inside volcanic plumes from GOME-2 was developed by Hörmann et al. (2013).

DOAS retrievals of BrO from satellites have varied over time, both w. r. t. the wavelength fit range as well as other DOAS fit settings. The first detection of tropospheric BrO from the GOME instrument by Wagner and Platt (1998), employs a wavelength range of $345 - 359.5\,nm$. Other studies use almost identical fit ranges for spectra from GOME (Richter et al., 1998; Hegels

et al., 1998; Chance, 1998; Richter et al., 2002) and for spectra from the Ozone Monitoring Instrument (OMI Chance, 2002). For SCIAMACHY and GOME-2 measurements, the short wavelength limit of the fit range was lowered to $336\,\mathrm{nm}$, and the upper limit varied between $347\,\mathrm{nm}$ (Afe et al., 2004; Begoin et al., 2010), $351-352\,\mathrm{nm}$ (Valks et al., 2009; Theys et al., 2009), and $360\,\mathrm{nm}$ (Heue et al., 2011; Sihler et al., 2012; Hörmann et al., 2013). The short wavelength limit of $336\,\mathrm{nm}$ was also found best in a long-term study combining GOME, SCIAMACHY, and GOME-2 measurements (Bougoudis et al., 2020). For recent global studies of $BrO$ from GOME-2 (Theys et al., 2011) and TROPOMI (Seo et al., 2019) the lower boundary was further lowered and a fit range of $332-359\,\mathrm{nm}$ and $334.6-358\,\mathrm{nm}$ was chosen respectively.

The DOAS $BrO$ fit used in this study is based on the $BrO$ fit developed by Sihler et al. (2012) for the retrieval of $BrO$ in arctic regions, which was also used for the detection of $BrO$ in volcanic plumes by Hörmann et al. (2013). There, a fit range from $336-360\,\mathrm{nm}$ was chosen. The omission of shorter wavelengths was mainly attributed to the strong $O_3$ absorption in the UV at high latitudes, caused by the large $O_3$ columns and high solar zenith angles (SZAs).

$SO_2$ is one major constituent of volcanic gas plumes, accounting for $1-25\,\%$ of the volcanic gas emissions (Textor et al., 2004). In contrast to $BrO$, there is no significant stratospheric column of $SO_2$. As $SO_2$ has differential absorption structures in the UV, it is widely used as a indicator for the detection of volcanic plumes (e. g. Hörmann et al., 2013; Fioletov et al., 2016; Carn et al., 2017).

DOAS $SO_2$ retrievals from satellites were performed starting with the launch of GOME in 1995 (Eisinger and Burrows, 1998; Thomas et al., 2005; Khokhar et al., 2005); later DOAS retrievals were implemented also for SCIAMACHY (Afe et al., 2004), GOME-2 (Rix et al., 2012; Hörmann et al., 2013), OMI (Theys et al., 2015) and most recently for TROPOMI (Theys et al., 2017). As $SO_2$ columns in volcanic plumes vary over multiple orders of magnitude, recent algorithms introduced a scheme using multiple fits in different wavelength ranges with varying $SO_2$ absorption strength (Hörmann et al., 2013; Theys et al., 2015; Warnach, 2022).

In this study, we report on the further improvement of the $BrO$ DOAS retrieval with a specific focus on the detection of $BrO$ in volcanic plumes and propose a new accurate $BrO$ retrieval for the detection within minor volcanic plumes. Retrieval improvements are investigated by varying the DOAS fit settings, by both considering lower wavelengths and the inclusion/exclusion of the formaldehyde (HCHO) cross section in the DOAS fit. The aim of this task was to achieve both a low statistical uncertainty and low systematic errors, especially low interferences caused by clouds, $O_3$ and HCHO.

There are several reasons why - in contrast to previous retrievals - lower wavelengths can be included in current $BrO$ DOAS retrievals for the detection of volcanic plumes. First, the introduction of the so-called Pukite terms (Puķīte et al., 2010) can largely compensate the spectral effects of the strong $O_3$ absorption. Second, volcanic plumes mainly occur at tropical latitudes, where $O_3$ columns and the SZAs are considerably lower compared to mid- and high latitudes. Third, the use of an earthshine reference spectrum recorded around the equator reduces the optical density of $O_3$ to zero at the equator, further reducing a potential $O_3$ interference at tropical latitudes.

The question whether it is advantageous to include HCHO in the $BrO$ DOAS fit is long debated, as the interference between $BrO$ and HCHO is a well known difficulty both in $BrO$ and HCHO DOAS retrievals (Theys et al., 2011; Vogel et al., 2013; De Smedt et al., 2018). The structure of the absorption cross-section of $BrO$ and HCHO are very similar (cf. Fig. 1).

Especially the fact, that the largest absorption peak of BrO overlaps with a major absorption peak of HCHO at 339 nm, can cause a spectral interference between both trace gases. For arctic applications, such as Wagner and Platt (1998), Richter et al. (1998), but also for the more recent study by Sihler et al. (2012), HCHO was not included as an absorber in the DOAS retrieval, because HCHO is almost not abundant in arctic regions. However, in equatorial regions, where HCHO can reach high column densities due to biogenic emissions or biomass burning, spectral interference with HCHO can become important. HCHO is therefore included within the DOAS analysis of the global BrO detection algorithms of GOME-2 by Theys et al. (2011) and the TROPOMI algorithm of polar BrO by Seo et al. (2019). Interferences between BrO and HCHO are observed for equatorial regions by Theys et al. (2011). Using 332 nm compared to 336 nm (used in Sihler et al., 2012) as the lower fit boundary reduces these interferences (Theys et al., 2011). Hence, both Theys et al. (2011) and Seo et al. (2019) employ fit ranges with a lower wavelength limit of 332 nm and 334.6 nm respectively.

The stratospheric background of BrO together with the potential BrO background of the free troposphere is up to one order of magnitude larger than typical volcanic BrO columns of small eruptions. Therefore, an imperfect subtraction of the background column can yield uncertainties in the order of small volcanic signals. Hence, an accurate background correction is essential. Fortunately, horizontal gradients of the stratospheric BrO column occur on a much larger scale than the extension of volcanic plumes. Moreover, while stratospheric gradients mainly originate from changes in the tropopause height (Sihler et al., 2012), volcanic plumes are dispersions of point sources. Thus, the patterns are spatially independent from each other and a separation can be achieved by the application of a spatial high-pass filter. Moreover, volcanic plumes can be independently detected due to their enhanced $SO_2$ column density. Background correction on this basis was for example implemented by Hörmann et al. (2013) by first masking the volcanic plume, which is well detectable via the $SO_2$ signal, and subtracting a 2D polynomial fitted to the area around the volcanic plume.

While our BrO DOAS retrieval is in principle applicable to the complete globe, similar to the retrievals presented in Theys et al. (2011); Sihler et al. (2012); Seo et al. (2019), it is specifically optimized for the most accurate detection of BrO originating from volcanic plumes, which are local/regional events whose spatial extent is typically of the order of $10 - 100$ km, only rarely exceeding 1000 km.

Even though our BrO retrieval algorithm is designed for TROPOMI observations, the improvements tested are – aside from satellite specific adaptations – in principle applicable to all hyperspectral satellite observations.

This paper is structured as follows: in Sect. 2, we describe the TROPOMI instrument and the basics of our BrO DOAS retrieval from satellite. The investigation of different DOAS fit settings is described in Sect. 3. For these fit settings the systematic influence of clouds, $O_3$ and HCHO are discussed, corrected, and the remaining systematic influences are quantified in Sect. 4. In combination with an investigation of the statistical uncertainties (Sect. 5.1), the most suitable fit settings for an accurate retrieval within minor volcanic plumes is proposed in Sect. 5. The proposed retrieval algorithm is tested on multiple volcanic plumes of different emission strengths and at different latitudes in Sect. 6 to demonstrate its accuracy and efficacy. Lastly, conclusions are drawn in Sect. 7.

## 2 Methods

### 2.1 The TROPOMI instrument

The TROPOspheric Monitoring Instrument (TROPOMI, Veefkind et al., 2012) is onboard the satellite Sentinel-5 Precursor (S-5P) of the European Space Agency (ESA). S-5P is a sun-synchronous, polar-orbiting satellite with an orbital inclination of $98.7°$ on a low-Earth orbit (altitude: $824\,\text{km}$) launched on 13 October 2017. Its equator crossing time is roughly at $13:30\,\text{h}$ mean local solar time and it has a repeat cycle of 17 days. S-5P and its single payload TROPOMI are designed to determine and monitor the atmospheric composition. TROPOMI uses a push-broom method to scan the Earth's surface. The opening angle of $108°$ leads to a ground swath of approx. $2600\,\text{km}$ (separated to $450$ pixels) and allows for an almost complete daily coverage of the whole globe.

TROPOMI is a hyperspectral imaging spectrometer measuring radiance spectra via four different detectors in the ultraviolet $270 - 320\,\text{nm}$ (UV), UV-visible $310 - 500\,\text{nm}$ (UVIS), near-infrared $675 - 775\,\text{nm}$ (NIR), and short-wave-infrared $2305 - 2385\,\text{nm}$ (SWIR). A detailed overview of the spectral characteristics for each detector is given in Veefkind et al. (2012).

#### 2.1.1 Selection of spectra: TROPOMI UVIS

For the DOAS retrieval of BrO the L1B radiance spectra from the UVIS channel are used. The UVIS channel has a along-track pixel size of $7\,\text{km}$, which was reduced to $5.5\,\text{km}$ on 6 August 2019. There are $450$ across-track pixels, whose size varies from $3.5\,\text{km}$ at nadir up to $14.4\,\text{km}$ at large viewing angles. Thus, the highest spatial resolution is $5.5 \times 3.5\,\text{km}^2 = 19.25\,\text{km}^2$.

#### 2.1.2 Additional input data

For the investigation of systematic errors and their correction several additional input data-sets are used in this study.

First, for the investigation of the influence of clouds onto the retrieval and for its correction, there are two cloud data-sets used: the cloud height (CH) and the cloud fraction (CF). Both data-sets are calculated using the Fast Retrieval Scheme for Clouds from the $O_2$ A-Band (FRESCO, Compernolle et al., 2021) algorithm, which derives a radiometric cloud fraction and cloud pressure using the reflectance spectrum at 760nm (in the $O_2$ A-band) assuming a Lambertian cloud model. The FRESCO data products provided within the $NO_2$ operational product (van Geffen et al., 2021; NO2, 2021). While the cloud fraction is directly inferred within the FRESCO algorithm, the cloud height is calculated from the FRESCO cloud pressure (CP), surface pressure (SP) and surface altitude (SA) – all provided within the $NO_2$ product (van Geffen et al., 2021; NO2, 2021) – via the barometric pressure formula. To ensure the use of the latest FRESCO product, we used the reprocessed version of the $NO_2$ product, provided via the Sentinel-5 Precursor Expert Hub (https://s5pexp.copernicus.eu/). For this study, we choose the FRESCO cloud products, as they are easily accessible via the operational $NO_2$ product and are therefore available for the complete TROPOMI sensing period. However, also other cloud products are available (a comparison of TROPOMI cloud products can be found in Latsch et al., 2022), such as the operational cloud product (OCRA/ROCINN, Loyola et al., 2018)

are available (cf. Latsch et al., 2022) and could in principle be used. For example the use of the cloud fractions provided by the Mainz Iterative Cloud Retrieval Utilities (MICRU, Sihler et al., 2020) was tested in Warnach (2022) and found suitable for application within the retrieval presented here. However, unfortunately it is not available for the complete TROPOMI measurement period.

Second, for the study of the influence of HCHO onto the BrO retrieval, HCHO slant column densities (SCDs) – provided within the operational TROPOMI HCHO L2 product – are used (De Smedt et al., 2018). The HCHO SCDs are derived from a DOAS fit in a similar fit wavelength range as BrO ($328.5 - 346\,\text{nm}$). In order to minimize interference with BrO, the BrO SCD is fixed using a SCD derived in an independent pre-fit in a larger fit wavelength range ($328.5 - 359\,\text{nm}$) (De Smedt et al., 2018).

Third, for the estimation of the stratospheric column strength the $O_3$ VCD is used. The $O_3$ VCD is derived directly from the BrO DOAS fit. We favour this $O_3$ VCD over the operational $O_3$ Level 2 product (1) because it is more practical and most importantly (2) because in difference to our fit, the operational $O_3$ product does not include $SO_2$ within the DOAS fit and is therefore affected stronger by $SO_2$-$O_3$ spectral interference leading to high inaccuracies within volcanic plumes. Since there are two different $O_3$ absorption cross-sections as well as the two »Pukite«-pseudo absorbers included in the BrO DOAS fit (see Table 1) to account for the spectral absorption signal of $O_3$, the contributions of the individual fitted $O_3$ terms are combined to the complete slant column $S_{O_3}$ following Puķīte and Wagner (2016):

$$S_{O_3} = S_{O_3,223\text{K}} + S_{O_3,243\text{K}} + S_{O_3,\lambda}\,\lambda + S_{O_3,\sigma}\,\sigma_{O_3} \tag{1}$$

where the slant columns $S_{O_3,223\text{K}}$, and $S_{O_3,243\text{K}}$ correspond to the cross sections for two temperatures while the two last terms $S_{O_3,\lambda}$, $S_{O_3,\sigma}$ correspond to the two Pukite-term pseudo-absorbers (Puķīte et al., 2010), $\lambda$ to the wavelength, and $\sigma_{O_3}$ to the $O_3$ absorption cross-sections. For simplicity, the wavelength $\lambda$ and the cross-section $\sigma_{O_3}$ was taken only at the wavelength of the first maximum of the absorption cross section closest to the lower wavelength boundary (e. g. for the fit wavelength range $323 - 360\,\text{nm}$: $\lambda = 325.0\,\text{nm}$ and $\sigma_{O_3} = 1.5 \times 10^{-20}\,\text{cm}^2\,\text{molecules}^{-1}$).

The resulting $O_3$ SCD $S_{O_3}$ is then converted to the geometric $O_3$ VCD via the geometric following the formalism described in Sect. 2.2.

## 2.2 The DOAS fitting routine

The DOAS method was introduced by Perner and Platt (1979) and is based on the attenuation of light in the atmosphere described by the Beer-Lambert-Bouguer law. The DOAS method makes use of the fact that molecules are distinguishable by their unique absorption structure originating from electronic, vibrational, and rotational excitations. These structures usually contain high frequency structures, which can be distinguished from the broad band extinction features of atmospheric scattering processes such as Mie and Rayleigh scattering. Essentially, the optical density $\tau$ of the absorption of the trace gas in question is retrieved by comparing the measurement spectrum's intensity I($\lambda$) with a so called »reference spectrum« I$_0(\lambda)$:

$$185 \quad \tau = ln\left(\frac{I_0(\lambda)}{I(\lambda)}\right) = \sum_i \sigma_i(\lambda) \cdot \int_L c_i(l)\, dl + P(\lambda) = \sum_i \sigma_i(\lambda) \cdot SCD_i + P(\lambda) \qquad (2)$$

where $\sigma_i(\lambda)$ is the absorption cross-section and $c_i(l)$ the concentration of the trace gas i, while the polynomial term $P(\lambda)$ accounts for broad-band absorption and scattering, e. g. Rayleigh and Mie scattering.

The result of the DOAS analysis will be the difference of the columns of the trace gases between reference and measurement spectrum, the slant column density (SCD). DOAS thus requires only knowledge about the absorption cross-sections $\sigma$ of the relevant absorbing molecules. Traditional DOAS algorithms employ a combination of a linear (for the contribution of the ref. spectra) and non-linear (for spectral shift and distortion compensation) fit in order to determine the column of the absorber. However, for satellite applications, where huge data-sets are recorded daily, a linearisation of the non-linear DOAS fit was developed by Beirle et al. (2013) and is used here. The fitting algorithm used here was implemented and described in detail by Borger et al. (2020).

The slant column density (SCD) retrieved in the DOAS fit is the integrated concentration along the light path. As the light path depends on the measurement geometry, i. e. viewing angle, solar angle, and the radiative transfer, the SCD is converted to the vertical column density (VCD, the vertically integrated concentration) via the so called air-mass-factor (AMF):

$$VCD = \frac{SCD}{AMF} \qquad (3)$$

One simple approximation for the AMF is the geometrical AMF, depending only on the solar zenith angle (SZA) and viewing zenith angle (VZA):

$$AMF_{geo} = \frac{1}{\cos(SZA)} + \frac{1}{\cos(VZA)} \qquad (4)$$

The geometric AMF is a valid approximation for stratospheric light paths in the UV if the SZA does not exceed $70°$ (Burrows et al., 2011) and accounts for the satellite and solar viewing geometry. For tropospheric columns, usually a more complex derivation using radiative transfer calculations is performed, including the trace gas profile (i. e. the plume height in the case of a volcanic plume), surface albedo and cloud cover. For trace gases close to the surface the AMF can be up to one order of magnitude lower compared to the geometric approximation (Theys et al., 2017).

In this study we employ only geometric AMFs, since (1) the focus of the study is to look at systematic influences and background corrections and (2) as the main focus of volcanic BrO is to derive the $BrO/SO_2$ ratio, for which the AMF divergence will cancel out in first approximation.

## 2.3 Selection of reference region

Traditionally, DOAS retrievals from satellites use the direct solar spectrum recorded on each day as a reference spectrum. This ensures that the DOAS retrieval can yield absolute slant columns of the respective trace gas. However, the direct solar spectra

are obtained through a different optical input channel and traverses different optical parts (see Veefkind et al., 2012). Thus, these differences will cause stripes in the across-track direction (Richter and Wagner, 2001; Chance, 2007). While this can be corrected via so called »destriping« algorithm (e. g. Chance, 2007; Boersma et al., 2011; Hörmann et al., 2016), in recent times it has been found advantageous to use an earthshine spectrum as reference spectrum for the retrieval (Theys et al., 2017; Seo et al., 2019). Ideally, such an earthshine spectrum is calculated from radiances obtained at regions where the respective trace gas is not abundant. Thus, regions over the equatorial Pacific are used (Sihler et al., 2012; Hörmann et al., 2013, 2016; Seo et al., 2019), far-off potential anthropogenic pollution sources and remote from volcanoes. In this study, we expand the region an earthshine spectrum is calculated using the complete equatorial latitude band as a reference region ($20° $ S to $20° $ N and $180° $ W to $180° $ E). This large band is chosen to (1) increase the statistics and (2) ensure that the earthshine reference is least susceptible to variation in the detector response over the day, i. e. that the earthshine spectrum is representative for each longitudes.

For $SO_2$, the assumption that remaining $SO_2$ is negligible holds true for this reference area. For $BrO$ however, there is still a significant stratospheric background signal over the equatorial region with VCDs of roughly $3.5 \times 10^{13}\,\mathrm{molecules\,cm^{-2}}$, as stated by Richter et al. (2002) and also found within this study. Thus, using the earthshine reference will still yield differential SCDs (dSCDs) instead of absolute SCDs. However, the stratospheric signal will introduce an offset in the resulting slant column densities, which is only slowly changing in spatial direction. In this study, the focus is on local volcanic plumes. Volcanic plumes are singular events, whose (1) spatial patterns usually have a high contrast compared to the background, and (2) spatial extent is almost always much smaller compared to stratospheric patterns. Thus, the stratospheric signal can be removed effectively by performing a local background correction for $SO_2$ and $BrO$ (see Sect. 6.2).

However, an earthshine spectrum using the complete equatorial latitude band might include influences from volcanoes as well as biogenic or anthropogenic influences. A comparison between the use of the new expanded earthshine spectrum calculated from the complete equatorial region [$\pm20° $ N, $\pm180° $ E], the earthshine spectrum from the pacific equatorial region only [$\pm20° $ N, $135 - 105° $ W], as well as using an irradiance spectrum is shown in Fig. 2 for measurements over the equatorial region [$\pm20° $ N, $\pm180° $ E]. It can be seen that the retrieved VCDs show no difference or offset between all three fits (here the stratospheric influence in the irradiance data is eliminated for comparison by subtracting the median $BrO$ VCD). The fit root-mean-square (RMS), however, is about $25\,\%$ lower for the earthshine fit and roughly $6 \times 10^{-3}$. This RMS distribution is in very good agreement with RMS reported over a pacific equatorial region by Seo et al. (2019, Fig. 11b) who employed a DOAS earthshine fit based on a large pacific equator region [$\pm20° $ N, $150 - 240° $ E] independently from the fit presented in this study.

In order to ensure that the inclusion of volcanic plumes within the reference spectrum will not introduce a noticeable contamination into the reference spectrum, we investigated the difference between including and excluding volcanic areas onto the retrieved mean SCD over the equator. This is done for two days in the appendix (cf. Sect. A): 2 October 2021, where only several, small plumes are present (cf. Fig. A1a, red areas), representative of normal conditions, and 30 July 2018, where a very large plume stretched over a large portion of the equatorial region (cf. Fig. A2a, red areas), representative of exceptionally strong volcanic activity within the equatorial region. For the normal conditions on 2 October 2021, excluding the volcanic areas

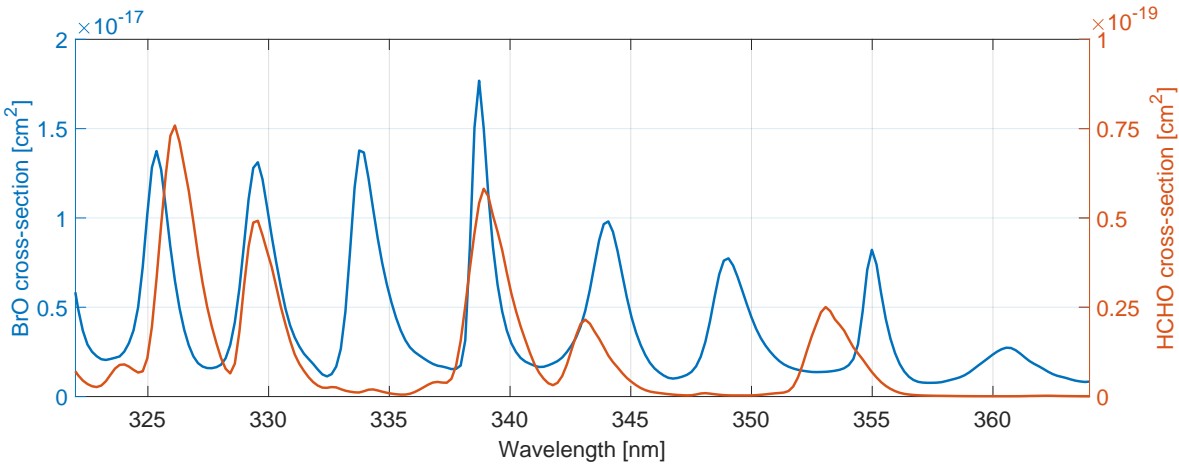

**Figure 1.** Absorption cross-section of $BrO$ (blue, based on Fleischmann et al., 2004) and $HCHO$ (red, based on Meller and Moortgat, 2000) convolved with a typical TROPOMI instrument spectral response function.

only leads to negligible changes in the $SO_2$ SCD (cf. Fig. A1c) and no detectable changes in the $BrO$ SCD (cf. Fig. A1d). For the exceptional conditions on 30 July 2018, there is a difference of several $10^{11}$ molecules $cm^{-2}$ visible between including and excluding the volcanic areas for $BrO$ SCDs (cf. Fig. A2d). However, this is 1.5 orders of magnitude below typical volcanic BrO columns ($1 \times 10^{13}$ molecules $cm^{-2}$) and therefore negligible. The same is the case for $SO_2$ SCD, where it is more pronounced, but still 1 order of magnitude below typical volcanic columns of $1 \times 10^{17}$ molecules $cm^{-2}$ (cf. Fig. A2c). Furthermore, the large plume on the 30 July 2018 stretches also over the pacific area typically used as a pacific reference region, e. g. $120°$ - $160°$ W, as used for the operational $SO_2$ product (Theys et al., 2017), or even more affected using $150°$ E $- 120°$ W (Seo et al., 2019). Thus, in this exceptional case using a pacific reference sector will also not be free of volcanic influence. To the contrary, in this case the influence is most likely stronger using a pacific reference area, as the plume affects a relatively larger portion of pixels within the reference area compared to our reference area which spans the complete equatorial band. It should further be noted that a constant offset expanding over all across-track detectors would be removed efficiently by our background correction algorithm and would therefore be irrelevant to our approach.

The earthshine reference spectra are calculated in the following way: Firstly, all radiances within the global latitude band [$180°$ W to $180°$ E] around the equator [$20°$ S to $20°$ N] are selected, as mentioned above. In a second step, the radiances are normalized with respect to their maximum intensity in order to prevent that scenes or geometries which result in a brighter backscatter radiation, e. g. cloudy scenes, to be systematically weighted stronger. In the last step, the mean of the normalized radiances is taken as the daily reference spectrum in the DOAS analysis. This process is performed separately for each spectra of the 450 across-track detector pixels of the TROPOMI instrument and for each day.

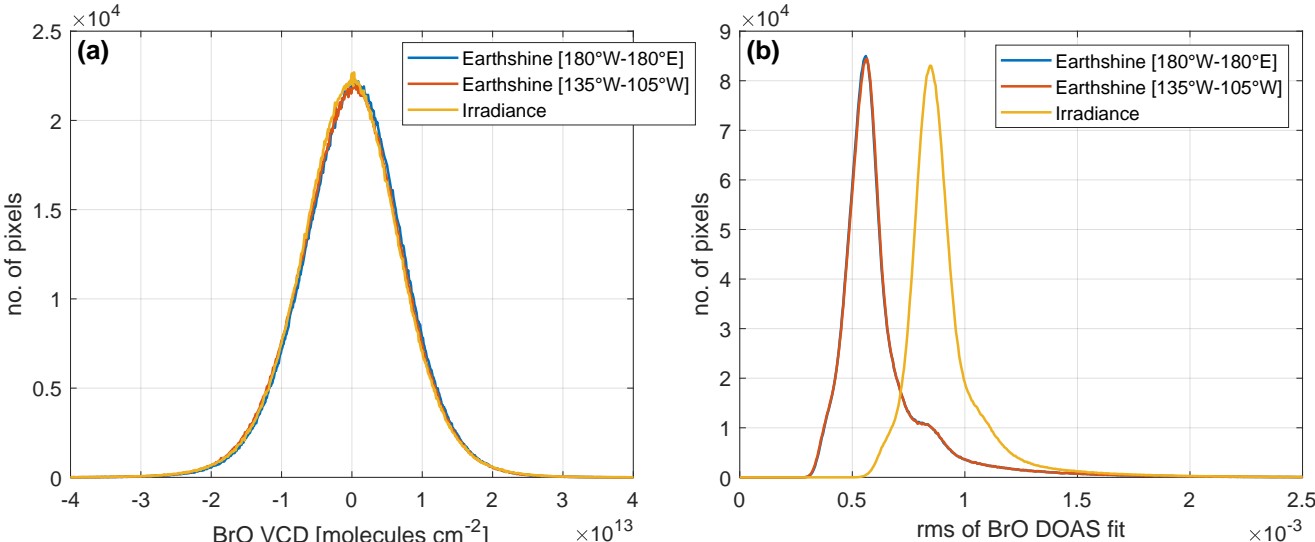

**Figure 2. (a)** Distribution of the BrO VCD and **(b)** the rms uncertainty of the DOAS fit for the complete equatorial region [$\pm 20°$ N, $\pm 180°$ E] on 1 October 2018 employing three different reference spectra: An earthshine spectrum calculated using the complete equatorial region [$\pm 20°$ N, $\pm 180°$ E] (blue), an earthshine spectrum calculated using the equatorial pacific region [$\pm 20°$N, 135-105°W] (red), as well as an irradiance spectrum (yellow). For comparability with the earthshine results, the median BrO VCD (corresponding to the median stratospheric column) is subtracted for the irradiance BrO VCDs.

## 3    Fit test setup

In the following sections, different wavelength ranges of the BrO DOAS fit are investigated. The performance with respect to the statistical variation, systematic spatial patterns of various origin and interference with HCHO is checked by comparing the results of four fit wavelength ranges, where the upper fit range boundary is unchanged at $360\,$nm, as varying the upper fit range boundary was found to be less critical for the fit performance (Vogel et al., 2013; Seo et al., 2019). This upper wavelength fit range boundary is thus chosen at $360\,$nm as chosen in Sihler et al. (2012), which is used as a baseline BrO DOAS fit. The lower wavelength boundary is varied, yielding the following wavelength ranges:

- *336 - 360 nm*: This smallest fit wavelength range includes four BrO absorption bands (see Fig. 1) and is the baseline wavelength range used by Sihler et al. (2012); Hörmann et al. (2013, 2016).

- *332 - 360 nm*: This fit wavelength range is chosen, because it adds one more BrO absorption band compared to the baseline, while adding no additional HCHO absorption band (see Fig. 1). This lower wavelength boundary is also used in Theys et al. (2011) (GOME-2) and Seo et al. (2019) (S-5P/TROPOMI). Theys et al. (2011) found that including this additional absorption band reduces the interference with HCHO.

- *323 - 360 nm*: This largest fit wavelength range was chosen, such that it includes seven BrO absorption bands while avoiding interference with very strong $O_3$ or $SO_2$ bands at wavelength below 320 nm. It was not used in DOAS satellite retrievals so far due to the strong $O_3$ absorption for high latitudes.

- *323 - 328.5 & 332 - 360 nm*: This encompasses also the largest fit wavelength range, but the HCHO absorption band around 329 nm, which overlaps with a BrO absorption band, is excluded in order to avoid HCHO interference (cf. Fig. 1).

All four fit wavelength ranges are implemented with two different fit settings, which differ only in the inclusion and exclusion of the HCHO absorption cross section in the DOAS fit. Thus, there is a total set of eight different fits compared. The fits including the HCHO absorption cross-section are labeled »HCHO«, the ones excluding HCHO are labeled »SR« (standard retrieval), i. e. the fit *HCHO 332 – 360 nm* corresponds to a DOAS fit evaluated in the wavelength range 332 nm to 360 nm with the HCHO absorption cross-section included, while the fit labeled *SR 332 – 360 nm* corresponds to the same DOAS fit just without the HCHO absorption cross-sections included. These labels will be used throughout this paper. The overview over the DOAS fit settings, and the included absorption cross-sections for both *SR* and *HCHO* fits are listed in Table 1. Since the sensitivity study described in Sect. 4 suggests that the fit *SR 323 – 360 nm* is the most suitable fit (cf. Sect. 5), most of the results in the main part of the paper will be shown only for this fit. The corresponding results for the other fits can be found in the appendix Sect. B2.

To illustrate the fit performance, the global BrO VCD map for 1 October 2018 for the fit *SR 323 – 360 nm* is shown in Fig. 3. Enhanced BrO occurs at high latitudes and is caused by the stratospheric signal, related to variations in the tropopause height (Sihler et al., 2012; Schöne, 2023). Hence, the variations of the BrO dSCD is directly linked to variations in the $O_3$ column, which in turn can be used as a proxy for the strength of the stratospheric BrO column (Sihler et al., 2012). The enhanced BrO columns in southern high latitudes can be additionally caused by tropospheric BrO events occurring during polar spring (Sihler et al., 2012; Herrmann et al., 2022). In the equatorial region there are also distinct spatial patterns in the BrO columns not originating from volcanic plumes. They might be related to clouds and will be investigated in more detail in Sect. 4.1. The reason for this cloud effect is not fully clear. The effect could be a spectroscopic artefact (e. g. via the Ring Spectrum) or a true shielding effect of a potential tropospheric BrO background column.

As all gradients in the BrO column of non-volcanic origin constitute a potential systematic error source, they ideally are removed via a background correction (e. g. via a spatial polynomial, Hörmann et al., 2013). In order to further improve the accuracy of the BrO retrieval, we investigate a more sophisticated correction scheme in Sect. 4.2.

# 4 Investigation of systematic effects

In this section, the systematic influences of following three effects onto the BrO retrieval are investigated and discussed for the eight different BrO fit settings: Potential systematic influences of $O_3$, influences of clouds (Sect. 4.1), and the spectral interference with HCHO (Sect. 4.3).

**Table 1.** Fit settings for the eight BrO DOAS retrievals: Each wavelength fit range are applied once including and excluding the HCHO cross-section. The proposed final wavelength fit range is highlighted in bold. The fit settings included also in the $SO_2$ DOAS fit are indicated by an asterisk in the column »incl. in $SO_2$ fit«.

| Wavelength fit ranges: | **$323 - 360$** nm |
| | $323 - 328.5$ & $332 - 360$ nm |
| | $332 - 360$ nm |
| | $336 - 360$ nm |
| Polynomial order: | 5 |
| Reference spectrum: | daily earthshine spectrum from equatorial region [$20°$ S $- 20°$ N; $180°$ W $- 180°$ E] (cf. Sect. 2.3) |

Absorption cross-sections:

| Species | Temperature | incl. in $SO_2$ fit[a] | Reference |
|---|---|---|---|
| $O_3$ | 223 K | * | Serdyuchenko et al. (2014) |
| $O_3$ | 243 K | * | Serdyuchenko et al. (2014) |
| $SO_2$ | 203 K | * | Bogumil et al. (2003) |
| BrO | 223 K | | Fleischmann et al. (2004) |
| $O_4$ | 203 K | | Thalman and Volkamer (2013) |
| OClO | 293 K | | Bogumil et al. (2003) |
| $NO_2$ | 220 K | | Vandaele et al. (1998) |
| HCHO[b] | 298 K | | Meller and Moortgat (2000) |

Pseudo absorbers:

| | | | |
|---|---|---|---|
| Ring, norm | 230 K | * | Wagner et al. (2009) |
| Ring, $\lambda$ | 230 K | * | Wagner et al. (2009) |
| Pukite, $O_3 \cdot \lambda$ | 223 K | * | Pukīte et al. (2010) |
| Pukite, $O_3{}^2$ | 223 K | * | Pukīte et al. (2010) |

Pseudo absorbers for instrumental effects:

| | | | |
|---|---|---|---|
| ISRF[c] | w, aw, k[d] | * | Beirle et al. (2017) |
| Shift & stretch | | * | Beirle et al. (2013) |

[a] For the $SO_2$ fit, the wavelength range is $312 - 324$ nm

[b] Not included in SR fits. Not included in the proposed final fit settings.

[c] Instrument Spectral Response Function

[d] w = width parameter; aw = asymetry parameter; k = shape parameter of a super-Gaussian distribution function

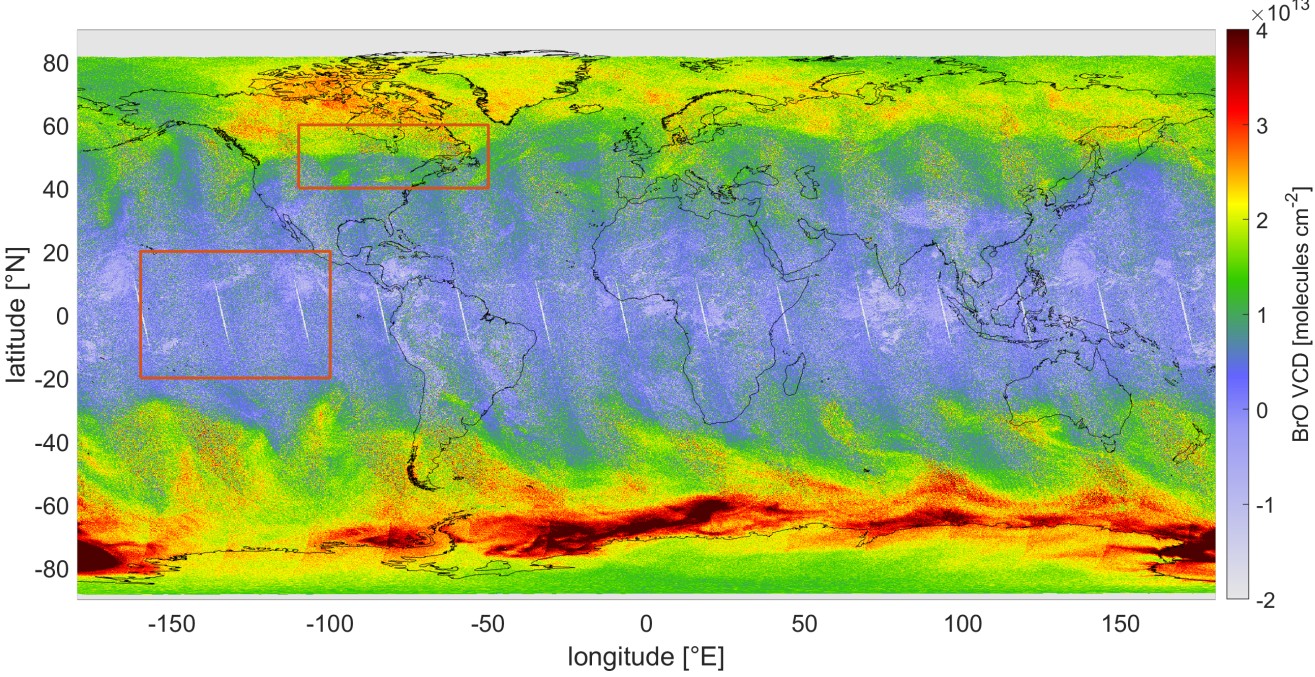

**Figure 3.** Global map of the BrO geometric VCD retrieved using the fit range *SR 323 – 360 nm* on 1 October 2018. The areas in the equator and mid-latitude region (marked by red rectangles) are used as case studies to investigate and quantify the influence of clouds and $O_3$ respectively (cf. Sect. 4.1).

## 4.1 Influence of clouds and $O_3$

The aforementioned structures in the global BrO map (cf. Fig. 3) are probably correlated to clouds (equatorial region) and the $O_3$ column (high-latitude regions). To showcase this, we look at two regions: First, a Pacific equatorial region ($[\pm 20° \, N, \, 160° -$ $100° \, W]$), where the cloud structures are assumed to dominate the systematic structures, and second a high-latitude region $[40° - 60° \, N, \, 110° - 50° \, W]$), where changes in the stratospheric column height are assumed to dominate the systematic structures in the BrO column. The former is depicted in Fig. 4 including both FRESCO CF (Fig. 4a) and FRESCO CH (Fig. 4c) as well as the BrO VCD for fit *SR 323 – 360 nm* (Fig. 4b). Furthermore, also the BrO VCD after applying the correction scheme for clouds and $O_3$ (described in Sect. 4.2) is included in Fig. 4d.

It can be seen that the patterns in the BrO VCDs correlate to the cloud patterns and lower BrO VCDs are retrieved in the presence of clouds. Furthermore, the impact of large cloud fractions and high cloud heights (e. g. around $15° \, N$, $110° \, W$) seems much stronger compared to large cloud fraction and low cloud height (e. g. arround $-15° \, N$, $110° \, W$). This suggests that both cloud fraction (which is a measure for the fraction of a pixel area covered by a cloud) as well as the top height of clouds is influencing the spectroscopic response and cause a different BrO VCD response. Ideally, we would like to correct the cloud influence based on knowledge of its physical cause. However, it is unclear if this is simply due to shielding of the tropospheric

BrO partial column below the clouds or a purely spectral response or has other origins. We therefore test a correction of these structures via an empirical scheme based on the two cloud parameters.

It is noteworthy that this decrease in the BrO column for cloudy scenes is an indication for cloud shielding of tropospheric BrO columns and can be used to derive and estimate for the tropospheric column, as done by Theys et al. (2011). However, as it is unclear if the cloud relation does not have a different origin and in this study we simply want to remove non-volcanic influences on the BrO column, this question is not indulged further here.

The high-latitude example is depicted in Fig. 5, including the FRESCO CF (Fig. 5a) as well as the $O_3$ VCD (Fig. 5c, taken from the BrO DOAS fit, cf. Sect. 2.1.2 for a description of the calculation of the $O_3$ VCD) and the BrO VCD (Fig. 5b). There is a clear gradient in the BrO VCD visible along $50°$ N, where BrO VCDs increase by more than $1 \times 10^{13}\,\mathrm{molecules\,cm^{-2}}$. This follows the gradient line of $O_3$ along the same latitude (cf. Fig. 5c). Both BrO and $O_3$ gradients coincide with the transition from mid-latitude to polar air masses indicated by the presence of the jet stream and a change in the tropopause height (cf. Fig. B1), where $O_3$ is higher and the stratospheric column is increasing. This gradient is overlayed by structures in the BrO VCD map south of $50°$ N, e.g. between $40 - 45°$ N and $85 - 95°$ W. These coincide with cloud cover (indicated by the cloud fraction) and show a positive BrO signal for high cloud cover. Both these gradients and structures in the BrO VCD are reduced after applying the correction scheme (cf. Sect. 4.2), as can be seen in the corresponding map of the corrected BrO VCD (Fig. 5d).

In the following, we will investigate the cloud and $O_3$ dependency for different latitudes. In order to investigate this potentially systematic effect of clouds on the BrO retrieval, the individual pixels are sorted in bins of $0.05$ cloud fraction and $500\,\mathrm{m}$ cloud height (CH-CF bins). This is done first for the complete equatorial latitude band ($[\pm 20°$ N, $\pm 180°$ E]). The magnitude of the BrO response is quantified by taking the mean BrO VCD of all pixels in each respective bin.

The mean BrO VCD is systematically decreasing, both for increasing CH and increasing CF. The corresponding plots for all eight fits can be found in the appendix Fig. B2. The relation of the mean BrO VCD both to CH and CF appears to be almost linear for all eight fit ranges, but with varying strength. Therefore this linearity seems independent of the fit settings chosen here allowing to derive a correction term, which will be described in Sect. 4.2.

While the cloud influence is very strong and rather linear for the equatorial region, there are also systematic structures in mid- and high latitudes (compare Fig. 3). Since these structures are not all related to the presence of clouds, but also to the $O_3$ columns (cf. Fig. 5) and possibly other factors, the same CH-CF binning is performed for $20°$ latitude bands over the whole globe, which are shown in Fig. 6 for the fit *SR 323 – 360 nm*. The almost linear dependency on CH and CF observed for the equator case is also found for neighbouring latitude bands from $30°$ S to $10°$ S and from $10°$ N to $30°$ N. For mid- to high latitudes north and south, this dependency of BrO on CH and CF is changing and/or is overlayed by other features.

While the general CH dependency of the BrO VCD (higher clouds, lower BrO VCD) is prevailing for high latitudes, the CF dependency appears to be changing its sign, so that a lower cloud fraction leads to a lower BrO VCD for $70°$ N - $90°$ N. However, there is an overlaying BrO signal for cloud heights between $1000$ and $5000\,\mathrm{m}$, independent of cloud fraction in this region, which complicates the interpretation. This feature is clearly visible for northern latitudes ($30°$ N - $90°$ N), and appears to shift to higher cloud heights for higher latitudes, i.e. this shifts from $0 - 3000\,\mathrm{m}$, to $1000 - 4000\,\mathrm{m}$ and to $2000 - 5000\,\mathrm{m}$

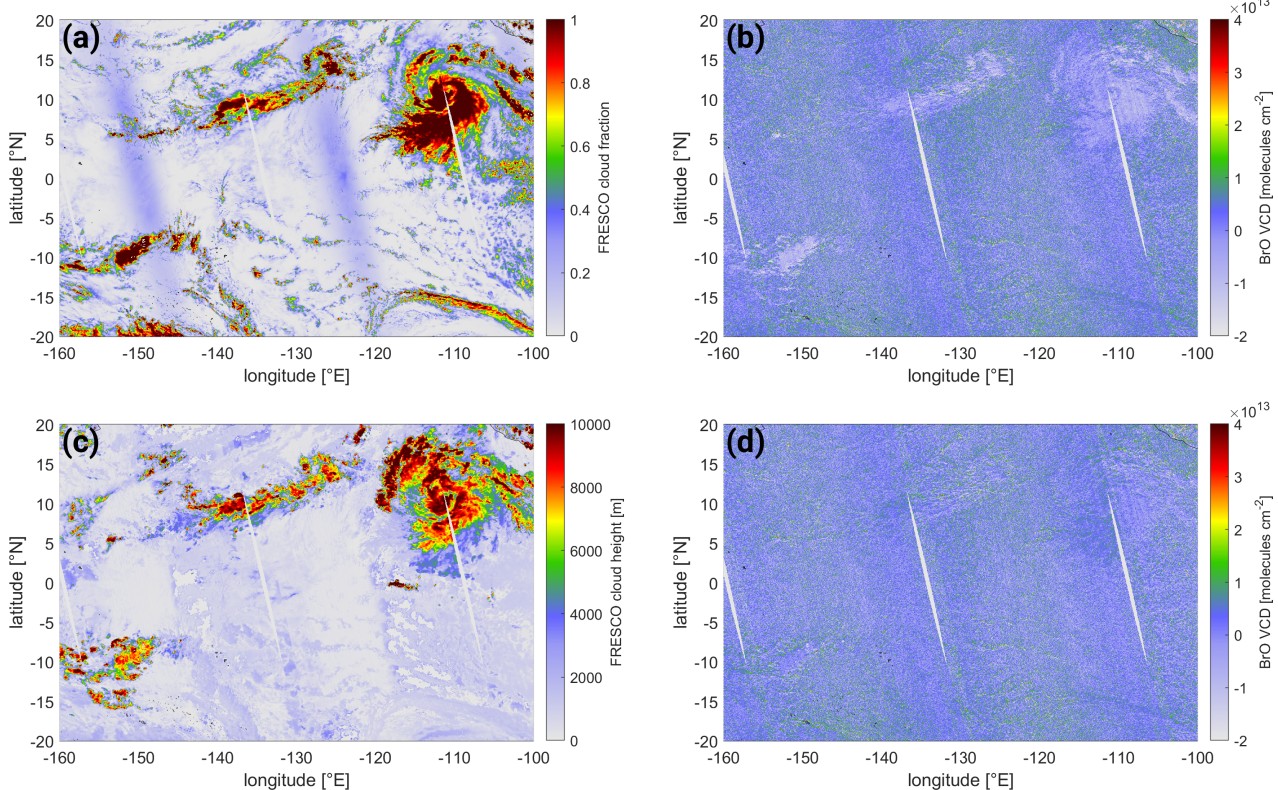

**Figure 4. (a)** FRESCO cloud fraction, **(b)** BrO VCD from fit *SR 323 – 360 nm*, **(c)** FRESCO cloud height, and **(d)** BrO VCD from fit *SR 323 – 360 nm* after applying the correction scheme. Data taken over the equatorial region [$\pm 20°$ N, $160° - 100°$ W] on 1 October 2018.

CH for $30°$ N - $50°$ N, $50°$ N - $70°$ N, and $70°$ N - $90°$ N respectively. These structures in the CH-CF plot are most likely caused by an increase in the stratospheric slant column (due to the high SZA at high latitudes). For latitudes south of $50°$ S there are additional structures occurring, which are most likely attributed to tropospheric BrO enhancements during polar spring.

In order to correct for these systematic effects on the BrO VCDs, different correction approaches are investigated in the next section (Sect. 4.2). Since the dependency on cloud parameters is latitudinal dependent, the correction approach is applied 365 to different latitude bands independently. Furthermore, since the systematic structures in high latitudes might be correlated to the stratospheric column, the inclusion of the $O_3$ VCD in the correction scheme as a first order indicator for the extent of the stratospheric column is tested.

### 4.2 Cloud and $O_3$ correction scheme

In order to investigate and correct the effects of the three parameters cloud fraction, cloud height, and $O_3$ VCD onto the BrO 370 VCDs, three multi-dimensional polynomial fitting schemes are tested on the BrO VCDs of 1 October 2018.

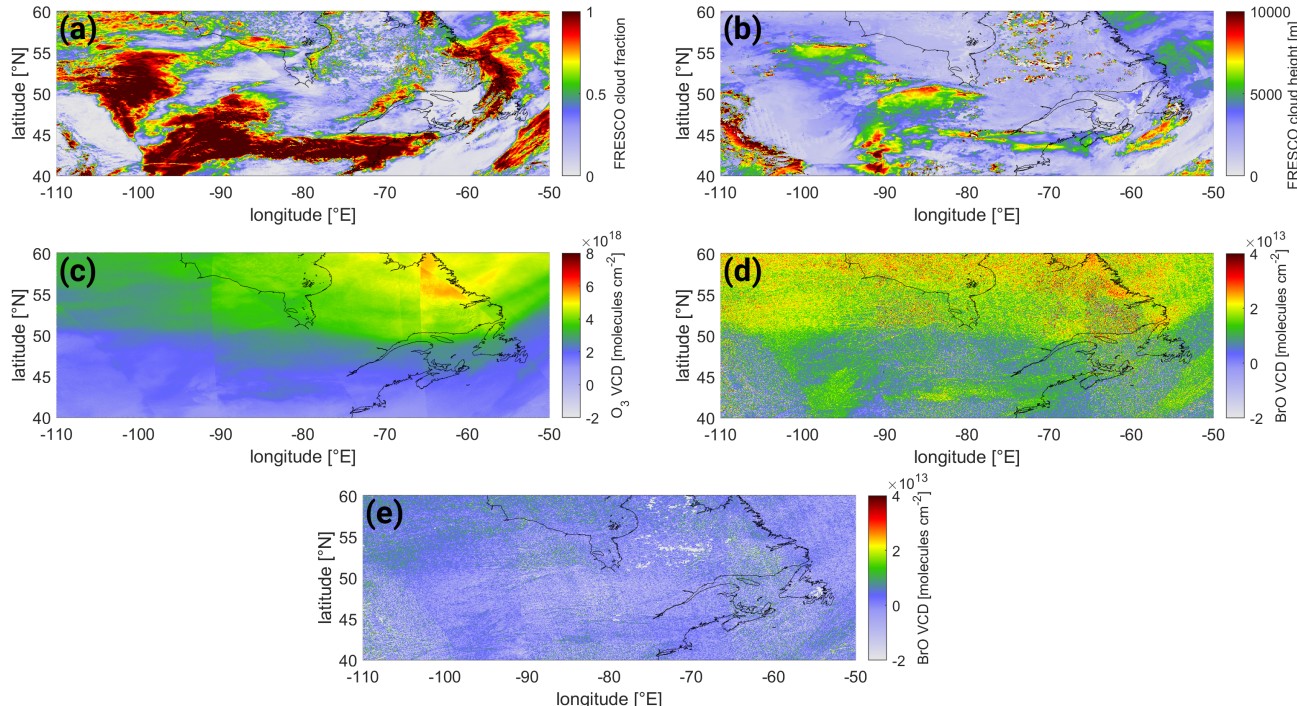

**Figure 5.** Maps of the **(a)** FRESCO cloud fraction, **(b)** FRESCO cloud height, **(c)** retrieved $O_3$ VCD, **(d)** BrO VCD from fit *SR 323 – 360 nm* without any correction and **(e)** BrO VCD from fit *SR 323 – 360 nm* after applying the correction scheme. Data taken over the northern high-latitude region $[40° − 60° \text{ N}, 110° − 50° \text{ W}]$ on 1 October 2018.

The three correction schemes are referred to as »ozone«, »latitude«, and »ozone latitude« in the following (see Table 2). All schemes include the two cloud parameters cloud fraction and cloud height derived from FRESCO. In addition, the schemes »ozone« and »ozone latitude« also include the $O_3$ VCD, taken from the BrO fit (cf. Sect. 2.1.2 for a detailed description about the calculation of the $O_3$ VCD). The $O_3$ VCD is then used in two of the three BrO correction schemes.

Each of the three BrO correction schemes is employed as a multi-dimensional polynomial P including the respective fitted parameters (see Table 2). The polynomial P comprising of p fitted parameters $x_1 \ldots x_p$ and their corresponding polynomial degrees $n_1 \ldots n_p$ is defined by in the following way:

$$P = \sum_{i_1=0}^{n_1} \ldots \sum_{i_p=0}^{n_p} c_{i_1,\ldots,i_p} \cdot x_1^{i_1} \ldots x_p^{i_p} \tag{5}$$

where $c_{i_1,\ldots,i_p}$ are the resulting fitted coefficients describing the dependency of the fitted quantity (in our case the BrO VCD) to the fitting parameters (in our case cloud parameters).


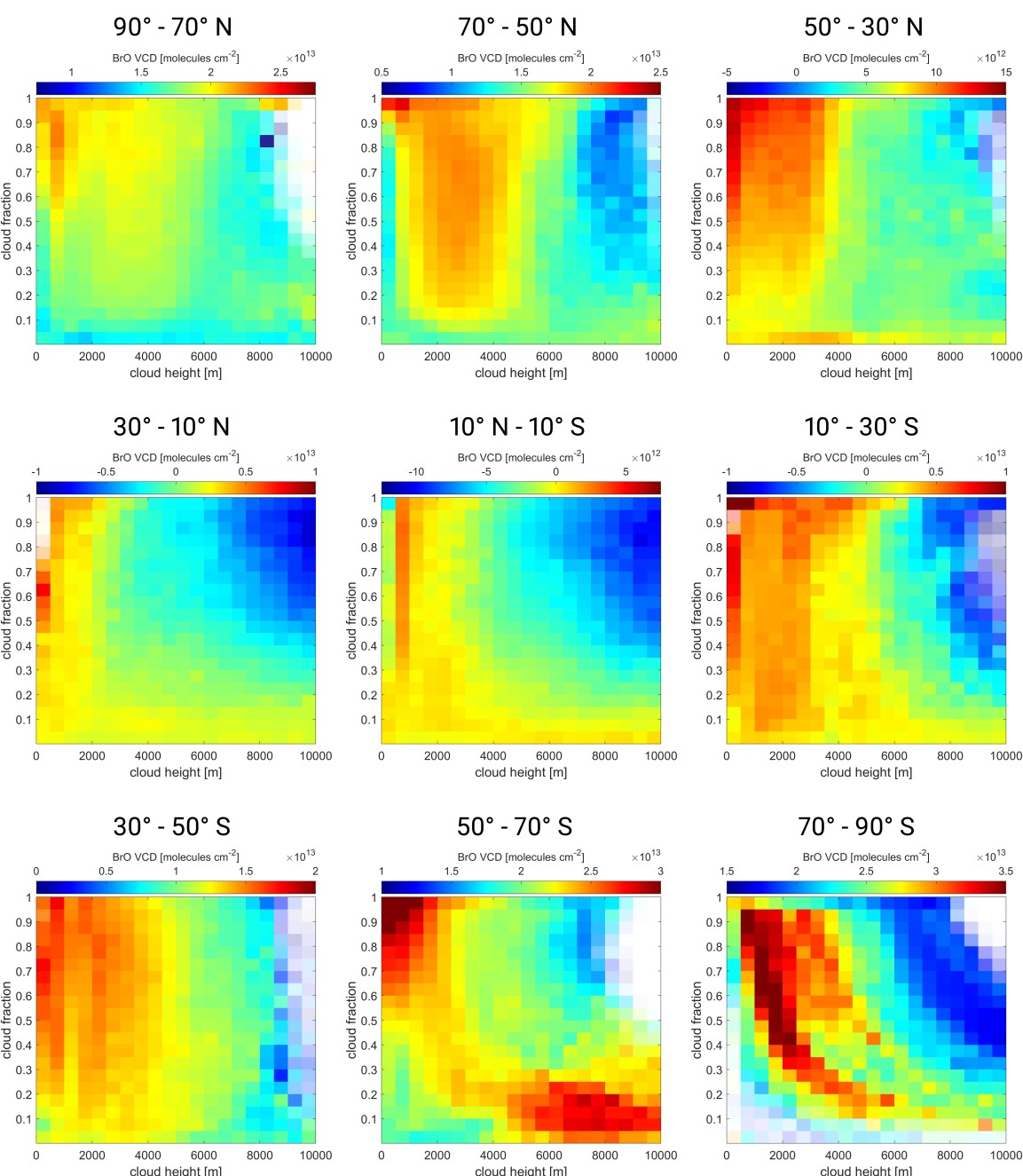

**Figure 6.** Mean BrO VCD for the cloud height - cloud fraction bins for the *SR 323 – 360 nm* fit for the global latitude bands in $20°$ steps from $90°$ N (upper left) to $90°$ S (lower right). The almost linear influence of both cloud height and cloud fraction is clearly visible for tropical latitudes (middle row). This changes for mid- to high latitudes (upper and lower row), where other influences related to the stratospheric column independent of the cloud parameters can also be seen. Note the different color scales for each subplot. They always encompass the same range of $2 \times 10^{13}$ molecules cm$^{-2}$, but start at different BrO VCDs.

**Table 2.** Overview over the Cloud/O$_3$ correction schemes. The applied latitude region, the fitted parameter and their polynomial degrees are listed for the three correction schemes.

| Name | latitude region | Fitted parameter | polynomial degree |
|---|---|---|---|
| ozone | 70° S to 70° N | cloud height | 2 |
| | | cloud fraction | 2 |
| | | O$_3$ VCD | 2 |
| latitude | 20° bands from 90° S to 90° N | cloud height | 2 |
| | | cloud fraction | 2 |
| ozone latitude | 20° bands from 90° S to 90° N | cloud height | 2 |
| | | cloud fraction | 2 |
| | | O$_3$ VCD | 2 |

This means for example, that for the correction scheme »latitude« (see Table 2), where the fitted parameters are the cloud height and the cloud fraction with polynomial degree of two for both parameter, the polynomial P looks like:

$$P = \sum_{i,j=0}^{2} c_{i,j} \cdot [\text{CH}]^i [\text{CF}]^j \tag{6}$$

$$P = \quad c_{0,0} \qquad\qquad + c_{1,0} [\text{CH}]^1 \qquad\qquad + c_{2,0} [\text{CH}]^2 \tag{7}$$

$$+ c_{0,1} [\text{CF}]^1 \qquad + c_{1,1} [\text{CH}]^1 [\text{CF}]^1 \qquad + c_{2,1} [\text{CH}]^2 [\text{CF}]^1 \tag{8}$$

$$+ c_{0,2} [\text{CF}]^2 \qquad + c_{1,2} [\text{CH}]^1 [\text{CF}]^2 \qquad + c_{2,2} [\text{CH}]^2 [\text{CF}]^2 \tag{9}$$

The principle output of this fit are the coefficients $c_{i_1,\dots,i_p}$. Afterwards the polynomial P is subtracted from the uncorrected BrO VCDs forming the corrected BrO VCDs.

In this section the polynomial P is tested for the complete global data-set. However, for the the schemes »latitude« and »ozone latitude« there are separate sets of coefficients $c_{i_1,\dots,i_p}$ for each 20° latitude band. Thus, for the retrieval of BrO inside a volcanic plume (Sect. 6) the correction coefficients can be calculated specifically for the latitude band around the volcano's latitude, ensuring a most accurate correction specific for each volcano's latitude.

In the following, the correction schemes will be applied and discussed for the fit *SR 323 – 360 nm* only. The correction schemes yield similar results for the other fits, but the fit *SR 323 – 360 nm* shows the best performance with respect to HCHO influence (see Sect. 4.3) and statistical uncertainties (see Sect. 5.1) and is therefore chosen as the best fit in the end.

The corrected global maps – after applying the correction schemes »ozone«, »latitude« and »ozone latitude« – are schown in Fig. 7a-c respectively for the settings *SR 323 – 360 nm* (excluding HCHO). Note that the scale of the BrO VCD color-code differs from the uncorrected map (cf. Fig. 3).

The systematic structures in the equatorial region are effectively removed both by »latitude« and »ozone latitude« schemes (Fig. 7b,c). For the »ozone« correction scheme (Fig. 7a), however, the systematic structures prevail reduced in magnitude.

The systematic structures in mid- to high-latitude are reduced for all three correction schemes. However, for the »latitude« scheme, which is the only one where the $O_3$ VCD is not included as a parameter, there are still significant structures visible for latitudes larger than $30°$ N and S. The structures are best reduced by the »ozone latitude« scheme. However, it is important to
note that even for this most sophisticated scheme, very strong BrO signals in the antarctic region uncorrelated with $O_3$ remain. These are probably tropospheric signals in the polar spring (compare Wagner and Platt, 1998; Sihler et al., 2012). If volcanic plumes are analysed for such conditions, the correction schemes developed in this study will not be able to separate the BrO in the volcanic plume form such events of enhanced tropospheric BrO.

For both, the »latitude« and the »ozone latitude« schemes, the correction is done separately for $20°$ latitude bands. It can
be seen that on the edges of the latitude bands jumps in the BrO VCD are occurring (see Fig. 7b,c). However, for the »ozone latitude« scheme (Fig. 7c) these only occur at high latitudes. As the latitude band's location can be adjusted freely, the latitude band will be chosen individually for each volcano in order to ensure that the volcano is located at its center and never on its edge (e. g. for Mt. Etna, Italy, located at $37°$ N, the latitude band $25° - 45°$ N is chosen for the background correction, cf. Sect. 6.3). Furthermore, only a $20°$ latitude band around the volcano should be considered. Only very large plumes – recurring
once every several years – might exceed this limit. For these cases the correction coefficient derived from the $20°$ latitude band surrounding the volcano are also applicable. For the development of a global BrO product including these corrections, the jumps on the edges of different correction bands might be avoided by an interpolation of the correction parameter with latitude.

In order to quantify the efficacy of the different correction schemes, the mean of the corrected BrO VCD is calculated for each CH-CF bin for each latitude band independently. The corresponding plots are depicted in Fig. 8 for all three correction
schemes as well as the bins without any correction. Ideally, there should be no or little dependency on the cloud parameters remaining after correction and no structures independent of the cloud parameter appearing.

All three correction schemes reduce the cloud dependency. It is noteworthy that the color-scale of the uncorrected plots is twice as large as the corrected ones.

It can be seen that the correction schemes working on each latitude band separately (»latitude« and »ozone latitude«) are
much better at reducing the dependencies, both on cloud fraction as well as on cloud height compared to the global »ozone« scheme, which produces new CH-CF gradients in equatorial regions. However, these gradients are weaker than the original gradients. The »latitude« correction reduces most cloud dependencies for latitude bins between $90°$ N and $50°$ S. However, cloud independent signals remain in mid- to high-latitudes. The inclusion of ozone in the correction scheme (»ozone latitude«) is further reducing the cloud dependencies, but most importantly drastically reduces the cloud independent structures. For this
scheme maximal differences in the CH-CF plots are typically $3 \times 10^{12}$ molecules cm$^{-2}$ for latitudes between $90°$ N to $50°$ S and $5 \times 10^{12}$ molecules cm$^{-2}$ for latitudes south of $50°$ S.

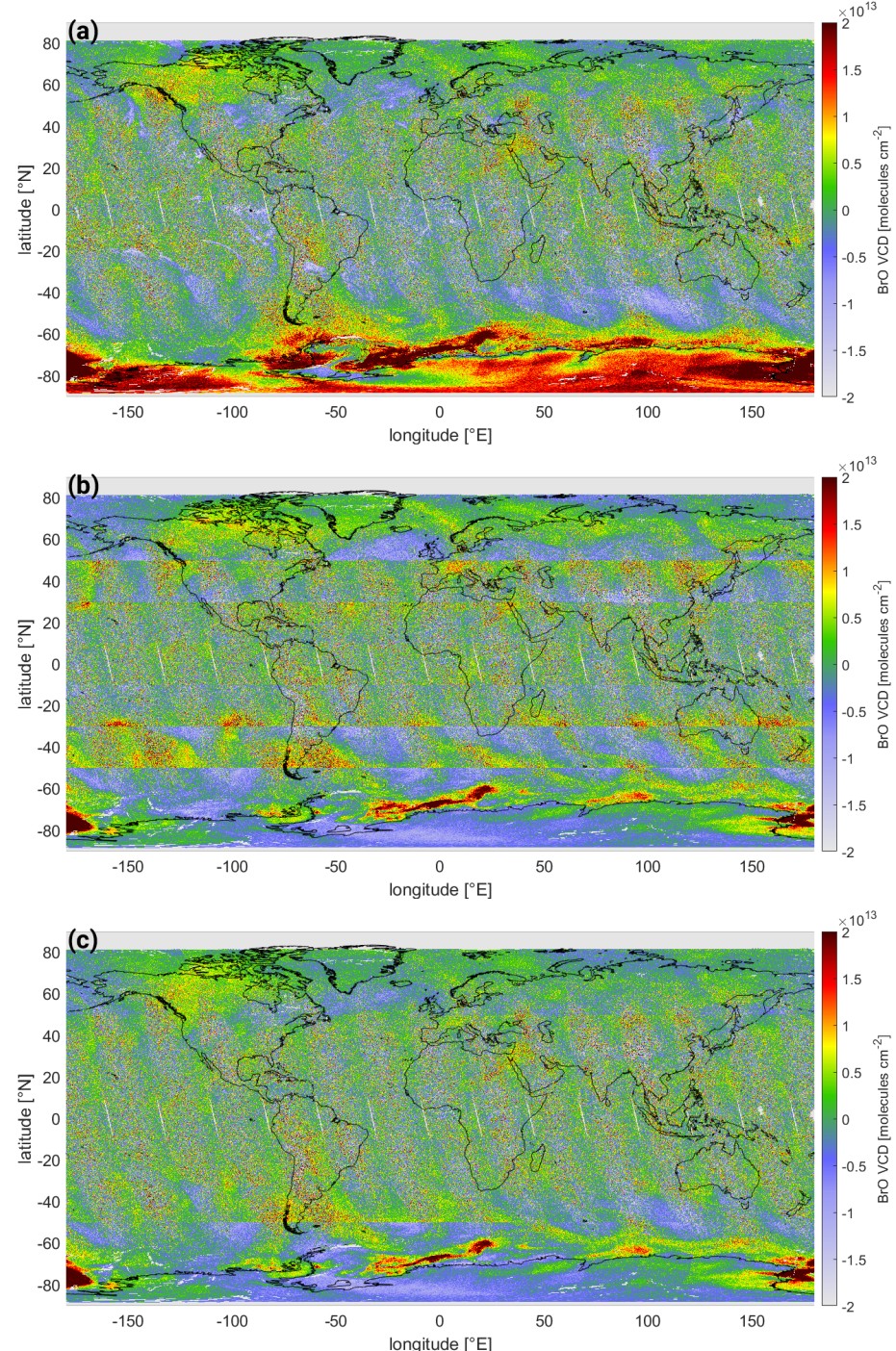

**Figure 7.** Global map of the BrO VCDs of the *SR 323 – 360 nm* fit **(a)** with »ozone « correction, **(b)** with CF and CH »latitude« correction only, and **(c)** CH, CF, and O$_3$ »ozone latitude« correction. The maps of the corresponding correction terms are plotted in Fig. B3.

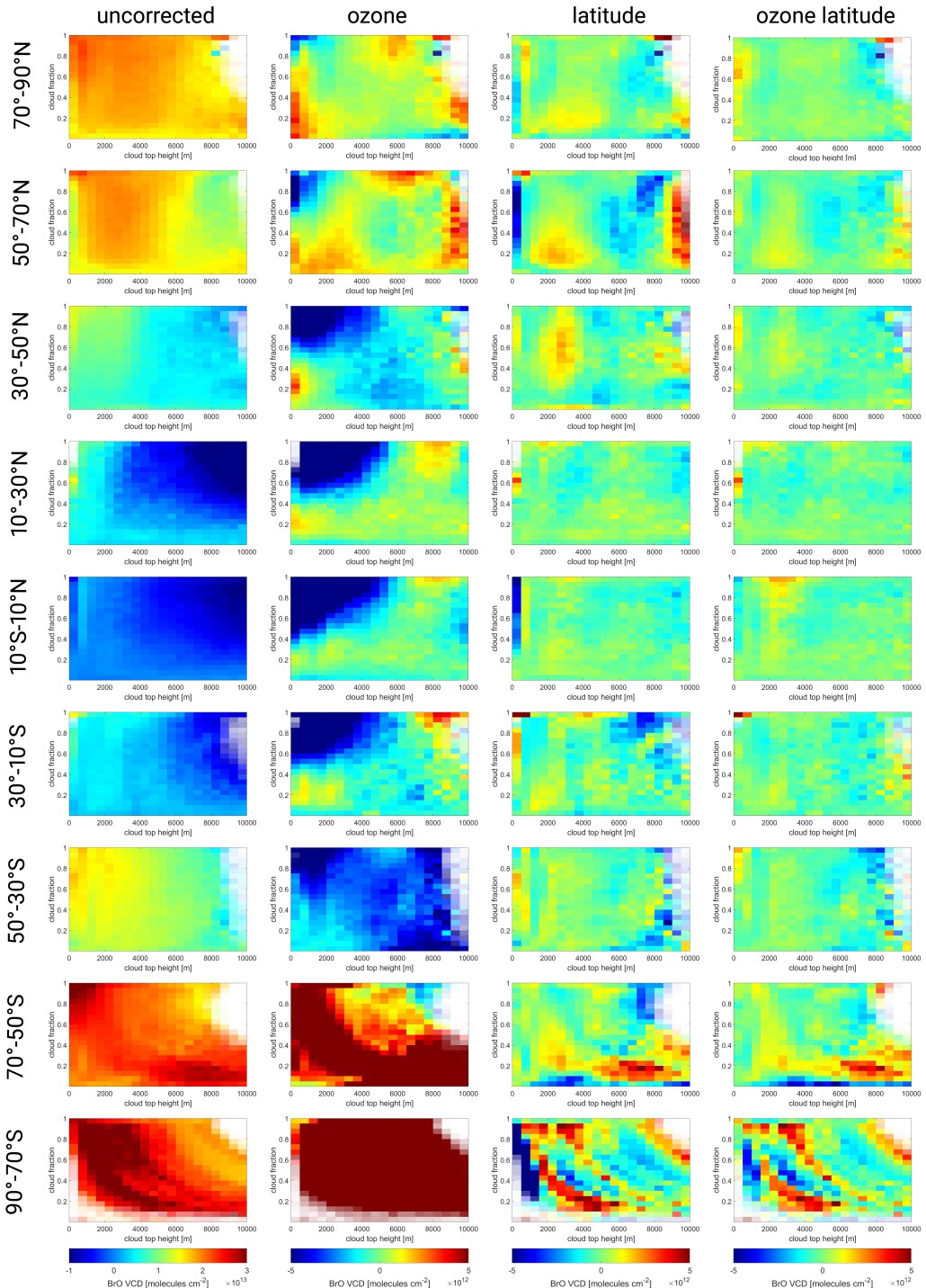

**Figure 8.** Mean BrO VCD as a function of the cloud height - cloud fraction bins for different latitude bands (rows from north to south) and correction schemes (from left to right): the uncorrected BrO VCD, and corrected BrO VCDs using ozone, latitude, and ozone latitude correction scheme. Note that the color scale range for the uncorrected plots is four times larger compared to the corrected plots.

Thus, including $O_3$ in the latitude dependent correction scheme (»ozone latitude«) appears to be the best correction both with respect to consistency in the global map (see Fig. 7c) and in the CH-CF bin plots (see Fig. 8). Therefore, this correction scheme is chosen for this study. In this approach the correction scheme is performed on separate $20°$ latitudinal bands. For the calculation of the correction factors, we thus propose to use the data on the $20°$ latitudinal band around the latitude of the volcanic plume.

From now on, all BrO VCDs used in this study will be by default corrected using the »ozone latitude« correction scheme.

## 4.3 Influence of formaldehyde

Formaldehyde (HCHO) is an organic compound emitted into the atmosphere mostly via biogenic emissions or biomass burning. Highest columns are observed in tropical regions (De Smedt et al., 2018). Since the cross-sections of BrO and HCHO in the UV range are very similar (see Fig. 1), atmospheric HCHO absorptions can cause systematic interference within the DOAS BrO retrieval.

The response of the eight different BrO DOAS fits to atmospheric HCHO absorption is investigated on the example of a biomass burning event in Africa between $0° - 20°$ S and $10°$ W $- 40°$ E on 1 October 2018.

The main questions is whether the inclusion of the HCHO absorption cross-section in the fit is necessary for the analysis of BrO in volcanic plumes, i. e. whether the exclusion of HCHO in the fit introduces a systematic offset in the BrO results in presence of enhanced HCHO absorption. One reason why it would be advantageous not to include the HCHO absorption cross-section is that the analysis yields a larger noise in the retrieved BrO SCDs if HCHO is included (as shown in Sect. 5.1). Equally important is the question, which fit wavelength range results in the lowest systematic influence of HCHO onto the BrO results and to quantify the strength of a potential false BrO signal in the presence of enhanced HCHO absorption.

As information on the atmospheric HCHO column, the HCHO SCDs from the operational S-5P/TROPOMI HCHO L2 product (De Smedt et al., 2018) are used. The cloud information (cloud fraction and cloud height) is taken from FRESCO as in the previous sections.

The HCHO SCD derived from the operational S-5P/TROPOMI HCHO L2 product for the selected region is plotted in Fig. 9 (noted »HCHO L2 SCD«, topmost left). There is a region of high HCHO columns exceeding $3 \times 10^{16}$ molecules cm$^{-2}$ on the African continent around $5°$ S and $29°$ E. Most importantly, this region is mostly cloud free (see Fig. 9, noted »FRESCO cloud fraction«, topmost right), ensuring minimal influence of clouds.

The BrO VCDs are corrected using the »ozone latitude« correction scheme described in Sect. 4.2. The corresponding maps for the eight fits are plotted in Fig. 9. The maps of the fits including HCHO (*HCHO*, Fig. 9 left column) do not reveal a distinct positive BrO VCD signal in the region of the HCHO event. To the contrary, there rather seems to be a systematic negative offset in this region, most visible for the fits *HCHO 323 – 360 nm* and *HCHO 323 – 328.5 & 332 – 360 nm*. The fits excluding HCHO (*SR*, Fig. 9 right column), show a positive offset in the region of high HCHO SCDs. This is most prominent for the fit range *SR 332 – 360 nm* (fourth from top, right column), where the BrO VCDs exceed $1 \times 10^{13}$ over a large area.

In order to investigate the potential systematic response of the BrO retrieval onto HCHO, the pixels are separated into bins of HCHO SCD and cloud fraction. In order to separate the HCHO effect from potential cloud influence (which is also an

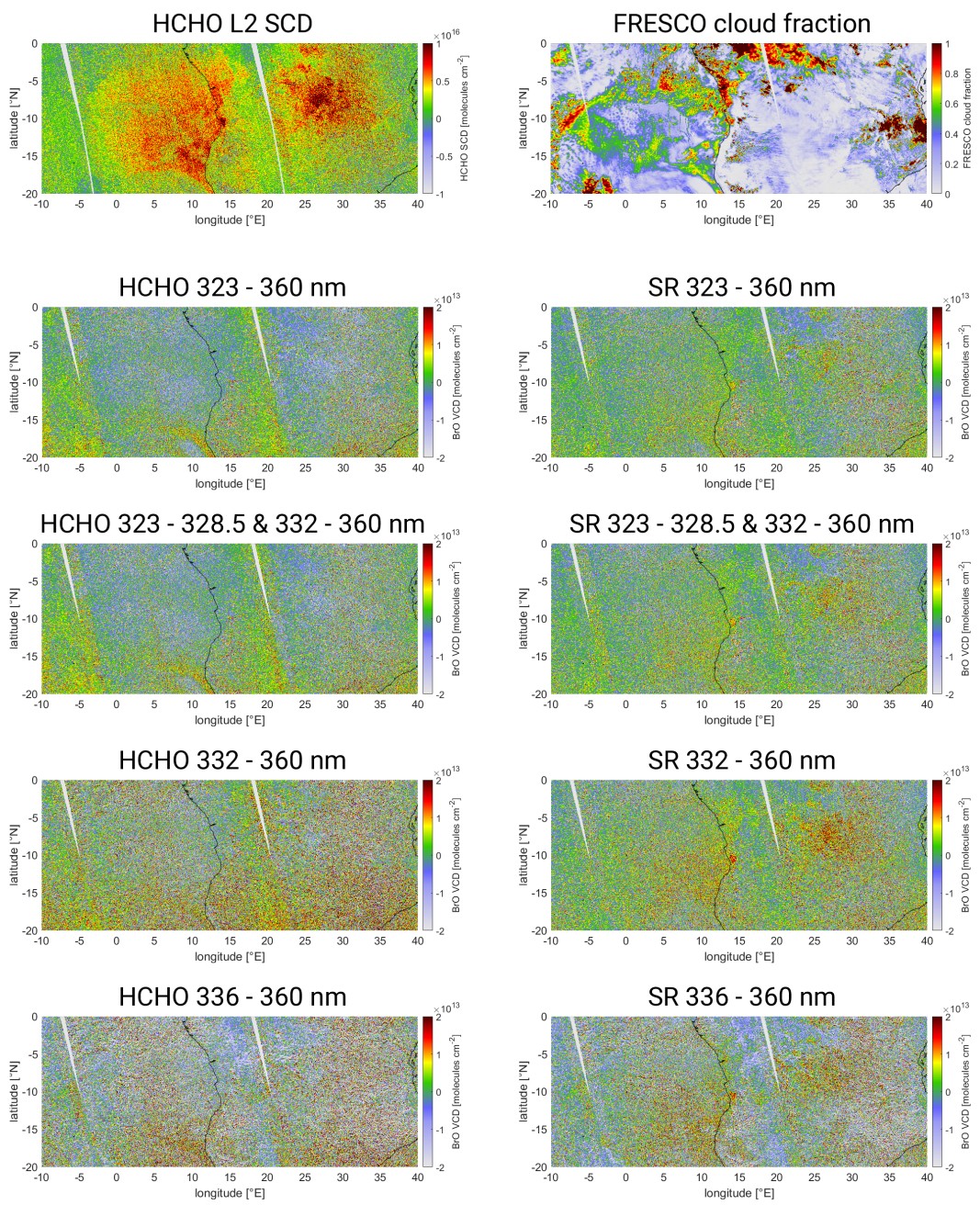

**Figure 9.** Maps of the auxiliary data and the BrO VCD for the eight fits over the African biomass burning region on 1 October 2018. *Topmost row, auxiliary data:* Operational S-5P/TROPOMI HCHO SCD (left) and FRESCO cloud fraction (right). *Below,* BrO *VCD:* BrO VCDs for the eight different fit ranges. The four fit wavelength ranges are listed from second top row to bottom row: *323 – 360 nm*, *323 – 328.5 & 332 – 360 nm*, *332 – 360 nm*, and *336 – 360 nm*. *Left column*: fits including HCHO (*HCHO*); *right column*: fits excluding HCHO (*SR*).

error source in the operational HCHO fit, De Smedt et al., 2018), at this point only cloud free pixels are considered, i. e. cloud fractions below 0.1. The resulting mean BrO VCDs for the different HCHO bins for all eights fits are shown in Fig. 10. For the fits including the HCHO cross-section (*HCHO*, solid lines in Fig. 10) the BrO VCD is anti-correlated to the HCHO SCD, i. e. negative HCHO SCDs are found together with positive BrO SCDs which decrease almost linearly towards higher HCHO
SCDs. This is consistent for all four fit ranges. However, *HCHO 332 – 360 nm* and *HCHO 336 – 360 nm* shows the strongest response, while the fits starting at $323 \, \mathrm{nm}$ show a slightly weaker response. The fits not including the HCHO cross-section (*SR*, dashed lines in Fig. 10) show a positive response to HCHO, i. e. BrO SCDs increase for increasing HCHO SCDs. However, this only occurs for high HCHO SCDs and varies strongly for the different fit ranges. The effect is strongest for the fit range *SR 332 – 360 nm*, where BrO VCDs exceed $1 \times 10^{13} \, \mathrm{molecules \, cm^{-2}}$, whereas it is much less pronounced for the other fit ranges,
where maximal BrO VCDs are in the range of $5 \times 10^{12} \, \mathrm{molecules \, cm^{-2}}$. In addition, for low HCHO SCDs the BrO VCD changes only slightly or remains constant. This is most prominent for the fit range *SR 323 – 360 nm* (green dashed line). Here, the BrO VCDs are zero for HCHO SCDs below $1.5 \times 10^{16} \, \mathrm{molecules \, cm^{-2}}$. For higher HCHO SCDs, the BrO VCDs slowly increase, reaching about $5 \times 10^{12} \, \mathrm{molecules \, cm^{-2}}$ for maximal HCHO SCDs of about $4 \times 10^{16} \, \mathrm{molecules \, cm^{-2}}$. Interestingly the fit range *SR 323 – 328.5 & 332 – 360 nm*, in which an overlapping BrO/HCHO absorption band is left out compared to *SR*
*323 – 360 nm* and thus a weaker response would be expected, shows a higher false positive signal.

In order to include also cloudy scenes, the mean BrO SCD is calculated for the complete HCHO-CF bins. The corresponding plots for the eight fits are shown in Fig. B4, where the four different fit wavelength ranges are plotted from left to right. The fits excluding the HCHO cross-section are in the upper row, while the fits including the HCHO cross-section are plotted in the lower row.

Both effects, the decreasing of BrO SCDs with increasing HCHO SCD for the fits including the HCHO cross-section (*HCHO*) as well as the opposite behavior of the fits not including the HCHO cross-section (*SR*), prevail also for cloudy scenes, but are sometimes weaker. Nevertheless, for volcanoes in regions where HCHO columns are high, it should be taken care for spatial HCHO patterns overlapping with the volcanic plumes (identified by its $SO_2$ signal).

It can be thus concluded, that the inclusion of the HCHO cross-section in the BrO DOAS retrieval is not necessary. Even
though there can be a systematic false positive BrO response produced by very large HCHO columns, this effect is very low for the fit range *SR 323 – 360 nm*, with no response for HCHO SCDs below $1.5 \times 10^{16} \, \mathrm{molecules \, cm^{-2}}$ and a response not exceeding $5 \times 10^{13} \, \mathrm{molecules \, cm^{-2}}$ for very large columns. HCHO columns above $1.5 \times 10^{16} \, \mathrm{molecules \, cm^{-2}}$ are only observed for tropical biomass burning events, mostly ocurring around the equator, i. e. the systematic impact of HCHO on the BrO retrieval using the *SR 323 – 360 nm* fit is negligible for most cases.

## 4.4 Quantification of the systematic effects

In order to quantify the uncertainty originating from remaining systematic uncertainties onto the retrieval, i. e. the potential »false« BrO signal created by clouds, $O_3$, or other systematic effects, we look at the BrO VCD of the fit *SR 323 – 360 nm* (after the correction with the »ozone latitude« correction scheme).

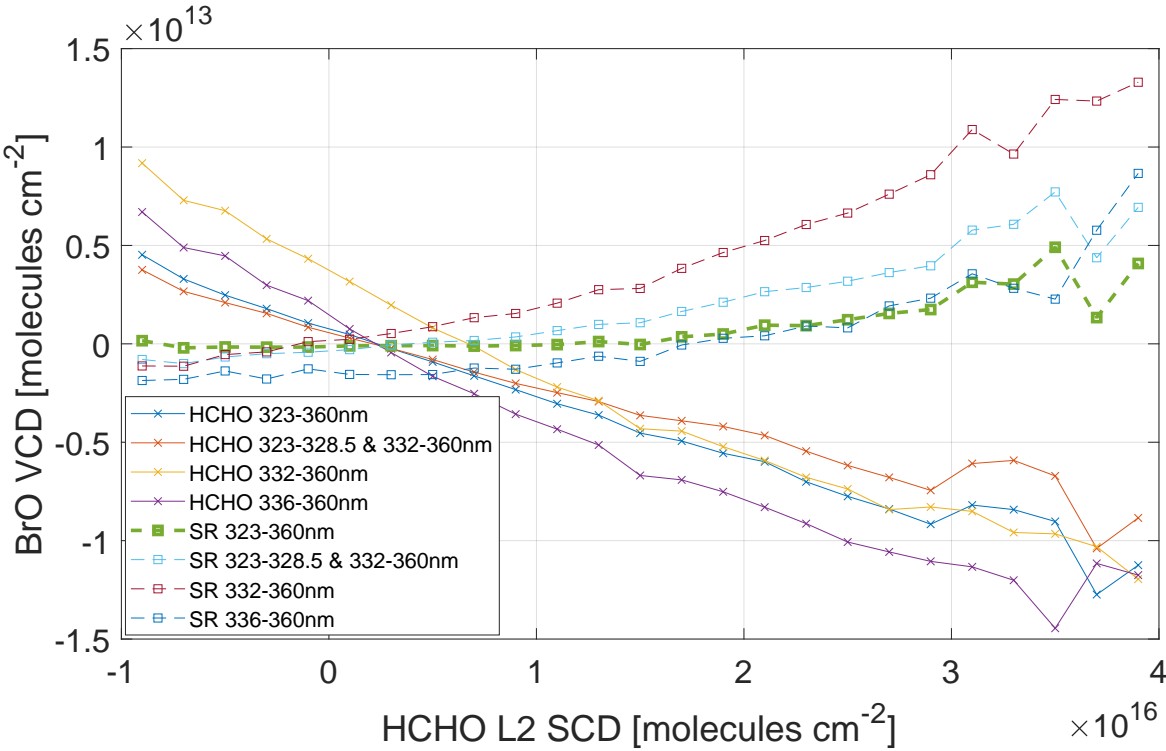

**Figure 10.** Mean BrO VCD as a function of HCHO SCD for the African biomass region on 1 October 2018 for the eight different fits including HCHO (*HCHO*) and excluding HCHO (*SR*). Only cloud free pixels (cloud fraction $< 0.1$) are considered.

The uncertainties of any data-set is comprised of both statistical and systematic uncertainties. By averaging over neighbour-

ing pixels the statistical variation can be gradually reduced and often the remaining uncertainties can be attributed to systematic effects. The statistical variation can be quantified by a Gaussian fit over the BrO VCD distribution and the statistical variation then estimated by the standard deviation. Statistical fluctuation can be reduced by spatial binning of neighbouring pixels, which reduces the standard deviation of the binned data by 1/square root of the number of pixel binned. Systematic effects however can generally not be eliminated spatial averaging/binning. Thus, ideally only the systematic effects remain for high binning

factors and the deviation from the expected decrease can be used as an estimate for the order of magnitude of systematic effects.

In order to estimate the systematic uncertainties, we employ a spatial binning of neighbouring pixels using binning factors of 1 to 100 in both spatial dimensions, corresponding to 1 to 10000 pixels binned respectively. For each binning factor as well as for each latitude band a Gaussian fit is done separately. The resulting standard deviations are plotted in Fig. 11.

It can be seen in Fig. 11 that the standard deviation is decreasing slower for increasing binning factors (square root of

the number of pixels binned) compared to the purely statistical model (black line in Fig. 11a) for all latitudes. For binning factors exceeding 20 (i. e. more than 400 pixels are binned together) the statistical model predicts a standard deviation of less than $5 \times 10^{11}\,\text{molecules}\,\text{cm}^{-2}$, while the measured data shows a standard deviation between 1 and $6 \times 10^{12}\,\text{molecules}\,\text{cm}^{-2}$

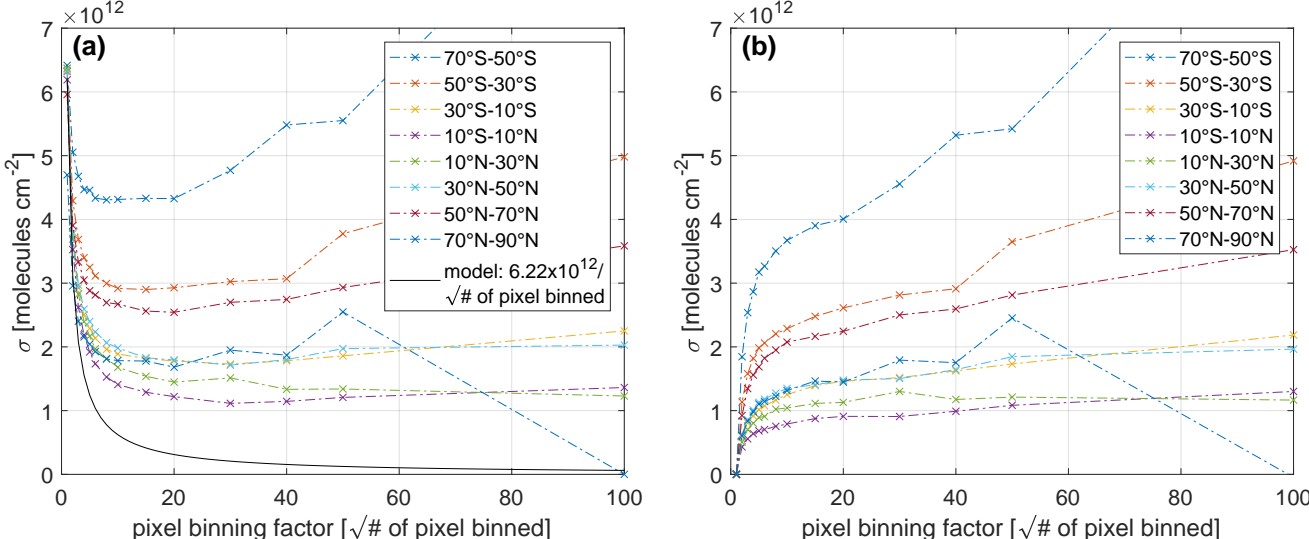

**Figure 11. (a)** The standard deviation of the BrO VCD distribution for different binnings for the seven latitude bands (colored lines). Also shown are the results of a model assuming that the total variation would be purely statistically distributed ($\frac{1}{\sqrt{N}}$, N=number of pixels binned, black line). For illustration purpose only the model curve for the equatorial region $[10°\,\text{S} - 10°\,\text{N}]$ is plotted. **(b)** The difference to the statistical model is plotted for the seven latitude bands. For each latitude band an individual model curve based on the respective standard deviation for pixel binning factor of 1 is used.

depending on the latitude. Thus, this remaining variance can be attributed to systematic effects. Around the equator ($10°\,\text{S}$ and $10°\,\text{N}$, Fig. 11b, violet square) the minimum standard deviation is almost $1 \times 10^{12}\,\text{molecules cm}^{-2}$, and this is chosen to be representative for the equator region. For the mid-latitudes below $50°$ latitude (triangles in Fig. 11), the bands on $10 - 30°\,\text{N}$ (green triangle), $30 - 50°\,\text{N}$ (light blue triangle), and $10 - 30°\,\text{S}$ (yellow triangle), the standard deviation is in the range of $1 - 1.5 \times 10^{12}\,\text{molecules cm}^{-2}$, while it is $2.5 \times 10^{12}\,\text{molecules cm}^{-2}$ for the band $30 - 50°\,\text{S}$ (light red triangle). We therefore choose the uncertainty to be $1.5 \times 10^{12}\,\text{molecules cm}^{-2}$ for mid-latitudes. For the high-latitudes north of $50°\,\text{N}$ (dark red diamond and dark blue circle) and south of $50°\,\text{S}$ (blue diamond), there is a large difference. The northern high latitudes (dark red diamond and dark blue circle) show a standard deviation of $2.5 \times 10^{12}\,\text{molecules cm}^{-2}$ and $2.5 \times 10^{12}\,\text{molecules cm}^{-2}$, similar to the latitude band $30 - 50°\,\text{S}$ (light red triangle). The same latitude band in the south ($50 - 70°\,\text{S}$, blue triangle), where the polar spring induces tropospheric BrO enhancements, which are not accounted for by the correction, yields a much higher standard deviation of $4.0 \times 10^{12}\,\text{molecules cm}^{-2}$. For high-latitudes, we therefore choose an uncertainty of $2.5 \times 10^{12}\,\text{molecules cm}^{-2}$ under normal conditions and an uncertainty of $4.0 \times 10^{12}\,\text{molecules cm}^{-2}$ for polar spring conditions. The latitude band $70 - 90°\,\text{S}$ did not yield a meaningful gaussian fit due to the strong interference of enhanced tropospheric BrO event in polar spring. These uncertainties are then used as general systematic uncertainties in Table 3.

It is noteworthy that for high latitude bands of $70°\,\text{S} - 50°\,\text{S}$, $50°\,\text{S} - 30°\,\text{S}$ and $50°\,\text{S} - 70°\,\text{N}$, there is even an increase in the standard deviation for binning factors above 20. This is due to the lower number of total pixel in this region. Due to the limited

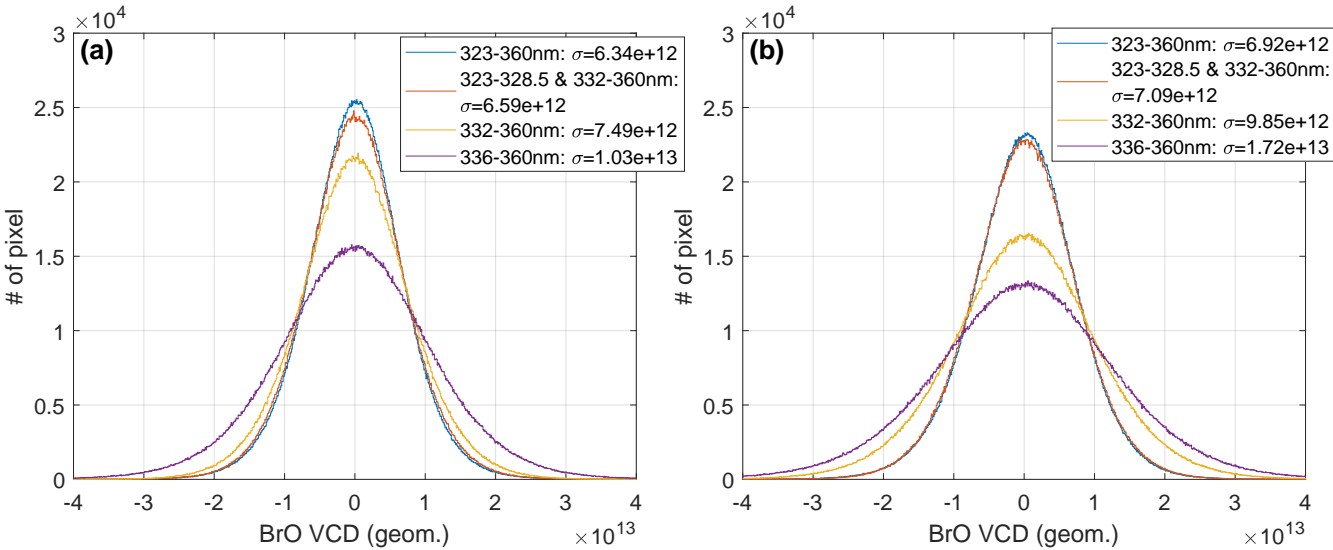

**Figure 12.** Distribution of BrO VCDs **(a)** for the four fits excluding HCHO (*SR*) and **(b)** including HCHO (*HCHO*). The corresponding standard deviations are given in the legend. All pixels on 1 October 2018 in the equator band between 20° S and 20° N over the whole longitude band are considered.

light availability in these regions (solar zenith angle $> 90°$), which will lead to an even smaller number of data-points after binning. This effect can be seen for all latitudes, where a slight increase in the standard deviation compared to lower binning factors is found.

## 5 Proposed fit settings for the retrieval of BrO in volcanic plumes

In addition to the systematic influences onto the BrO retrieval, the statistical uncertainties of the eight BrO DOAS fits also has to be taken into account in deciding which fit settings is optimal.

### 5.1 Quantification of the statistical uncertainties

In order to investigate the statistical performance of the BrO DOAS fit, the distribution of BrO VCDs in the equatorial region [$\pm 20°$ N, $180°$ W - $180°$ E] is examined for all eight fit settings (see Fig. 12). The distributions are separated into the fits excluding the HCHO cross-section (*SR*, left), and the ones including the HCHO cross-section (*HCHO*, right), so that the same fit wavelength ranges have the same colors in both subplots. Increasing the fit wavelength range towards lower wavelength leads to a significant reduction in the statistical variation for both fit settings. For the *SR* fits the smallest fit range (*SR 336 – 360 nm*, which is the baseline fit range from Sihler et al. (2012)) shows a standard deviation $\sigma$ of $1.03 \times 10^{13}$ molecules cm$^{-2}$, which subsequently decreases to $7.49 \times 10^{12}$ molecules cm$^{-2}$ (*SR 332 – 360 nm*) and finally $6.34 \times 10^{12}$ molecules cm$^{-2}$ (*SR 323 –*

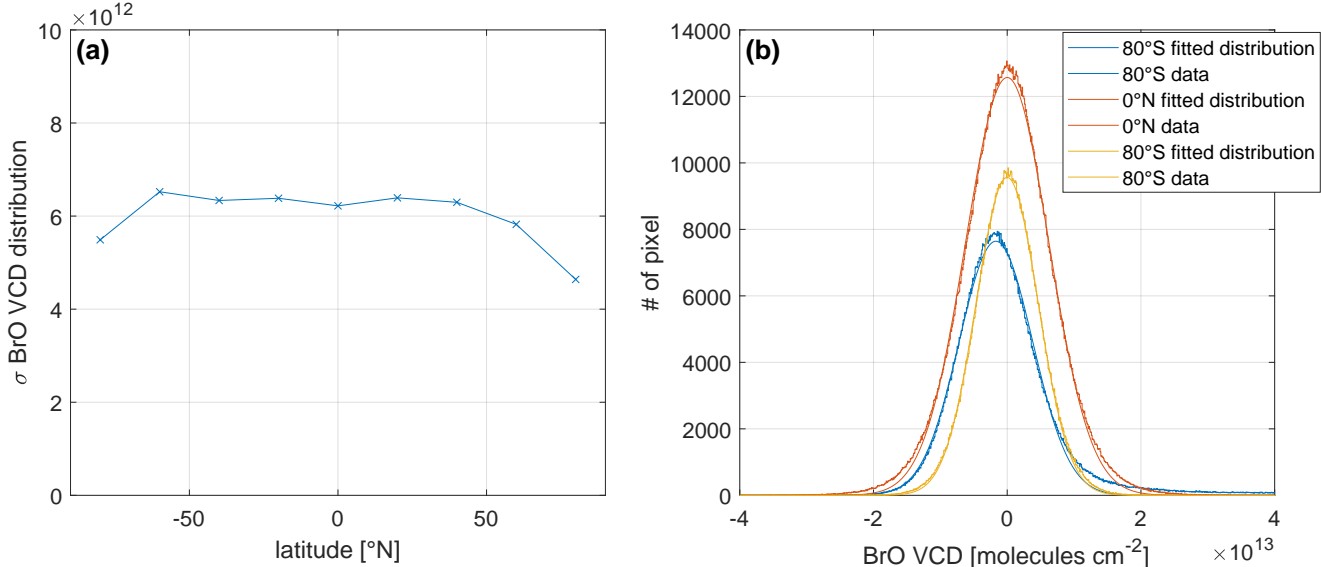

**Figure 13.** Latitudinal variation of the statistical uncertainty. **(a)** Standard deviation $\sigma$ of the distribution of the BrO VCD of fit *SR 323 – 360 nm* for different latitude bands on 1 October 2018. **(b)** BrO VCD distribution of fit *SR 323 – 360 nm* for $80°$ S (blue), $0°$ S (red), and $80°$ N (yellow). The corresponding fitted gaussian distributions are indicated by a solid line in the respective color.

$360\,nm$) when the wavelength range is expanded. This is to be expected as a larger wavelength range increases the information content.

In comparison to the *SR* fits, the *HCHO* fits have a larger standard deviation for all wavelength ranges, leading to a minimal $\sigma$ of $6.92 \times 10^{12}$ molecules cm$^{-2}$ (compared to $6.34 \times 10^{12}$ molecules cm$^{-2}$ for the corresponding *SR* fit) and a maximum $\sigma$ of $1.71 \times 10^{13}$ molecules cm$^{-2}$ (compared to $1.03 \times 10^{13}$ molecules cm$^{-2}$ for the corresponding *SR* fit).

Thus, in order to achieve the lowest statistical uncertainties, the largest fit wavelength range excluding the HCHO cross-section should be chosen (*SR 323 – 360 nm*).

Looking at the whole globe for the fit range *SR 323 – 360 nm* shows that the standard deviation is around $6.5 \times 10^{12}$ molecules cm$^{-2}$ for the mid-low latitude band between $70°$ S and $50°$ N (see Fig. 13a). In the arctic region, the standard deviation is gradually decreasing towards high northern latitudes, which can be due to the higher AMFs in high latitudes, i. e. the variation of the SCD is divided by a larger number. In the Antarctic region, the standard deviation also decreases for a latitude of $80°$ S. However, the distribution is not well represented by a gaussian distribution (see Fig. 13b, blue lines). There is a positive tail of elevated

BrO VCDs due to enhanced tropospheric BrO columns in polar spring events. Furthermore, the distribution is not centered around zero, i. e. the correction term is offset.

Concluding, the statistical variation of the fit *SR 323 – 360 nm* can be best estimated by the standard deviation of $6.5 \times 10^{12}$ molecules cm$^{-2}$ or less for all latitudes.

## 5.2 Summary of the findings

The investigation of the eight different fits can be summarized by the following:

- The statistical uncertainty decreases if (1) the wavelength fit range of the fit is increased and (2) the HCHO cross section is not included in the DOAS fit.

- The systematic influence of clouds and the stratospheric background can be well corrected by the correction scheme described in Sect. 4.1 for all eight fits. Thus, there is no preference for a fit range or fit setting.

- The influence of HCHO is lowest for the fit *SR 323 – 360 nm*. The BrO column shows no cross-sensitivity for moderate HCHO columns and the lowest increase for very high HCHO columns.

Thus, the fit range *SR 323 – 360 nm* is chosen for the BrO DOAS evaluation, since it shows the lowest statistical variation and is least susceptible to HCHO interference. The complete overview of the DOAS fit settings is listed in Table 1, considering the wavelength fit range of $323 - 360\,\text{nm}$ and excluding the absorption cross-section of HCHO. The chosen best fit settings 570 are recommended also for further studies on BrO in volcanic plumes.

## 5.3 Estimation of the combined statistical and systematic uncertainties

In order to give an estimate for the total error under all the various cases and locations where volcanic plumes occur, three latitude regions are distinguished: Equator is the latitude band between $20°$ S and $20°$ N, mid-latitudes stretch from the equatorial region to $50°$ N and $50°$ S, and high latitudes are further north/south. In addition, there are two special cases distinguished: 575 The polar spring in high latitudes with increased tropospheric BrO columns and biomass burning events which are typically located close to the equator. Since biomass burning can seldomly also occur in mid-latitude or boreal summer, these cases are included as well.

Finally, for each of these cases the statistical uncertainty and the systematic uncertainties are added quadratically:

$$\sigma_{combined} = \sqrt{\left(\sum \sigma_{systematic}\right)^2 + \left(\sigma_{statistical}\right)^2} \tag{10}$$

The systematic uncertainties comprise those derived in Sect. 4.4 for each latitude region and the systematic uncertainties caused by large HCHO columns, derived in Sect. 4.3 (depicted in Fig. 10). The statistical uncertainties are derived for each latitude region in Sect. 5.1 (depicted in Fig. 13).

The added systematic, statistical, and the combined overall uncertainties are shown for all cases in Table 3. Under normal conditions (that means in the absence of high HCHO columns and polar tropospheric BrO enhancements), the combined 585 uncertainty is between 6.6 and $7.0 \times 10^{12}\,\text{molecules}\,\text{cm}^{-2}$. Such conditions are representative for most of the volcanoes, e. g. the island volcanoes in the equatorial region (Galapagos, Vanuatu, Mayon) but also all mid- as well as high-latitude volcanoes such as Etna, Kilauea, and La Palma as well as Kamchatka, Iceland and the Aleutian volcanic regions.

The volcanoes with a risk of a general influence of biomass burning are located in the equator region in the vicinity of rain forest. These are most notably the African rift valley volcanoes Nyiragongo/Nyamuragira, the northern Andes volcanoes – e. g. Nevado del Ruiz – and the Central American volcanic Arc volcanoes – e. g. Masaya and Fuego. Here, an uncertainty of $7.4 \times 10^{12}\,\mathrm{molecules\,cm^{-2}}$ has to be assumed. While mid- to high-latitude volcanic regions can also be affected by high HCHO columns which would increase the combined uncertainty by roughly $0.8 \times 10^{12}\,\mathrm{molecules\,cm^{-2}}$ to 7.6 to $8.2 \times 10^{12}\,\mathrm{molecules\,cm^{-2}}$, this is expected only under very rare circumstances.

## 6 Retrieval of $BrO$ inside volcanic plumes

In order to accurately retrieve $BrO$ inside volcanic plumes, potential remaining local background gradients have to be additionally accounted for. For this information on the $SO_2$ signal is used to mask the volcanic plume signal and derive a local background correction. The retrieval steps are: The proposed $BrO$ DOAS fit *SR 323 − 360 nm* (cf. Sect. 5) is performed and the ozone latitude correction (cf. 4.2) is applied. The plume is masked based on the $SO_2$ signal, whose retrieval is explained in Sect. 6.1. This mask is used for a secondary local background correction (cf. Sect. 6.2). The results of all the different steps is discussed in detail in Sect. 6.3.

### 6.1 $SO_2$ retrieval

In this study, an $SO_2$ retrieval is employed purely to identify the volcanic plume for an accurate local background correction of $BrO$ (see next section). Thus, only the fit range of $312 − 324\,\mathrm{nm}$, already sensitive to small $SO_2$ columns, is employed. This fit range is used in the $SO_2$ verification algorithm for TROPOMI (developed by C. Hörmann, MPIC, Mainz), and refined by S. Warnach, MPIC, within the verification for the upcoming Sentinel-5 mission (van Roozendael and the Sentinel 5 Verification team) as well as on synthetic spectra within the $SO_2$ verification for the upcoming Sentinel-4 mission (Wagner and the Sentinel 4 Verification team). In all these verification exercises very good agreement was found to the operational algorithm of S-5P/TROPOMI (Theys et al., 2017).

The DOAS fit settings for the $SO_2$ retrieval includes the $SO_2$ absorption cross-section ($203\,\mathrm{K}$, Bogumil et al., 2003), and two $O_3$ absorption cross-sections ($223\,\&\,243\,\mathrm{K}$, Serdyuchenko et al., 2014). The other fit parameters (pseudo-absorbers), and the Fraunhofer reference are identical to those of the $BrO$ fit. All the fit parameter included are noted in Table 1 in the column »incl. in $SO_2$ fit«. A detailed description of the $SO_2$ retrieval can be found in Warnach (2022).

### 6.2 Local background correction

In order to correct for a potential remaining spatial pattern in the background distribution of $SO_2$ as well as $BrO$, a spatial polynomial correction is applied. This spatial polynomial was already successfully implemented as a local background correction in Hörmann (2013). Even though the $BrO$ and $SO_2$ retrieval and correction schemes presented in this study are much less affected by background gradients due to the inclusion of Pukite-Terms for $O_3$ and the aforementioned cloud-ozone correction scheme, there are still potential small spatial patterns remaining, which should be further reduced.

**Table 3.** Overview of the systematic and statistical error estimates for different latitudes and scenes. Recently active volcanic regions are listed for each scenario.

| Error class | systematic | | statistical | combined | affected volcano/ |
|---|---|---|---|---|---|
| Error origin | general | HCHO | | | volcanic region |
| molecules $cm^{-2}$ | $\times 10^{12}$ | $\times 10^{12}$ | $\times 10^{12}$ | $\times 10^{12}$ | |
| Equator: | | | | | |
| normal | 1.0 | – | 6.5 | 6.6 | Vanuatu islands |
| | | | | | New Zealand |
| | | | | | Galapagos islands, Ecuador |
| | | | | | Piton de la Fournaise, France |
| | | | | | Anak Krakatau, Indonesia |
| biomass burning | 1.0 | 2.5 | 6.5 | 7.4 | DR Congo |
| | | | | | Central American Volcanic Arc |
| | | | | | Central American Volcanic Arc |
| | | | | | South East Asian volcanoes |
| Mid-latitudes: | | | | | |
| normal | 1.5 | – | 6.5 | 6.7 | Italian volcanoes |
| | | | | | Canaries, Spain |
| | | | | | Azores, Portugal |
| | | | | | Hawaii, USA |
| | | | | | Mainland USA |
| | | | | | Chile and Peru |
| | | | | | Japan |
| biomass burning | 1.5 | 2.5 | 6.5 | 7.6 | – |
| High-latitudes: | | | | | |
| normal | 2.5 | – | 6.5 | 7.0 | Aleutian islands[†], USA |
| | | | | | Kamtchatka, Russia |
| | | | | | Iceland volcanoes |
| polar spring | 4.0 | – | 6.5 | 7.6 | Mt. Erebus, Antarctica |
| biomass burning | 2.5 | 2.5 | 6.5 | 8.2 | – |

The retrieved VCD $V_m$ can be decomposed into the background VCD $V_b$ and the volcanic VCD $V_v$:

$\quad V_m = V_b + V_v$ (11)

The spatial polynomial correction scheme makes use of the fact that volcanic plumes can usually be considered as a localized phenomenon of several $100\,\mathrm{km}$ extent within a large scale background distribution pattern on the scale of $1000\,\mathrm{km}$ for both $BrO$ and $SO_2$. Furthermore, both volcanic and background distributions are independent from each other and it can be assumed that the background is smooth with respect to the scale of the volcanic plume and background information gathered spatially

around the plume can be interpolated over the plume region.

The background VCD $V_b$ for each pixel k is modeled by two-dimensional polynomial of degree three:

$$[V_b]_k = \sum_{i,j=0}^{3} c_{i,j} \cdot x_k^i \cdot y_k^j \tag{12}$$

where x is the pixel latitude and y the longitude corrected by a latitudinal dependent squeeze:

$$[V_b]_k = \sum_{i,j=0}^{3} c_{i,j} \cdot [\mathrm{lat}]_k^i \cdot [\mathrm{lon} \cdot \cos(\mathrm{lat})]_k^j \tag{13}$$

In order to calculate the polynomial, first the volcanic plume is masked. For this a $SO_2$ VCD threshold is chosen, above which a pixel is masked as influenced by the volcanic plume. This is chosen in this study as four times the standard deviation of the $SO_2$ background distribution (for TROPOMI the standard deviation is typically in the range of $5 \times 10^{15}\,\mathrm{molecules\,cm^{-2}}$, i. e. the threshold is in the range of $2 \times 10^{16}\,\mathrm{molecules\,cm^{-2}}$, compare Warnach, 2022). The polynomial coefficients $c_{i,j}$ are then determined for this subset of data for both $BrO$ and $SO_2$ independently and applied to each pixel of the complete data-set,

leading to the background VCD. The resulting (volcanic) VCD is obtained by subtracting the background VCD $V_b$ from the retrieved VCD $V_m$.

## 6.3 Results

The complete sequence of processing steps of the retrieval of $BrO$ for a volcanic plume is shown for the example of the plume of Mt. Etna, Italy, on 25 December 2018. The $BrO$ and $SO_2$ VCD maps derived from the respective DOAS retrievals

as well as the calculated $O_3$ VCD and the FRESCO cloud fraction are depicted in Fig. 14a-d respectively. The volcanic plume is clearly visible in the $SO_2$ map. This is also the case for $BrO$, but there are also large gradients visible in the scale of $5 \times 10^{13}\,\mathrm{molecules\,cm^{-2}}$, which is in the order of the volcanic $BrO$ signal itself. The processing steps of the $BrO$ correction scheme is shown in Fig. 15. The uncorrected $BrO$ VCD (shown in panel a) is first corrected using the cloud-ozone correction term (whose correction $BrO$ VCD is shown in panel b), which is derived using the $BrO$ VCDs from the $20°$ latitude band

around encompassing the volcano (in this case $25° - 45°\,\mathrm{N}$). This correction term removes most of the large-scale biases in

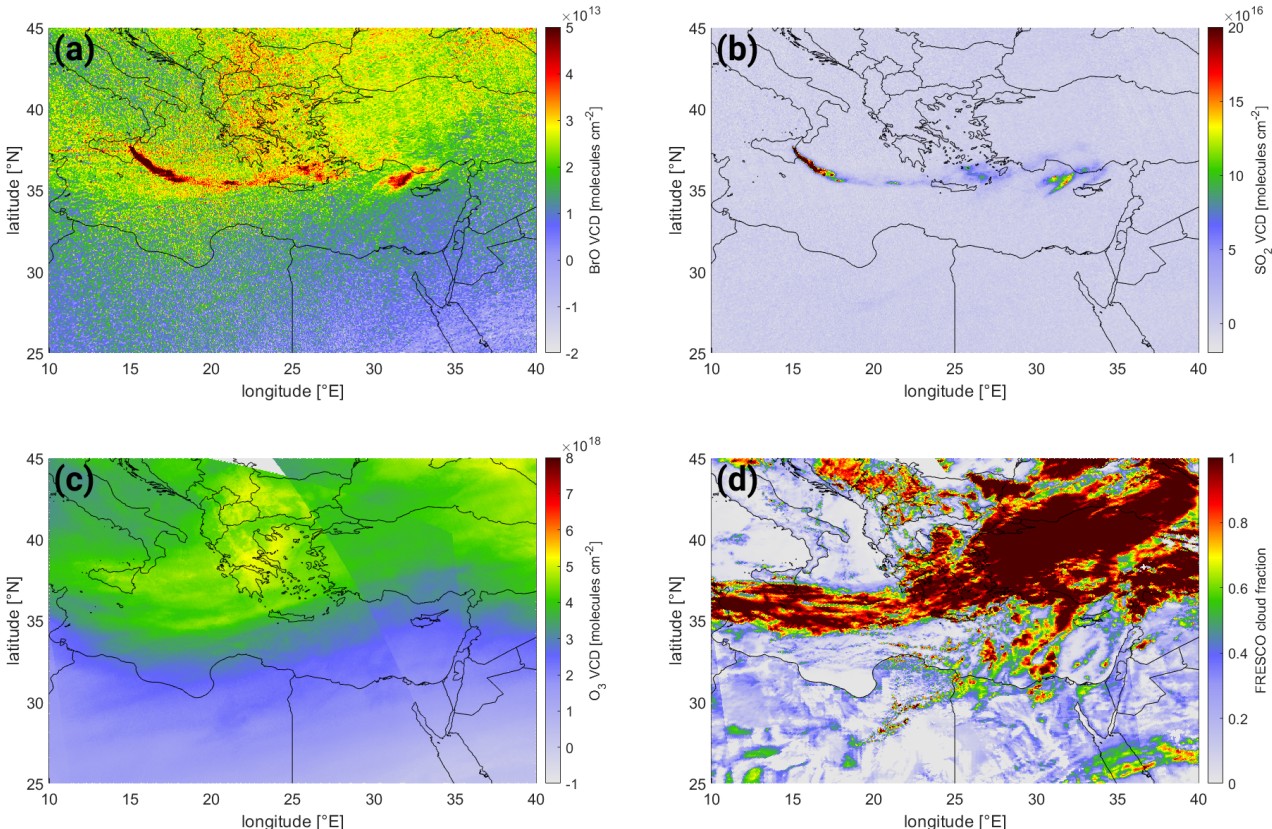

**Figure 14.** Plume of Mt. Etna on 25 December 2018. **(a)** The $\mathrm{BrO}$ VCD map and **(b)** the $\mathrm{SO_2}$ VCD map without correction. **(c)** The $\mathrm{O_3}$ VCD derived from the $\mathrm{BrO}$ fit and **(d)** the cloud fraction (from FRESCO), both used in the »ozone latitude« correction scheme.

the $\mathrm{BrO}$ map, but also fine structures originating from cloud shielding, resulting in the ozone latitude corrected $\mathrm{BrO}$ VCD map (shown in panel c). Lastly, the volcanic plume is masked based on the $\mathrm{SO_2}$ signal (using a three-$\sigma$ threshold, panel d) and the the remaining pixel in the region used for a local background correction (the correction term is shown in panel f), resulting in the final $\mathrm{BrO}$ VCD (shown in panel e). There, the systematic background features are well reduced.

In the following, several examples of different $\mathrm{BrO}$ VCD strength and at different latitudes are displayed and discussed in order to show the efficacy of the retrieval in different circumstances.

### 6.3.1   The plume of Sheveluch, Kamchatka, Russia on 18 April 2019

An example of a small scale eruption at high latitudes is the plume of Sheveluch, Kamtchatka on 18 April 2019. While only low $\mathrm{SO_2}$ VCDs are observed – not exceeding $8 \times 10^{16}\,\mathrm{molecules\,cm^{-2}}$ (see Fig. 16a) – there is still a clearly enhanced $\mathrm{BrO}$ 655    VCDs visible coinciding with the $\mathrm{SO_2}$ pattern (see Fig. 16b) in the order of several $\times 10^{13}\,\mathrm{molecules\,cm^{-2}}$. However, there is

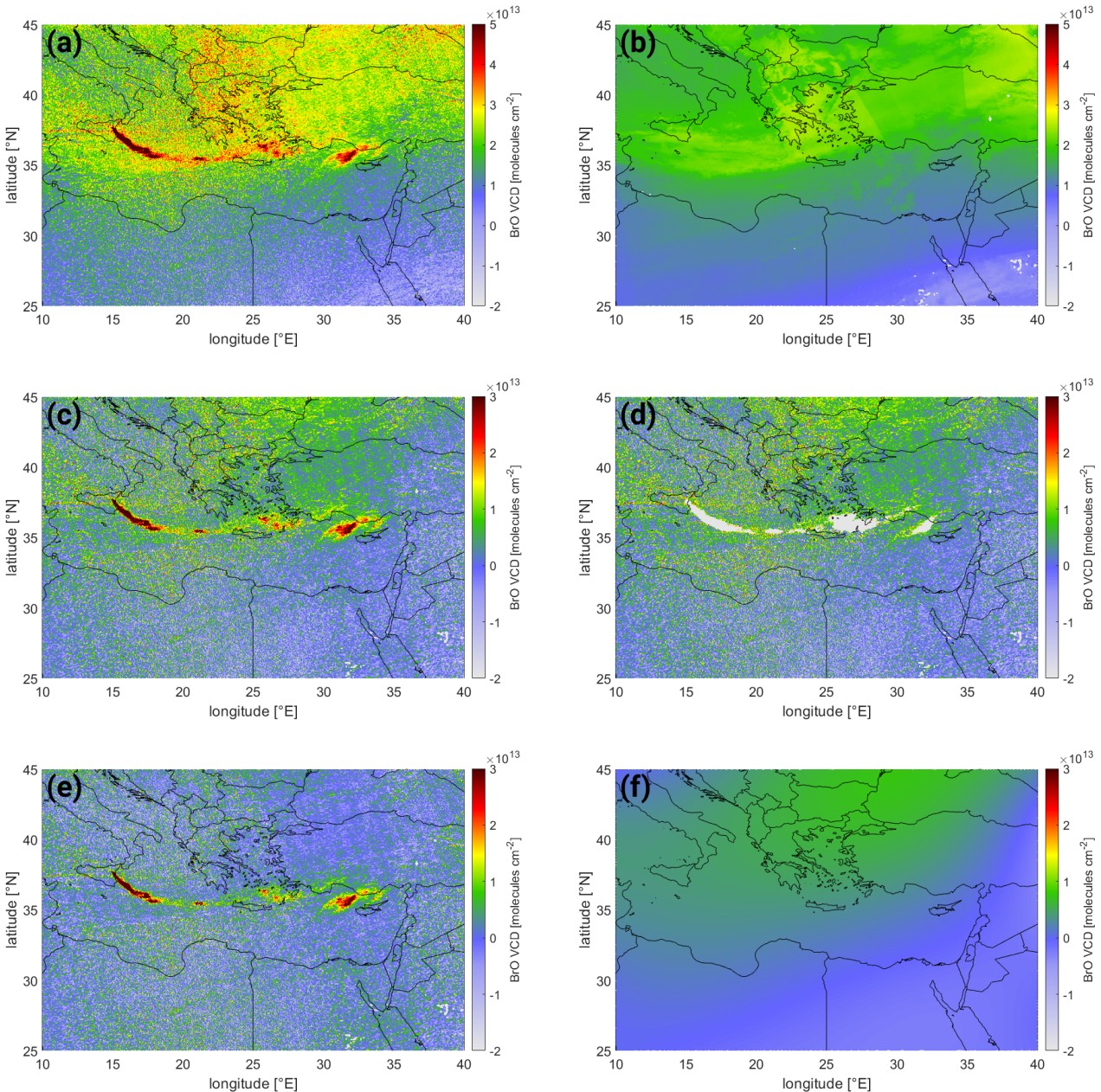

**Figure 15.** Overview of the processing of the BrO VCD for a volcanic event (Mt. Etna, 25 December 2018): **(a)** The uncorrected data is first corrected using **(b)** the ozone latitude correction term to yield **(c)** the ozone latitude corrected BrO VCD. For the calculation of the spatial polynomial correction **(d)** the masked plume is used and a third order spatial 2D polynomial term is fitted, yielding **(f)** the local background correction BrO map. **(e)** Subtraction from **(c)** yields the final corrected BrO VCD map.

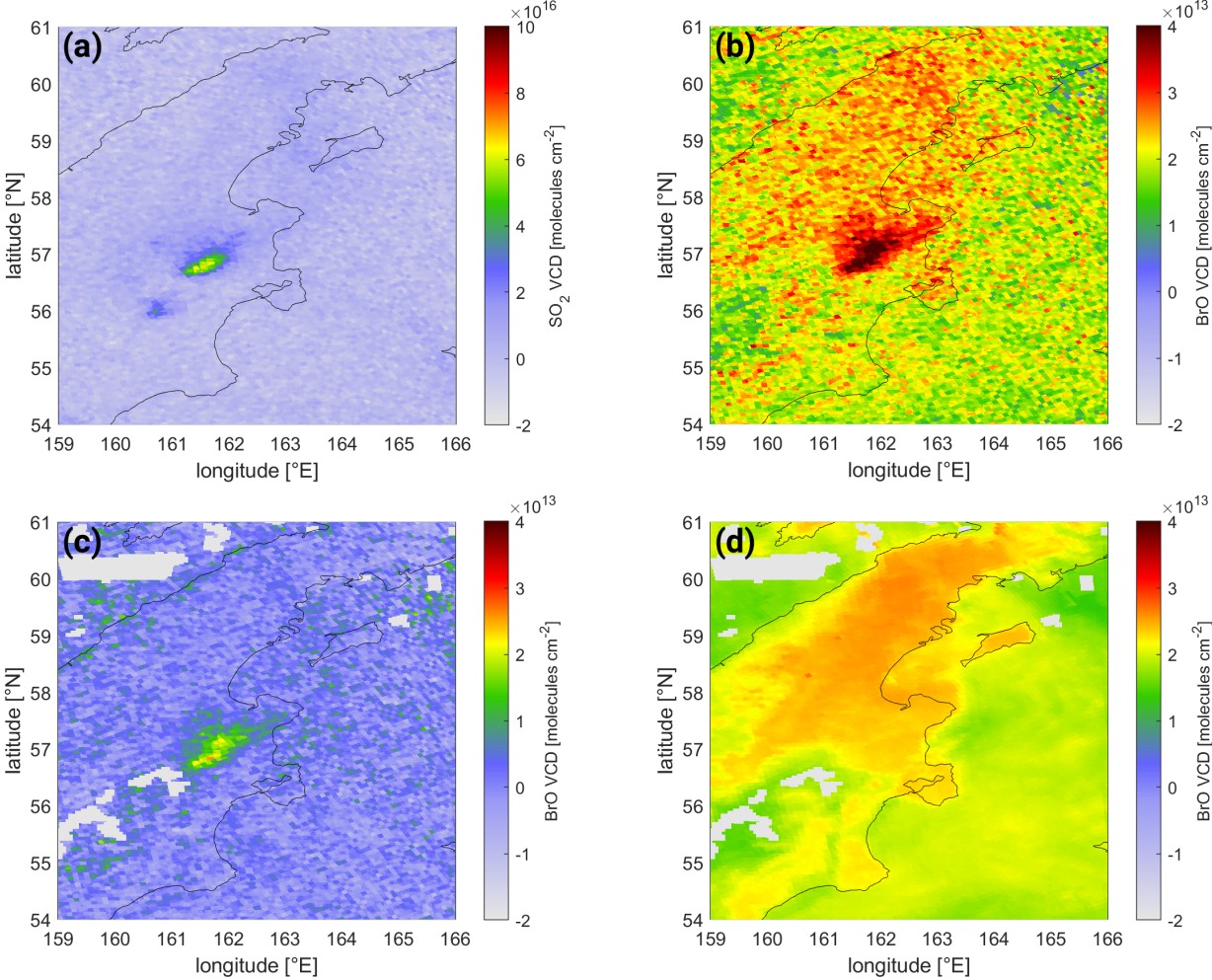

**Figure 16.** Eruption of Sheveluch, Kamchatka, Russia, on 18 April 2019. **(a)** the $SO_2$ VCD map, **(b)** the original $BrO$ VCD map, **(c)** the corrected $BrO$ VCD map, and **(d)** the $BrO$ correction term consisting of the »ozone latitude« $BrO$ correction and the local background correction.

also elevated background $BrO$ of up to $3 \times 10^{13}\,\mathrm{molecules\,cm^{-2}}$ visible north of the plume over the island. These background structures are clearly identified by the correction term (see Fig. 16d) and the resulting $BrO$ VCD map (see Fig. 16c) shows a smooth background of roughly zero.

It is noteworthy that this is an example where the correction term actually shows a large gradient at the location of the plume (at $57°$ N and $161° - 162°$ E) where the correction term varies by more than $5 \times 10^{12}\,\mathrm{molecules\,cm^{-2}}$. This gradient is caused by a cloud edge in this area (see Fig. C1c,d).

### 6.3.2 The plume of Ambrym volcano, Vanuatu on 30 September 2018

Another minor eruption where a BrO signal is detected is the eruption of Ambrym volcano on 30 September 2018 (see Fig. 17). While the $SO_2$ signal can be detected over several $100\,km$, maximum $SO_2$ columns do not exceed $1.5 \times 10^{17}\,molecules\,cm^{-2}$ (see Fig. 17a). Nevertheless, BrO VCDs exceed $3 \times 10^{13}\,molecules\,cm^{-2}$ indicating that the volcanic emissions are relatively BrO-rich (see Fig. 17b). In contrast to the high-latitude Sheveluch eruption (see 6.3.1), the BrO background in this case is much weaker (see Fig. 17b), since the plume is located close to the equator where the reference spectrum is taken. Furthermore, there are only low clouds in the area (see Fig. C2d), hence cloud influence is low. Nevertheless, the correction term applies minor corrections for cloudy regions, e. g. for the high cloud at $14°$ S $168°$ E Fig. C2d, where the correction factor accounts for the negative BrO VCD (see Fig. 17d), and slightly homogenizes the background (see Fig. 17c).

### 6.3.3 The eruption of Mt. Etna, Italy on 29 January 2019

An example for a medium eruption, with a very localized volcanic plume is the Mt. Etna eruption on 29 January 2019 (see Fig. 18). This eruption is part of a phase of eruptive activity which started with weak degassing in December 2018 culminating in an explosive VEI-2 eruption on 24 December 2018 (Calvari et al., 2020) accompanied by lava flows (GVP, 2019). Activity subsided to strombolian activity throughout January and February 2019 (GVP, 2019). The eruptive event on 29 January 2019 falls within a week of increased activity during the last week of January (GVP, 2019). The event is a medium eruption, with maximum $SO_2$ VCD of $5 \times 10^{17}\,molecules\,cm^{-2}$, but with a very localized plume (see Fig. 18a). $SO_2$ VCDs exceeding $1 \times 10^{17}\,molecules\,cm^{-2}$ only occur within the first $50\,km$ downwind from the volcano. A clear BrO signal of up to $8 \times 10^{13}\,molecules\,cm^{-2}$ is also only observed in this area around the volcano (see Fig. 18b, exceeding the upper boundary of the colormap scale). Further downwind only a very faint positive BrO signal can be seen. The background signal is in the order of $2 \times 10^{13}\,molecules\,cm^{-2}$ is well matched by the correction term (see Fig. 18d), yielding the corrected BrO VCD map (see Fig. 18c), where both the strong signal at the volcano and the weaker outflow is visible.

### 7 Conclusions

We present a new retrieval scheme designed specifically for the accurate detection of BrO inside minor volcanic plumes from satellite spectra. Expanding the DOAS fit wavelength range to lower wavelength of $323\,nm$ and thus a fit range of $323-360\,nm$ reduces the statistical uncertainty, without increasing any systematic influences. This is made possible by the introduction of so called »Pukite« terms (Pukīte et al., 2010), which account for second order effects of the strong absorption of $O_3$ at lower wavelengths. In addition, this wavelength range was shown to be least susceptible for a systematic influence caused by interference due to HCHO. While no systematic bias is found for HCHO columns $< 2 \times 10^{16}\,molecules\,cm^{-2}$, HCHO columns between 2 and $4 \times 10^{16}\,molecules\,cm^{-2}$ could introduce a positive BrO VCD bias of roughly $2.5 \times 10^{12}\,molecules\,cm^{-2}$.

Secondly, we propose a sophisticated, empirical correction scheme for the non-volcanic BrO background, based on cloud fraction, cloud altitude, as well as on the $O_3$ VCD, all correlating with spatially distinct patterns in the BrO VCD map. The

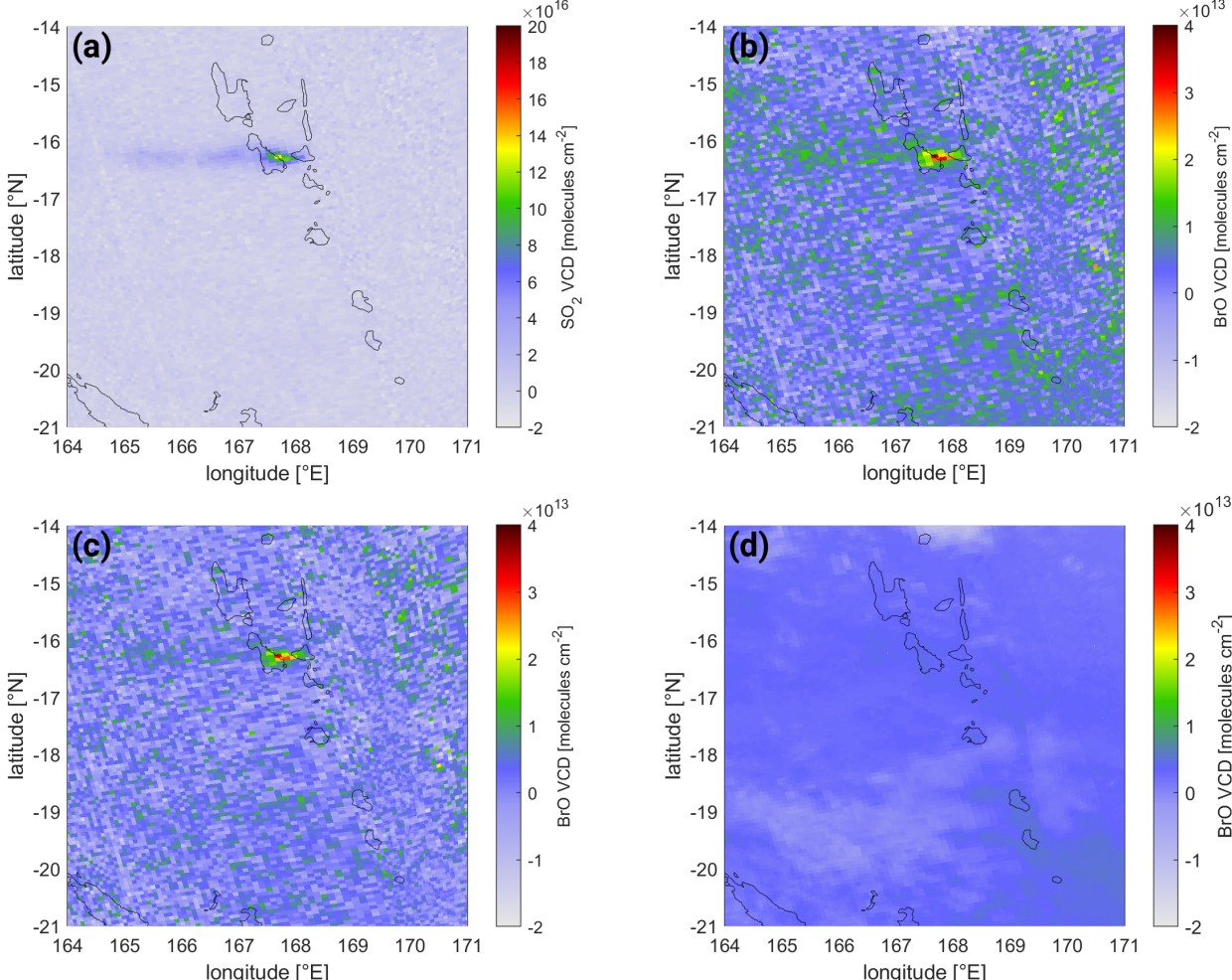

**Figure 17.** Eruption of Ambrym volcano, Vanuatu, on 30 September 2018. **(a)** the $SO_2$ VCD map, **(b)** the original $BrO$ VCD map, **(c)** the corrected $BrO$ VCD map, and **(d)** the $BrO$ correction term consisting of the »ozone latitude« $BrO$ correction and the local background correction.

origin of the cloud effect is not clear. Possible effects can be partially shielding of the tropospheric $BrO$ column, but also non-linear spectroscopic effects. $O_3$ is indicative for a change in the $BrO$ stratospheric column. A combined empirical multi-dimensional polynomial including cloud fraction, cloud height and the measured $O_3$ VCD applied to the latitude band of the volcanic plume location proves to best account for these structures. This correction scheme reduces the systematic influence at the equator by one order of magnitude from $1 \times 10^{13}\,\mathrm{molecules\,cm^{-2}}$ to $1 \times 10^{12}\,\mathrm{molecules\,cm^{-2}}$.

The statistical uncertainty of the retrieved $BrO$ VCD is found to be roughly $6.5 \times 10^{12}\,\mathrm{molecules\,cm^{-2}}$, thus at least a factor of two larger than the systematic effects. Compared to other recent $BrO$ DOAS fitting schemes (Sihler et al., 2012; Hörmann

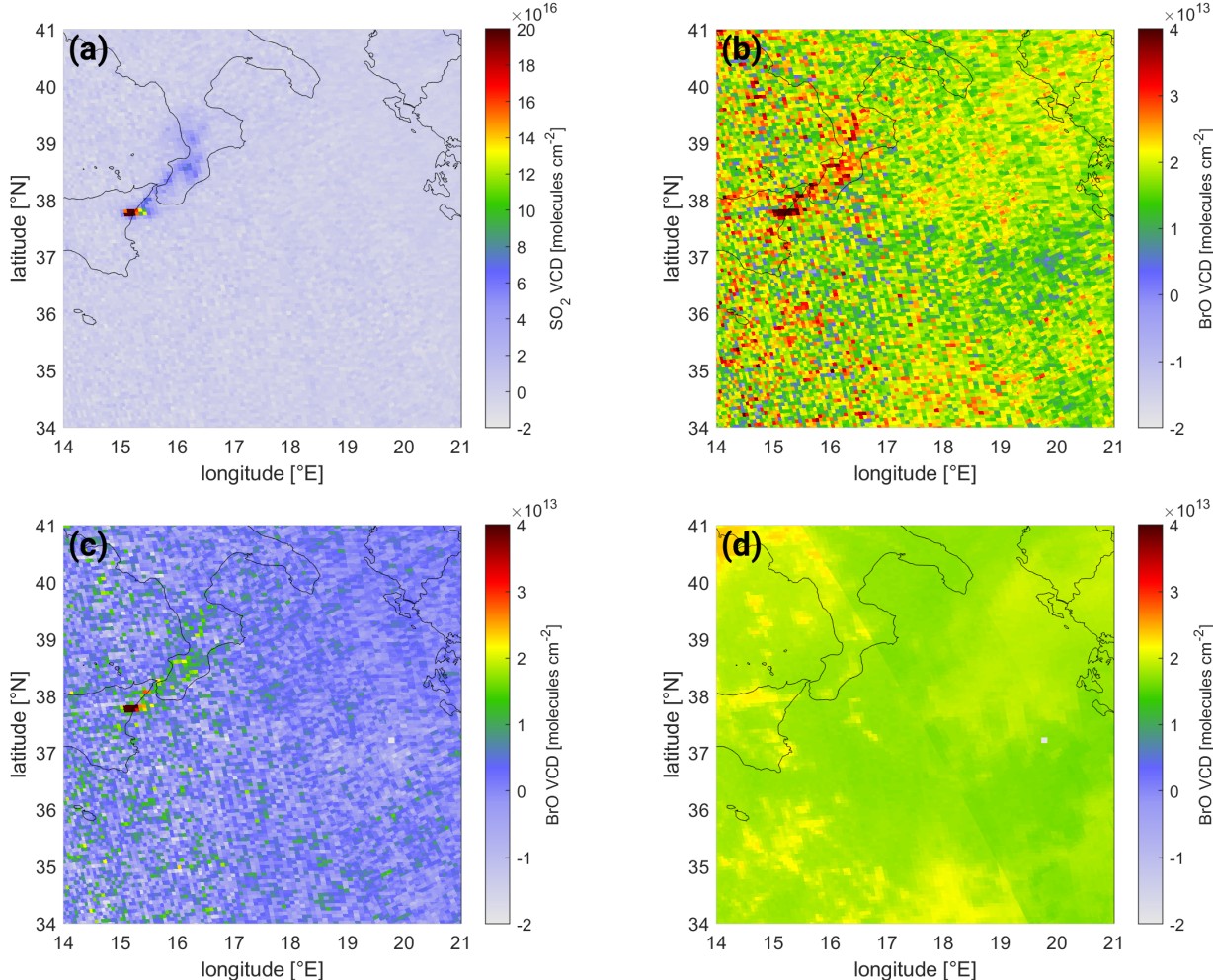

**Figure 18.** Eruption of Mount Etna, Italy, on 29 January 2019. **(a)** the $SO_2$ VCD map, **(b)** the original BrO VCD map, **(c)** the corrected BrO VCD map, and **(d)** the BrO correction term consisting of the »ozone latitude« BrO correction and the local background correction.

700 et al., 2013) the statistical noise is reduced by a factor of up to 1.8, mainly due to the use of the larger wavelength fit range of $323 - 360\,\text{nm}$. It is noteworthy, that averaging over longer periods of time could further reduce the statistical variation.

In several example volcanic plumes we show, that the developed background correction scheme successfully removes BrO background gradients. In combination with the precise BrO DOAS retrieval, this leads to the more accurate detection of BrO than for previous retrievals even in volcanic plumes during minor eruptive activity.

705 The algorithm developed here will allow much more detailed investigation of volcanic BrO and the volcanic $BrO/SO_2$ ratio on a global scale.

Even though our BrO retrieval algorithm is designed for TROPOMI observations, the improvements proposed here are in principle applicable to the BrO observations inside volcanic plumes from any hyperspectral satellite observations. However, satellite specific adjustments might be required.

## Appendix A: Influence of volcanic plumes onto earthshine reference spectrum

In order to quantify the influence of the presence of volcanic plumes within the equatorial reference spectra region onto the BrO VCDs, we selected two example days: 2 October 2021, where only several, small plumes are present (cf. Fig. A1a, red areas), representative of normal conditions, and 30 July 2018, where a very large plume stretched over a large portion of the equatorial region (cf. Fig. A2a, red areas), representative of exceptionally strong volcanic activity within the equatorial region. For both days we identified areas affected by a volcanic plume based on the $SO_2$ signal (as done in Warnach, 2022, cf. Sect. 5.2), marked as red areas. Lastly, we calculated the mean BrO VCD within the equatorial region independently for each across-track detector both including and excluding the affected volcanic areas. The difference between both should be equivalent to the contamination of the earthshine reference spectrum.

For 2 October 2020, there is no difference in the BrO visible and for $SO_2$ only a small difference for a few detectors (cf. Fig. A1c,d). Our interpretation is that typical signals are too weak to exceed the noise of the BrO retrieval (as the signal-to-noise is two orders of magnitudes larger than for $SO_2$) and that typically only a very small fraction of pixels are affected by volcanic plumes. On 30 July 2018, which is representative for an exceptionally strong volcanic plume, there is only a contamination of the $SO_2$ SCD of $6 \times 10^{15}$ molecules cm$^{-2}$ (cf. Fig. A2a), which is more than one order of magnitude lower than typical volcanic $SO_2$ SCDs (which are on the order of $1 \times 10^{17}$ molecules cm$^{-2}$). For BrO, the difference is even smaller and several $1 \times 10^{11}$ molecules cm$^{-2}$ which is also at least 1.5 orders of magnitude lower than typical volcanic BrO SCDs (which are on the order of $1 \times 10^{13}$ molecules cm$^{-2}$).

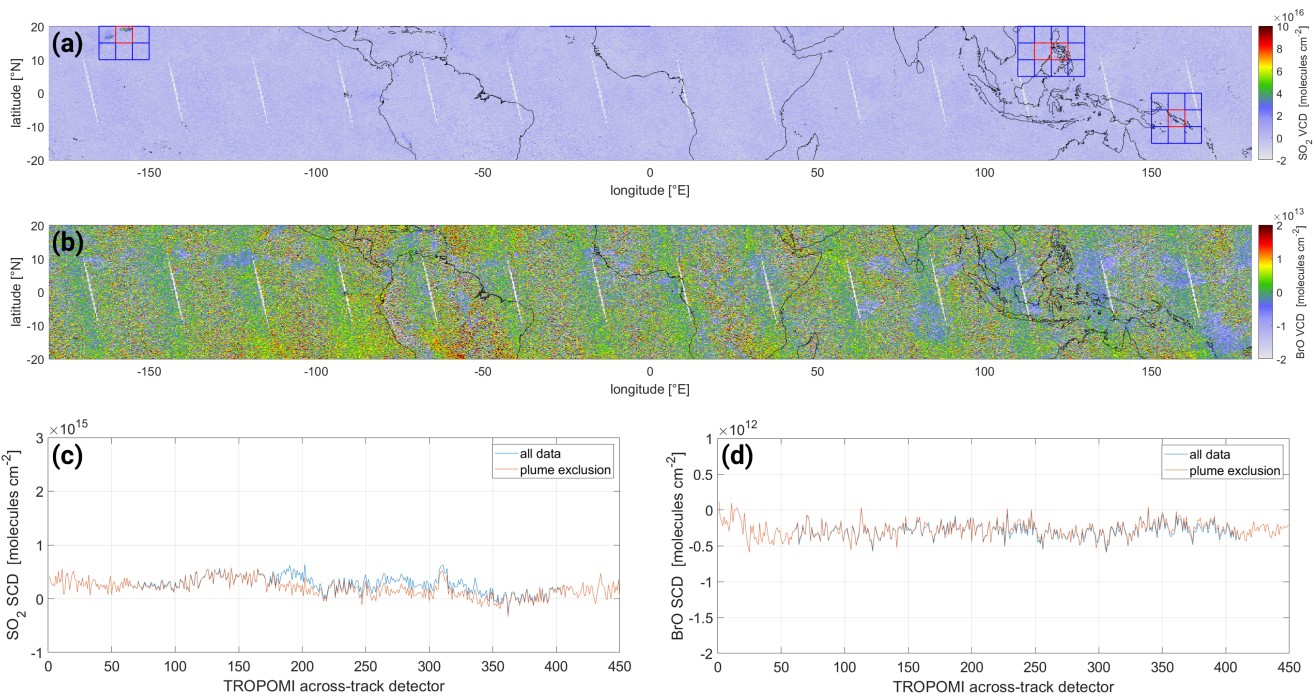

**Figure A1. (a)** Map of the $SO_2$ VCD for the equatorial region on 2 October 2021. The areas of enhanced volcanic signals are marked in red squares. **(b)** Map of the $BrO$ VCD for the equatorial region on 2 October 2021. **(c)** Mean $SO_2$ SCD for each across-track detector considering all pixels (blue) and excluding pixels with volcanic gas columns (red). **(d)** Same plot for the across-track dependent mean BrO SCD.

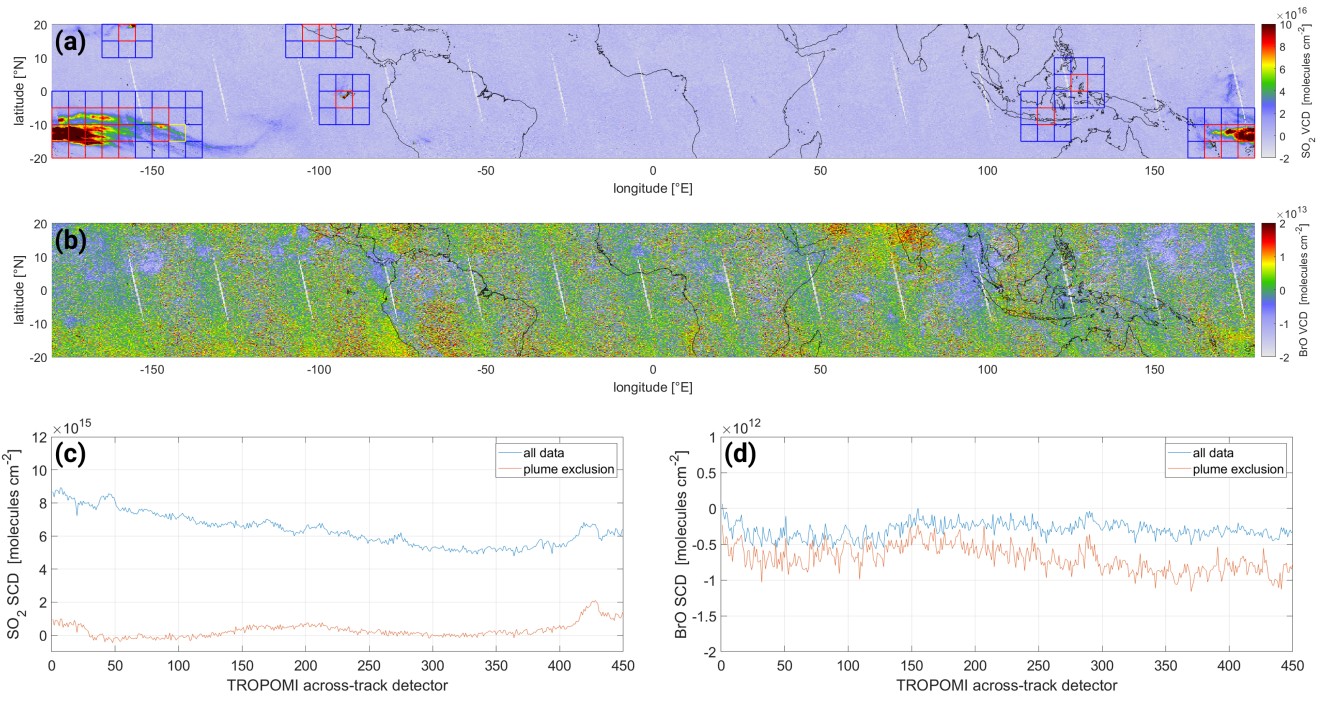

**Figure A2. (a)** Map of the $SO_2$ VCD for the equatorial region on 30 July 2018. The areas of enhanced volcanic signals are marked in red squares. **(b)** Map of the $BrO$ VCD for the equatorial region on 30 July 2018. **(c)** Mean $SO_2$ SCD for each across-track detector considering all pixels (blue) and excluding pixels with volcanic gas columns (red). **(d)** Same plot for the across-track dependent mean BrO SCD.

## Appendix B: Detailed data for systematic influences

### B1 High-latitude air-masses influence on $BrO$ and $O_3$

In order to interpret the coinciding gradients of $O_3$ and $BrO$ in Fig. 5 with respect to meteorology, we look at the location of
the jet stream estimated as band of strong wind-speed at $9\,km$ altitude and the tropopause height estimated as the height where
the potential vorticity is 2 potential vorticity units (PVU). The corresponding data are taken from ECMWF ERA-5 model data
at 18:00 h UTC for the respective region and is depicted in Fig. B1.

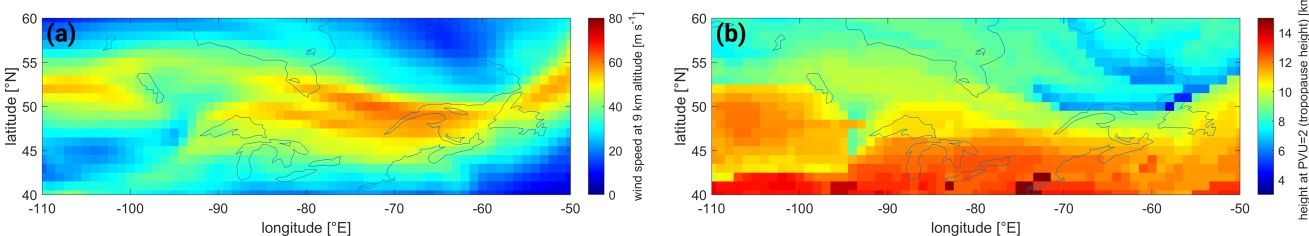

**Figure B1.** Meteorological conditions for the high-latitude case (cf. Fig. 5) on 1 October 2018, taken from ECMWF ERA-5 data at 18:00 h
UTC: **(a)** the wind speed at 9km indicating the presence of the jet stream and **(b)** the tropopause height indicated by the height where the
potential vorticity is equal to 2 potential vorticity units.

### B2 Cloud and $O_3$ influences

$BrO$ VCD as a function of CF and CH at the equatorial region on 1 October 2018 for all eight fit settings (see Fig. B2).
The relation is stronger for larger fit wavelength ranges (left side) and seems also slightly increased for fits including HCHO
(bottom row). The response is weakest for the fits with wavelength range $332-360\,nm$ both for *SR* and *HCHO* fits.

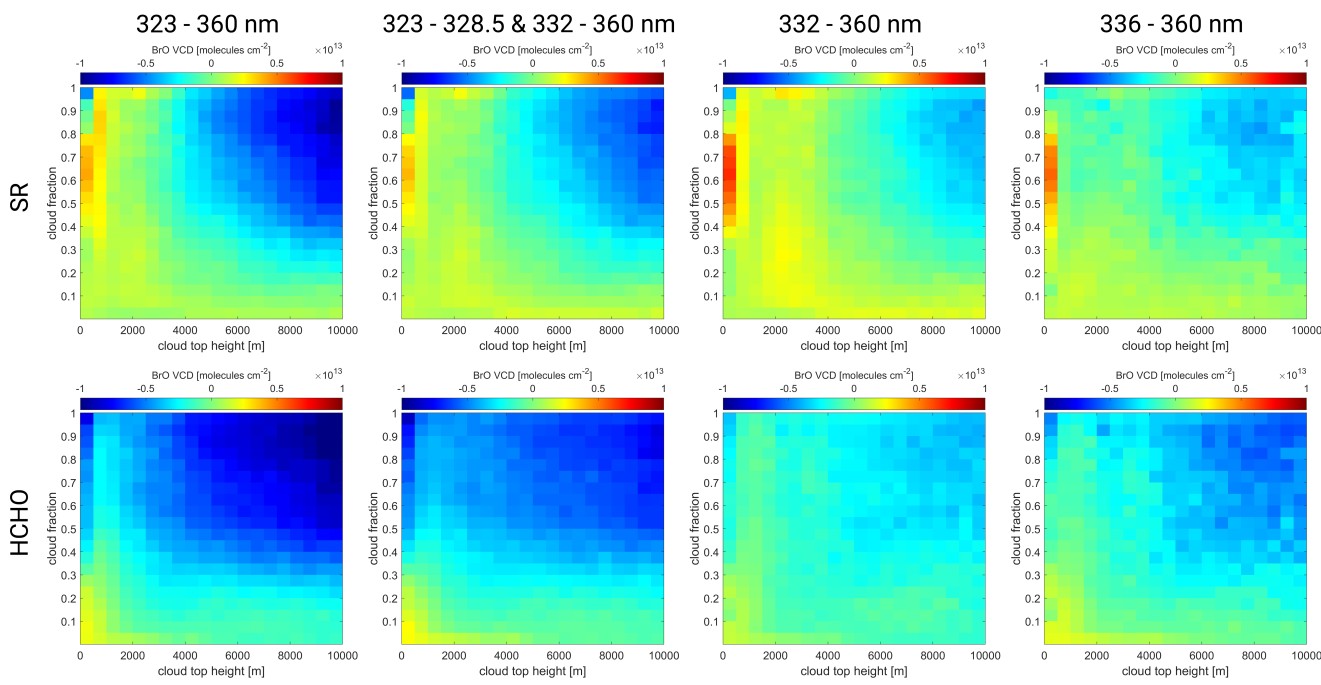

**Figure B2.** Mean BrO VCD dependency on the cloud height – cloud fraction bins for the equatorial region [$\pm20°$ N, $\pm180°$ E] for the eight different BrO fits. Top row: Fits not including HCHO absorption cross section (*SR*), fit ranges: *323-360 nm*, *323-328.5 & 332-360 nm*, *332-360 nm*, *336-360 nm* (f. l. t. r). Bottom row: Fits including the HCHO absorption cross section (*HCHO*), for each fit range from the top row respectively.

## B3 Cloud and $O_3$ correction scheme

Fig. B3 shows the correction terms for the three different ozone cloud correction schemes. The corresponding corrected BrO VCD maps are shown in Fig. 7.

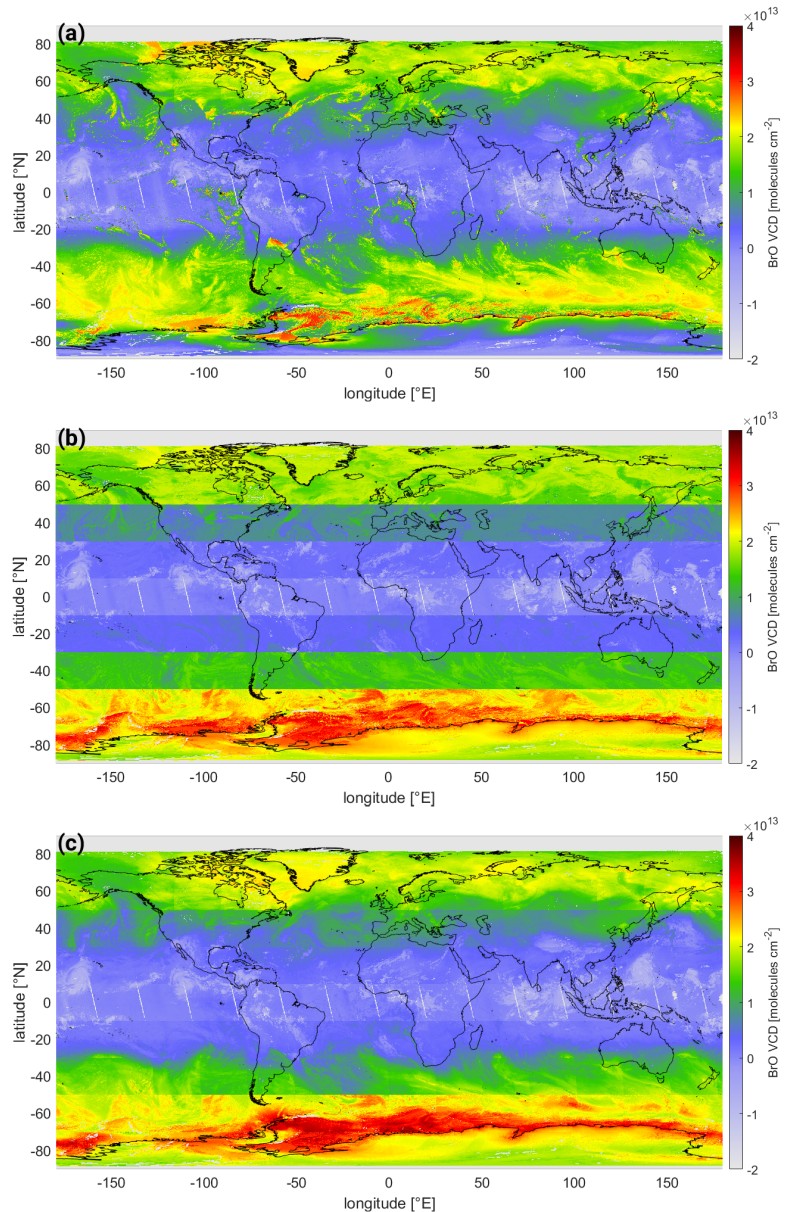

**Figure B3.** Global map of the correction term of the three correction schemes for the *SR 323 – 360 nm* fit: **(a)** »ozone« **(b)** »latitude«, and **(c)** »ozone latitude« correction schemes. The $\mathrm{BrO}$ VCD map prior to correction is plotted in Fig. 3, and the corresponding corrected maps for all three schemes is plotted in Fig. 7.

 **B4  Influence of formaldehyde**

Fig. B4 shows the mean BrO SCD seperated by the HCHO-CF bins for each of the eight fit settings of the African biomass burning region.

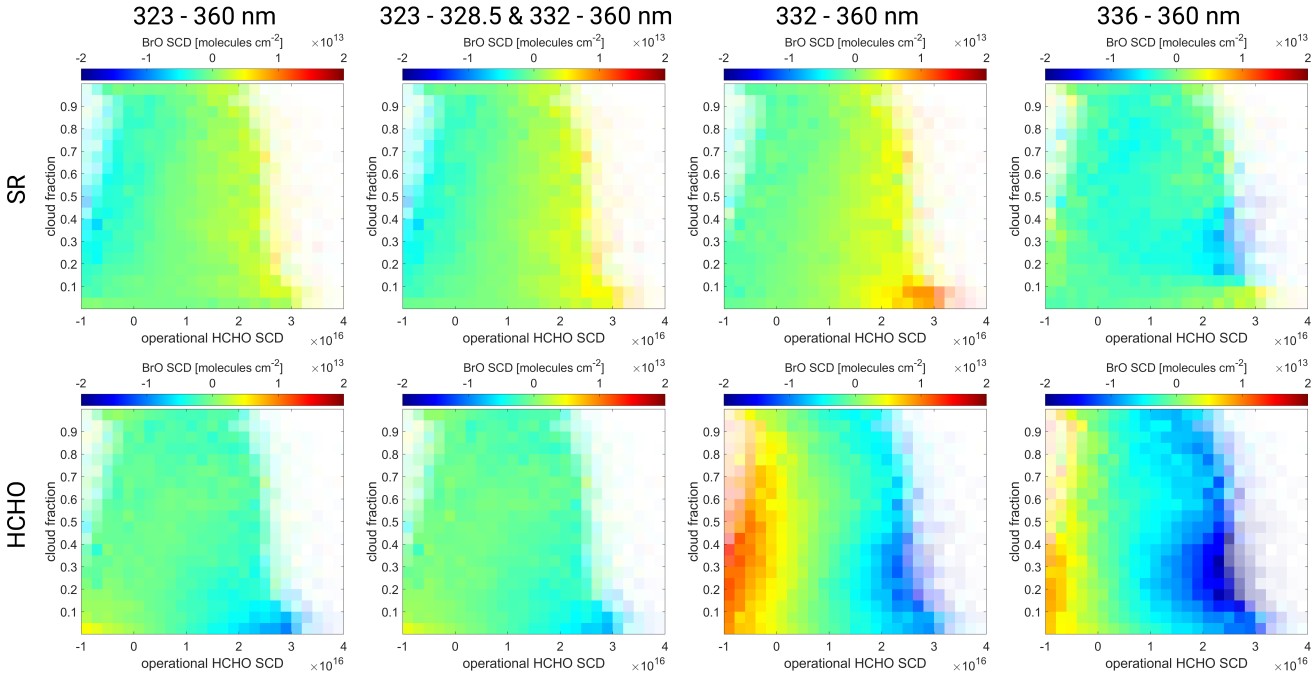

**Figure B4.** Mean BrO SCD as a function cloud fraction and HCHO SCD bins for the African biomass region on 1 October 2018 for the four different fit ranges (from left to right): *323 – 360 nm*, *323 – 328.5 & 332 – 360 nm*, *332 – 360 nm*, and *336 – 360 nm*. *Upper row*: fits excluding HCHO (*SR*); *lower row*: fits including HCHO (*HCHO*).

In this section the auxiliary data used for the cloud ozone correction scheme are shown for every example eruption detailed

in Sect. 6.3. All plots include the $SO_2$ map to indicate the plume dispersion pattern and the $O_3$ VCD as well as the FRESCO cloud height and cloud fraction, all three used as input for the cloud ozone correction scheme.

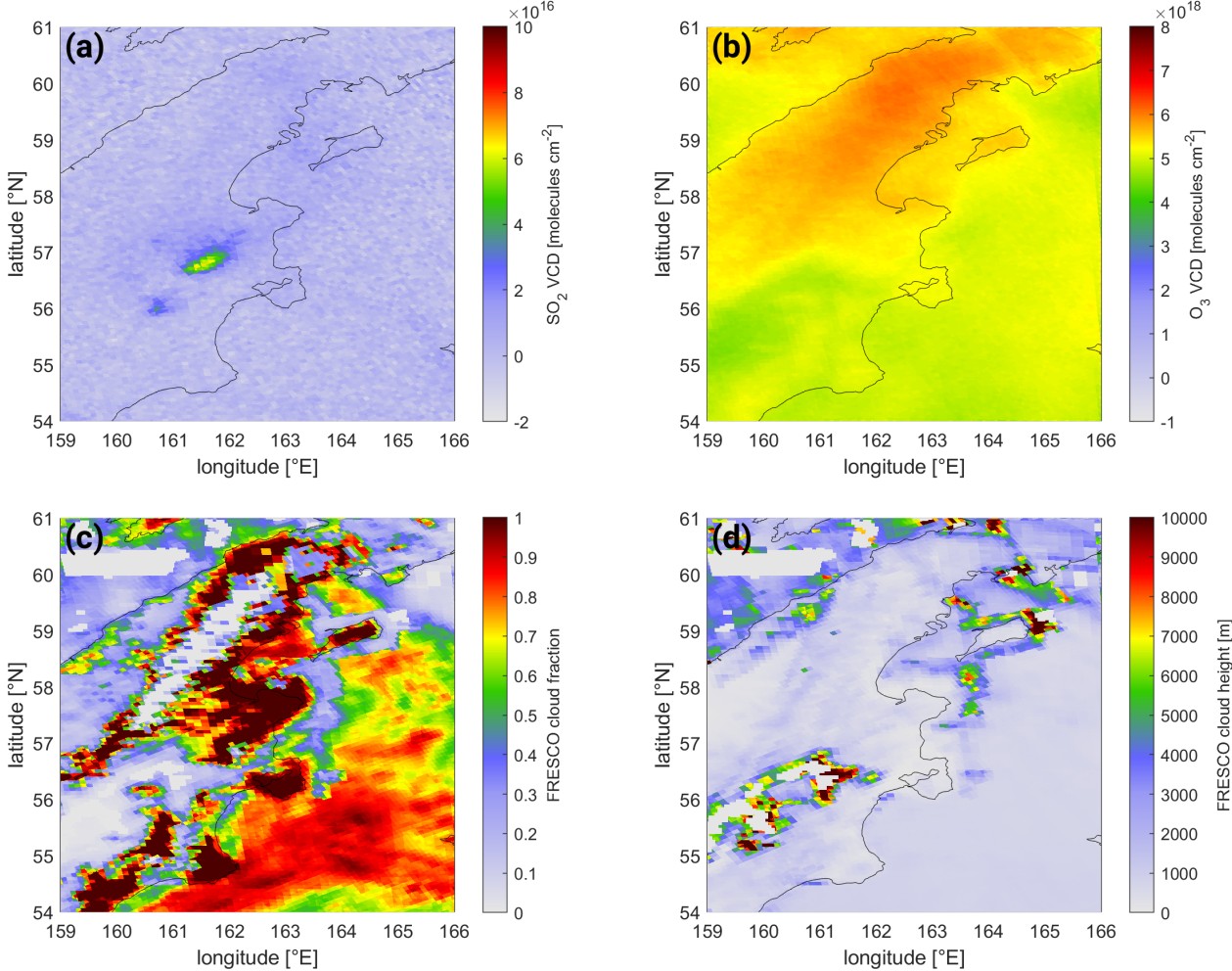

**Figure C1.** Auxiliary data used for correction scheme for the eruption of Sheveluch, Kamchatka, Russia, on 18 April 2019. Maps of **(a)** the $SO_2$ VCD, **(b)** the $O_3$ VCD, **(c)** the FRESCO cloud fraction, and **(d)** the FRESCO cloud height. The corresponding uncorrected and corrected BrO VCD map and the BrO background correction map can be found in Fig. 16.

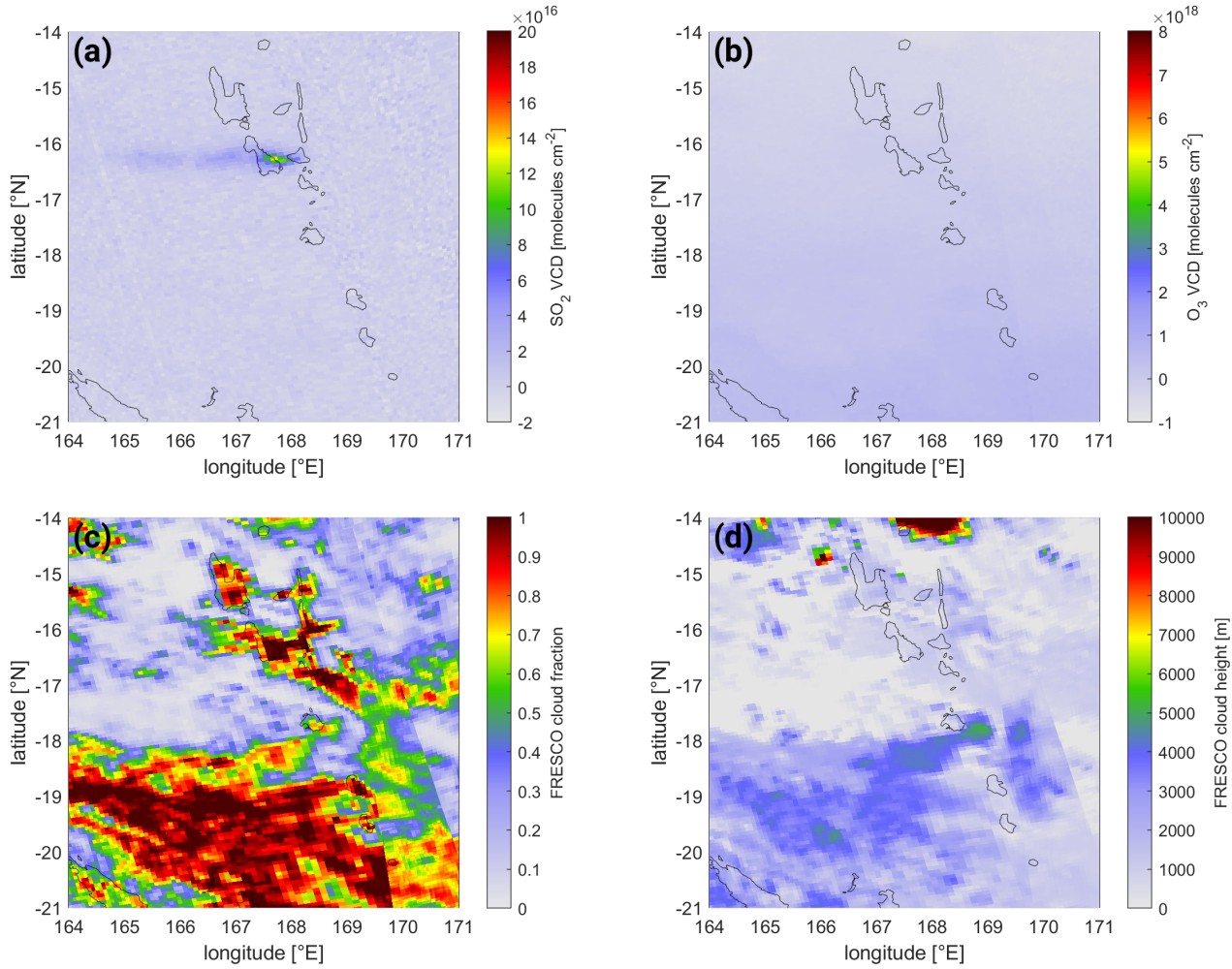

**Figure C2.** Auxiliary data used for correction scheme for the eruption of Ambrym, Vanuatu, on 30 September 2018. Maps of **(a)** the $SO_2$ VCD, **(b)** the $O_3$ VCD, **(c)** the FRESCO cloud fraction, and **(d)** the FRESCO cloud height.The corresponding uncorrected and corrected $BrO$ VCD map and the $BrO$ background correction map can be found in Fig. 17.

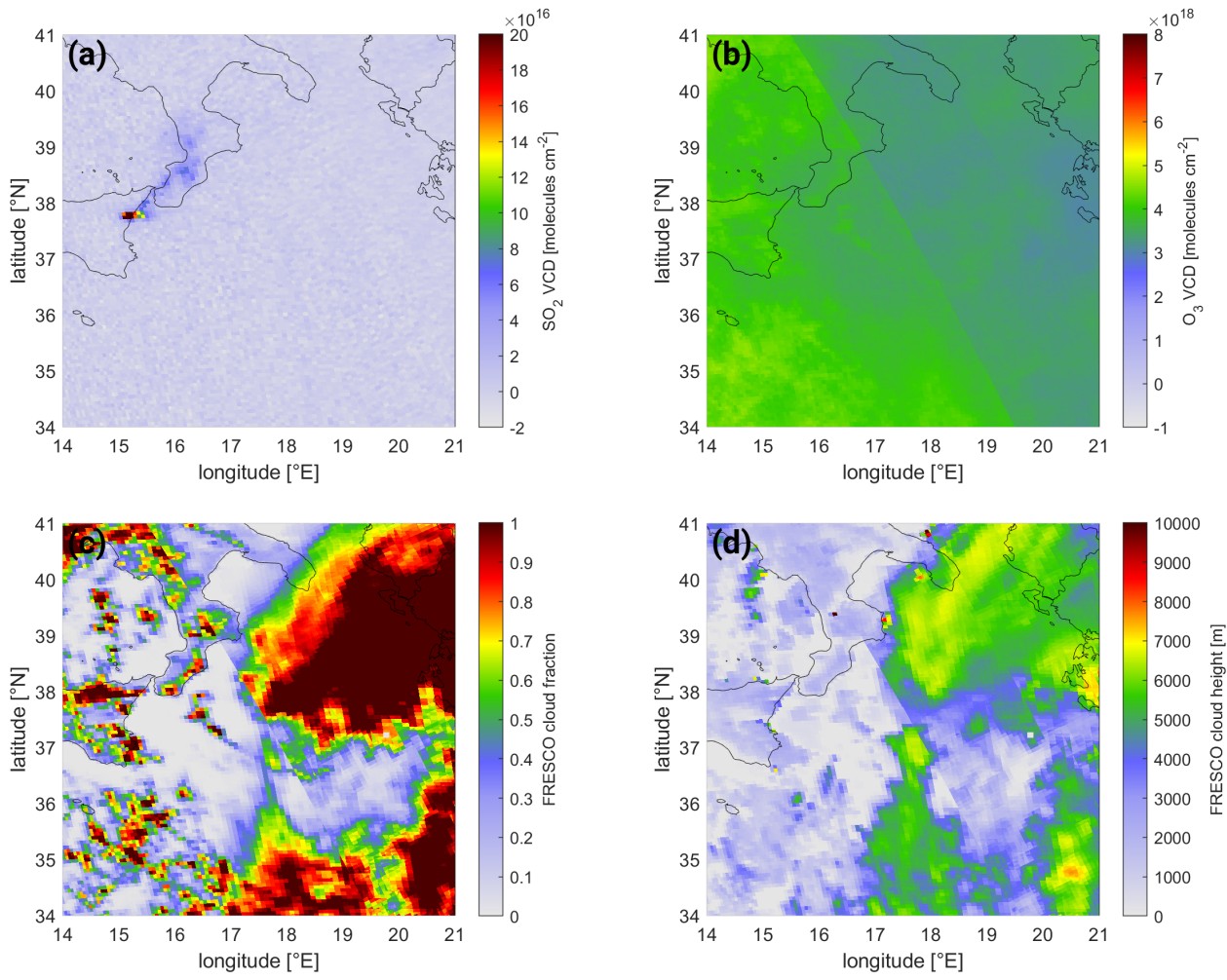

**Figure C3.** Auxiliary data used for correction scheme for the eruption of Mount Etna, Italy, on 29 January 2019. Maps of **(a)** the $SO_2$ VCD, **(b)** the $O_3$ VCD, **(c)** the FRESCO cloud fraction, and **(d)** the FRESCO cloud height. The corresponding uncorrected and corrected $BrO$ VCD map and the $BrO$ background correction map can be found in Fig. 18.

*Data availability.* The TROPOMI $BrO$ VCDs (uncorrected, corrected and the correction terms) as well as the $SO_2$ VCDs are available upon request.

*Author contributions.* SW, SB, and TW contributed to the conception and design of the study. SW performed the data evaluation and all the calculations. CB provided the code-base and infrastructure for the DOAS retrieval. SW prepared the manuscript in collaboration with SB and TW and all the other co-authors. All authors approved the submitted version.

*Competing interests.* The authors declare that they have no conflict of interest.

*Acknowledgements.* We thank ESA and the S-5P/TROPOMI level 1 and level 2 team for providing the L1B spectra and the $NO_2$ L2 data. Moreover, we would like to thank them for the achieved high data quality of TROPOMI and its products. We thank DLR for financial support under funding number 50EE1811B. We thank the Max Planck Computing and Data Facility (MPCDF) for providing access to the high-performing computing cluster used for the DOAS evaluation and post-processing calculations. We thank Jonas Kuhn and Moritz Schöne, both at the IUP Heidelberg, and Steffen Ziegler, MPIC, for valuable discussions about the $BrO$ DOAS retrieval. We thank Natural Earth (naturalearthdata.com) for making their 10m coastline and borders publicly available, which was used to create the coastline and borders in the Earth's maps included in this paper.

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
