# Peer review of "A new accurate retrieval of bromine monoxide inside minor volcanic plumes from Sentinel-5 Precursor/TROPOMI"

_EGUsphere, 2023_

## Author Comment (AC1)

We like to thank the reviewer for his/her review and the useful and constructive remarks. We are also thankful for the opportunity to revise our manuscript in light of the reviewers' comments. We are confident that we addressed all reviewers' comments and trust that the revised manuscript is can now be accepted for publication.

In the following, we give the answers to the anonymous referee #1 regarding the preprint manuscript Warnach et al.: Referee comments are reproduced in black, the author answers in red. Changes to the text in the revised manuscript are also given in red and are enclosed in quotes ("…").

All line numbers in this answer refer to the preprint version of the manuscript.

**General comments**

I have read with interest the manuscript entitled "A new accurate retrieval of bromine monoxide inside minor volcanic plumes from Sentinel-5 Precursor/TROPOMI" submitted to the EGUsphere/AMT by S. Warnach and collaborators. The study presents a thorough empirical and quantitative investigation of the choice of different retrieval parameters to obtain column densities of BrO in volcanic plumes, which can be applied for global observations with the TROPOMI satellite instrument. In essence, the study investigates the systematic effects of the choice of different spectral intervals and the interference of HCHO, and, most notably, it presents an empirical background correction scheme using latitude-dependent information on O3 and cloud height and fraction parameters derived from independent algorithms applied to TROPOMI observations. Most of these ideas are present in previous studies, and for this reason, in my opinion the main merit of the study does not lie on its originality, but rather on the meticulous analysis to find optimal retrieval parameters and to quantify the expected uncertainty for results obtained at different regions of the world.

The background correction scheme seems in a way similar to the DOAS algorithm, in the sense of separating "structured" information from "smooth" information, but it does so in the spatial domain instead of the spectral domain. The rationale behind is that the choice of an imperfect background, here proposed as the average radiance at a latitude band around the globe, leads to a bias that vary in space smoothly in relation to the strong variation caused by a volcanic plume. The further exploration of the co-location of volcanic SO2 leads to an even more accurate representation of the background and retrieval of the volcanic signal.

GC1) It would be interesting to see a discussion on why the region selected for the background: a band around the equator, which includes many potential sources of volcanic or biogenic interference, does not lead to more noisy results. Is this just an effect of reducing random noise by averaging more pixels? Were the examples presented representative of general conditions? A map of uncorrected SO2 columns in the band used for the background presented in the examples could give a visual representation on how "clean" this background was.

As this comment overlaps with the comment to Section 2.3 from referee #2, we deemed it beneficial for both referee comments to be address by a single answer, which encompasses all aspects of this topic.

We thank the reviewer for these positive remarks. Indeed, our approach is novel and this warrants some justification, as up to now, the region used to calculate the earthshine reference was chosen over the pacific only (Theys et al., 2017, Seo et al., 2018). There the assumption that no volcanic plume can be assumed is valid in almost all cases.

We chose a new approach using a reference spectrum stretching over the complete equatorial region for two reasons:

Firstly, and importantly, we chose a band stretching over the complete equatorial region to better account for variations in the spectral response over the day. We found that there are weak stripe features introduced over the day. Therefore, using a pacific reference spectrum, we observed cross-track stripes at locations at a different longitude (e.g. Europe). These vanished when using a reference spectrum obtained from the same orbit (e.g. over equatorial Africa). We found that using a reference spectrum of the complete equatorial band reduces this stripe pattern while best representing the equatorial reference spectrum of the complete day.

Secondly, using the complete band, (1) enhances the statistics for each detector and (2) ensures that each across-track detector has (nearly) the same number of pixels used for the calculation. This minimizes possible inconsistencies between the across-track detectors.

Nevertheless, it has to be ensured that influence of potential sources do not out-weight these advantages. Therefore, for comparison with the previous methods, we also applied a pacific earthshine reference spectrum fit calculated using (135-105°W, +-20°N). We included the results for October 1 in figure 2, and changed its caption to:

(a)                                                                    (b)

[Figure]

"Figure 2: (a) Distribution of the BrO VCD and (b) the rms uncertainty of the DOAS fit for the complete equatorial region [+-20°N, +-180°E] on 1 October 2018 employing three different reference spectra: An earthshine spectrum calculated using the complete equatorial region [+-

20°N, +-180°E] (blue), an earthshine spectrum calculated using the equatorial pacific region [+-20°N, 135°W-105°W] (red), as well as an irradiance spectrum (yellow). For comparability with the earthshine results, the median BrO VCD (corresponding to the median stratospheric column) is subtracted for the Irradiance BrO VCDs."

We moved lines 191 – 196 after line 185, and changed them to:

"However, an earthshine spectrum using the complete equatorial latitude band might include influences from volcanoes as well as biogenic or anthropogenic influences. A comparison between the use of the new expanded area earthshine spectrum calculated from the complete equatorial region [+-20°N, +-180°E], the earthshine spectrum from the pacific equatorial region only [+-20°N, 105°-135°W], as well as using an irradiance spectrum is shown in Fig. 2 for measurements over the equatorial region [+-20°N, +-180°E] on 1 October 2018. It can be seen that the retrieved VCD distribution shows no difference or offset between all three fits (here the stratospheric influence in the irradiance data is eliminated for comparison by subtracting the median BrO VCD). The fit root-mean-square (RMS), however, is about 25% lower (at roughly 6x $10^{-3}$) for both earthshine fits compared to the irradiance fit. This RMS distribution is in very good agreement with RMS reported over a pacific equatorial region by Seo et al. (2019, Fig. 11b) who employed a DOAS earthshine fit based on a large pacific equator region (+-30°N, 150-240°E) independently from the fit presented in this study."

We added a section at the beginning of the appendix (Appendix A) to investigate the strength of a potential contamination of volcanic plumes:

"In order to quantify the influence of the presence of volcanic plumes within the equatorial reference spectrum region onto the BrO VCDs, we selected two example days: 2 October 2021, where only several, small plumes are present (cf. Fig. A1a, red areas), representative of normal conditions, and 30 July 2018, where a very large plume stretched over a large portion of the equatorial region (cf. Fig. A2a, red areas), representative of exceptionally strong volcanic activity within the equatorial region. For both days we identified areas affected by a volcanic plume based on the $SO_2$ signal (as done in Warnach, 2022, cf. Sect. 5.2). Lastly, we calculated the mean BrO VCD within the equatorial region independently for each across-track detector both including and excluding the affected volcanic areas. The difference between both should be equivalent to the contamination of the earthshine reference spectrum.

For the 2 October 2020, there is no difference in the BrO and for $SO_2$ only a small difference for a few detectors (cf. Fig. A1c,d). Our interpretation is that typical signals are too weak to exceed the noise of the BrO retrieval (as the signal-to-noise is two orders of magnitudes larger than for $SO_2$) and that typically only a very small fraction of pixels are affected by volcanic plumes. On the 30 July 2018, which is representative for an exceptionally strong volcanic plume, there is only a contamination of the $SO_2$ SCD of 6x$10^{15}$ molecules $cm^{-2}$ is more than one order of magnitude lower than typical volcanic $SO_2$ SCDs (which are on the order of 1x$10^{17}$ molecules $cm^{-2}$). For BrO, the difference is even smaller and less than 1x$10^{11}$

molecules cm$^{-2}$ which is also at least 1.5 orders of magnitude lower than typical volcanic signals (which are on the order of 1x10$^{13}$ molecules cm$^{-2}$)."

We added two plot for the 30 July 2018 and 2 October 2020 to showcase the differences:

(a)

[Figure]

"Figure A1: (a) Map of the SO$_2$ VCD for the equatorial region on 2 October 2021. The areas of enhanced volcanic signals are marked in red squares. (b) Map of the BrO VCD for the equatorial region on 2 October 2021. (c) Mean SO$_2$ VCD for each across-track detector considering all pixel (blue) and excluding pixel with volcanic gas columns (red). (d) Same plot for the across-track dependent mean BrO VCD."

(a)

[Figure]

[Figure]

"Figure A2: (a) Map of the $SO_2$ VCD for the equatorial region on 30 July 2018. The areas of enhanced volcanic signals are marked in red squares. (b) Map of the BrO VCD for the equatorial region on 30 July 2018. (c) Mean $SO_2$ VCD for each across-track detector considering all pixels (blue) and excluding pixels with volcanic gas columns (red). (d) Same plot for the across-track dependent mean BrO VCD."

In the main body of the manuscript, we added the following text after lines 191-196 (which are modified already above):

„In order to ensure that the inclusion of volcanic plumes within the reference spectrum will not introduce a noticeable contamination into the reference spectrum, we investigated the difference between including and excluding volcanic areas onto the retrieved mean SCD over the equator. This is done for two days in the appendix (cf. Sect. A): 2 October 2021, where only several, small plumes are present (cf. Fig. A1a, red areas), representative of normal conditions, and 30 July 2018, where a very large plume stretched over a large portion of the equatorial region (cf. Fig. A2a, red areas), representative of exceptionally strong volcanic activity within the equatorial region. For the normal conditions on 2 October 2021, excluding the volcanic areas only leads to negligible changes in the $SO_2$ SCD (cf. Fig. A1c) and no detectable changes in the BrO SCD (cf. Fig. A1d). For the exceptional conditions on 30 July 2018, there is a difference of several $10^{11}$ molecules cm$^{-2}$ visible between including and excluding the volcanic areas for BrO SCDs (cf. Fig. A2d). However, this is 1.5 orders of magnitude below typical volcanic BrO columns ($1 \times 10^{13}$ molecules cm$^{-2}$) and therefore negligible. The same is the case for $SO_2$ SCD, where it is more pronounced, but still 1 order of magnitude below typical volcanic columns of $1 \times 10^{17}$ molecules cm$^{-2}$ (cf. Fig. A2c). Furthermore, the large plume on the 30 July 2018 stretches also over the pacific area typically used as a pacific reference region (e.g. 120°-160°W, as used for the operational SO2 product, Theys et al., 2017, or even more affected using 150°E − 120°W, Seo et al., 2019). Thus, in this exceptional case using a pacific reference sector will also not

be free of volcanic influence. To the contrary, in this case the influence is most likely stronger using a pacific reference area, as the plume affects a relatively larger portion of pixels within the reference area compared to our reference area which spans the complete equatorial band. It should further be noted that a constant offset expanding over all across-track detectors would be removed efficiently by our background correction algorithm and would therefore be irrelevant to our approach."

Regarding the reviewer's suggestion to include an uncorrected SO2 map of the equatorial region, we include one in the appendix together with the corresponding BrO statistics (cf. Fig. A1a and A2a).

GC2) Because of the importance of the auxiliary information required for the background correction scheme, namely O3 column densities and cloud parameters, it would be good to provide a brief description of how those parameters were derived.

We thank the reviewer for this comment, as indeed a detailed explanation of neither the $O_3$ VCD nor the cloud parameters is given in Sect. 2.1.2 where it would be most prominent. The cloud parameters are included in Sect. 2.1.2. Since this question overlaps with SC6, describe the changes to the manuscript in the answer to SC6.

The detailed description of the calculation of the $O_3$ VCD is given in section 4.2 in lines 304-314 ("Since there are two [...] molecules$^{-1}$"). This text is now moved to Sect. 2.1.2 after line 135 and the reference to section 4.2 in Line 135 is removed.

The paragraph starting in line 134 now reads:

"Third, for the estimation [...] is derived directly from the DOAS fit. Since there are two different O3 absorption cross-sections as well as two 'Pukite'-pseudo absorbers included in the fit (see Table 1) [...] molecules-1). The resulting $O_3$ SCD is then converted to the geometric $O_3$ VCD following the formalism in Sect. 2.2."

Line 304f. now reads:

„In addition, the schemes ‚ozone' and 'ozone latitude' also include the O3 VCD taken from the BrO fit (cf. Sect. 2.1.2 for a detailed description of the calculation of the O3 VCD). The O3 VCD is then used in two of the three BrO correction schemes. "

[continuing in Line 316]. Furthermore, we added a reference to Sect. 2.1.2 in line 262, which now reads:

"The high-latitude example is depicted in Fig. 5, including the FRESCO CF (Fig. 5a) as well as the O3 VCD (Fig. 5c, taken from the BrO DOAS fit, cf. Sect. 2.1.2 for a description of the calculation of the O3 VCD) [...]"

GC3) The writing style could be more concise. There is room for improvement in avoiding the repetitive, and a bit confusing introduction of what is going to be presented on each section with reference to what has been presented before. Instead of saying for example "in this section we will do X, using what was done in Sect. Y", I suggest presenting directly the new step without going back to the previous steps. The introduction to the sections, as done in the manuscript, adds little to readability. Shortening this will improve the flow of the text.

We thank the reviewer for this comment and agree that the introductions can be shortened.

We therefore changed the introduction of Sect. 4 (lines 236-241) to read as follows:

"In this section, the systematic influences of following three effects onto the BrO retrieval are investigated and discussed for the eight different BrO fit settings: Potential systematic influences of $O_3$, influences of clouds (Sect. 4.1), and the spectral interference with HCHO (Sect. 4.3). "

At the beginning of Section 4.3 we removed in line 383: "This section is looking at the interference between HCHO and BrO for the eight BrO DOAS fits (defined in Sect. 3)." And changed the beginning of the following sentence to: "The response of the eight different BrO DOAS fits [...]"

In line 437, we changed the paragraph to:

"In order to quantify the uncertainty originating from remaining systematic uncertainties onto the retrieval, i. e. the potential 'false' BrO signal created by clouds, $O_3$ or other systematic effects, we look at the BrO VCD of the fit SR 323 – 360 nm (after the correction with the 'ozone latitude' correction scheme)"

In Line 508, we changed it to:

"In order to give an estimate for the total error under all the various cases and locations where volcanic plumes occur, three latitude regions are distinguished:"

Line 531 is changed to:

"In order to accurately retrieve BrO inside volcanic plumes, potential remaining local background gradients have to be accounted for. For this information on the $SO_2$ signal is used to mask the volcanic plume signal and derive a local background correction."

By presenting concrete suggestion on optimal retrieval parameters for volcanic BrO from TROPOMI, this article can make an important contribution to operational global retrievals on volcanoes, complementing the existing capabilities of TROPOMI to detect $SO_2$ from even weak volcanic plumes.

We thank the reviewer again for his/her general comments, which have raised valid questions and suggestions, whose answers substantially improved the manuscript.

**Specific comments (SC)**

SC1) The title is a bit awkward. It indicates to present a retrieval of volcanic BrO from a satellite sensor. It should instead refer to an algorithm to retrieve volcanic BrO *column densities* from satellite sensor *data.* The authors may judge if the algorithm is only valid to this specific satellite sensor.

We thank the reviewer for this most relevant remark. The title can indeed be more precise and specific in what is measured and used. We therefore change the title to:

"A new accurate retrieval algorithm of bromine monoxide columns inside minor volcanic plumes from Sentinel-5 Precursor/TROPOMI observations"

Furthermore, it is indeed important to indicate that the retrieval can in principle be also be applied to other satellite sensors. We therefore expand the last partial sentence of the abstract ", which allows [...] bromine-rich volcanoes." with:

". We present a new and accurate retrieval algorithm of BrO columns from TROPOMI observations, which allows for the detection of even slightly enhanced BrO amounts inside minor eruptive plumes of bromine-rich volcanoes. While designed specifically for TROPOMI observations, the retrieval algorithm is in general also applicable to other hyperspectral satellite observations. However, some parts might require adaptation."

We add in the introduction after line 94:

"Even though our BrO retrieval algorithm is designed for TROPOMI observations, the improvements tested are – aside from satellite specific adaptations – in principle applicable to all hyperspectral satellite observations."

At the end of the conclusion (line 650), we add the following:

"Even though our BrO retrieval algorithm is designed for TROPOMI observations, the improvements proposed here are in principle applicable to the BrO observations inside volcanic plumes from any hyperspectral satellite observations. However, satellite specific adjustments might be required."

SC2) The abstract could be shortened by reducing the information on the first two paragraphs to concentrate on the contribution of the study.

We changed the two paragraphs to read as follows:

"Bromine monoxide (BrO) is a key radical in the atmosphere, influencing the chemical state of the atmosphere, most notably the abundance of ozone ($O_3$). Ozone depletion caused by the release of bromine has been observed and modeled in polar regions, salt pans, and in particular inside volcanic plumes. Furthermore, the molar ratio of BrO and $SO_2$ – which can

be detected simultaneously via spectroscopic measurements using the Differential Optical Absorption Spectroscopy (DOAS) method - is a proxy for the magmatic composition of a volcano and potentially an eruption forecast parameter.

The detection of BrO in volcanic plumes from satellite spectroscopic observations is limited by the precision and sensitivity of the retrieval, which so far only allowed for the detection of BrO during major eruptions. The unprecedented spatial resolution of up to 3.5x5.5km2 and high signal-to-noise ratio of the TROPospheric Monitoring Instrument (TROPOMI) onboard Sentinel-5 Precursor (S-5P) enables to observe and monitor volcanic bromine release globally even for minor eruptions or even quiescent degassing."

SC3) Expand on importance of measuring BrO. How much can O3 be destroyed and by which mechanisms? How the columns of BrO or column ratios of BrO respect to SO2 can help to understand volcanic processes?

We expand the second sentence of the introduction "Volcanic bromine […] ten-year period (Brenna et al., 2019).". It now reads as follows:

"Also Volcanic bromine release is thought to affect atmospheric O3 chemistry (Bobrowski et al., 2003; von Glasow, 2010; Surl et al., 2021). However, O3 measurements inside volcanic plumes are limited and the strength of the O3 destruction of volcanic bromine is still not fully clear (an estimation can be found in the review by Surl et al., 2015). Recently, the development of chemiluminescence instruments for the detection of O3 inside volcanic plumes show promising results (Rüth, 2021; Bräutigam, 2022) and could help to better constrain the volcanic O3 destruction, as they allow interference-free measurements of O3 in volcanic plumes, in contrast to the standard UV-O3 monitors often used today. Another important aspect is the influence of bromine on the stratospheric O3 chemistry. Strong eruptions ejecting bromine into the tropical stratosphere could potentially impact global stratospheric O3 abundance over a ten-year period (Brenna et al., 2019)."

In addition, we change lines 35-38 "and the BrO/SO2 ratio […] Warnach et al., 2019)." to now read the following:

". Ground-based (Bobrowski and Platt, 2007; Gutmann et al., 2018), as well as satellite observations (Hörmann et al., 2013) report inter-volcanic variations of the BrO/SO2 ratio of three orders of magnitude, strongly suggesting a link to differences in the geological settings of the individual volcanoes (Platt et al., 2015). Additionally, variations of the BrO/SO2 over time have been attributed as a proxy for changes in the volcanic systems, for instance at Mt. Etna, Italy (Bobrowski and Giuffrida, 2012), Nevado del Ruiz, Colombia (Lübcke et al., 2014), Cotopaxi, Ecuador (Dinger et al., 2018) and Tungurahua, Ecuador (Warnach et al., 2019). These variations are suggested to be linked to differences in the partitioning of bromine and sulphur from the magmatic melt, i. e. that bromine and sulphur are released at different depths below the surface. However, the interpretation remains difficult, as the partitioning of bromine from the melt is not yet well-constrained."

We added the following references:

Bräutigam, E.: Construction of an airborne chemiluminescence ozone monitor for volcanic plumes, Bachelor thesis, Heidelberg University, https://doi.org/10.11588/heidok.00032085, 2022.

Platt, U. and Bobrowski, N.: Quantification of volcanic reactive halogen emissions, p. 115–132, Cambridge University Press, https://doi.org/10.1017/CBO9781107415683.011, 2015.

Rüth, M.: Characterisation of a chemiluminescence ozone monitor for volcanic applications, Bachelor thesis, University Heidelberg, https://doi.org/10.11588/heidok.00029947, 2021.

Surl, L., Donohoue, D., Aiuppa, A., Bobrowski, N., and von Glasow, R.: Quantification of the depletion of ozone in the plume of Mount Etna, Atmospheric Chemistry and Physics, 15, 2613–2628, https://doi.org/10.5194/acp-15-2613-2015, 2015.

And removed the following references:

Bobrowski, N., von Glasow, R., Giuffrida, G. B., Tedesco, D., Aiuppa, A., Yalire, M., Arellano, S., Johansson, M., and Galle, B.: Gas emission strength and evolution of the molar ratio of BrO/SO2 in the plume of Nyiragongo in comparison to Etna, Journal of Geophysical Research: Atmospheres, 120, 277–291, https://doi.org/10.1002/2013jd021069, 2015.

Kern, C., Sihler, H., Vogel, L., Rivera, C., Herrera, M., and Platt, U.: Halogen oxide measurements at Masaya Volcano, Nicaragua using active long path differential optical absorption spectroscopy, Bulletin of Volcanology, 71, 659–670, https://doi.org/10.1007/s00445-008-0252-8, 2009.

SC4) The description of the DOAS method supported by Eq. 1 misses the essential feature of separation between high- and low-frequency components of the optical depth, that characterizes this method.

We thank the reviewer for noticing this. Indeed, we forgot to include the Polynomial term fitted in the DOAS algorithm to account for broad band absorption structures. We added the polynomial term "P(lambda)" in eq. 1 and added after eq. 1 prior to line 144:

"where sigma is the absorption cross-section and c the concentration of the trace gas i, while the polynomial term P(lambda) accounts for broad-band absorption and scattering, e. g. Rayleigh and Mie scattering."

SC5) Explain carefully the motivation to use the entire equatorial Earth-shine band (at all longitudes) for the background "reference" correction. This region includes quite different albedo regions (land, ocean), and many sources of BrO. The motivation of this choice is far from obvious.

This question is included in GC1) and addressed/answered in detail there.

SC6) Provide essentials of the FRESCO algorithm to obtain cloud height and fraction products used for the sensitivity study.

We changed the sentence "Both data-sets are calculated […]" in line 121 to:

"Both data-sets are calculated using the Fast Retrieval Scheme for Clouds from the $O_2$ A-Band (FRESCO, Compernolle et al., 2021) algorithm, which derives a radiometric cloud fraction and cloud pressure using the reflectance spectrum at 760nm (in the $O_2$ A-band) assuming a Lambertian cloud model. The FRESCO data products are provided within the NO2 operational product (van Geffen et al., 2021, NO2, 2021)."

SC7) Provide essentials of the algorithm used to retrieve the HCHO product used for the sensitivity study. In particular, refer to how BrO interference was treated in such retrieval or if it was neglected?

We expanded the paragraph starting in line 132 to:

"Second, for the study of the influence of HCHO onto the BrO retrieval, the HCHO slant column densities (SCDs) -- provided within the operational TROPOMI HCHO L2 product -- are used (De Smedt et al., 2018). The HCHO SCDs are derived from a DOAS fit in a similar fit wavelength range as BrO (328.5-346nm). In order to minimize interference with BrO, the BrO SCD is fixed using a SCD derived in an independent pre-fit in a larger fit wavelength range (328.5-359nm) (De Smedt et al., 2018)."

SC8) Section 6.1 introduces SO2 measurements, here some of the acronyms are spelled out for the first time. The first two paragraphs of this section could be moved to the Introduction.

We moved the first two paragraphs of section 6.1 (lines 537-545) in the introduction after line 56. We additionally checked that all acronyms are spelled out at the first mention.

SC9) The spectral evaluation setup for SO2 could be included in Table 1.

We like to thank the reviewer for this suggestion and included the spectral evaluation setup for SO2 in table 1 by adding a column "incl. in SO2 fit". In addition, we included the phrase "All the fit parameter included are noted in Table 1 in the column 'incl. in SO2 fit'." in line 557.

**Technical comments (TC, followed by line number)**

TC1) L33- Spell out "sulphur dioxide" before chemical formula on first mention.

We changed the text as suggested.

TC2) L35- The payload is not designed to "determine" the composition, i.e., the instrument cannot determine the composition (this is determined by natural processes), but to measure certain properties of the atmosphere.

We replaced "determine" with "gain information on". It now reads "[…] tool both to gain information on the composition of a volcanic plume […]"

TC3) L38- Define acronyms on first mention (GOME, SCIAMACHY).

We added the full designations for GOME & SCIAMACHY in line 38 and OMI in line 46.

TC4) L49- Correct spelling of "measurments".

We changed it to "measurements"

TC5) L60- Use consistent notation for all molecular species, i.e. "O3" instead of "ozone".

We changed this throughout the manuscript, except when talking about the "cloud ozone correction scheme"

TC6) L63- What is lower altitudes and higher latitudes? Better to indicate "tropical", "mid-latitude" etc., or even better to tell percentage of active volcanoes within +-30 deg, to back up this assertion.

We changed the latitude designation to "tropical, mid, and high latitudes" to now read the following:

"Second, volcanic plumes mainly occur at tropical latitudes, where O3 columns and the SZAs are considerably lower compared to mid- and high latitudes. Third, the use of an earthshine reference spectrum recorded around the equator reduces the optical density of O3 to zero at the equator, further reducing a potential O3 interference at tropical latitudes."

TC7) L81- Add "typical" before "volcanic BrO columns of small eruptions".

We added it accordingly.

TC8) L83- Define "VCD" on first mention.

We thank the reviewer for noticing this. As it is not needed to talk about VCDs in the introduction, and on each other instance in the introduction we talk about "columns" we changed it to "column" in this line as well in order to increase readability.

TC9) L103- Use upper case for the name "Precursor".

We changed it accordingly.

TC10) L112- Correct spelling of "characteristica".

We changed it to "characteristics".

TC11) L113- Better to write "Selection of spectra" instead of "spectra… used" as sub-section title.

We thank the reviewer for this comment. We changed the wording to "Selection of spectra: TROPOMI UVIS"

TC12) L125- Use upper case and complete name for "Sentinel 5-Precursor Expert Users Data Hub".

We changed it accordingly.

TC13) L174- Correct "far off".

We changed it to "far-off"

TC14) L229- Define "dSCD".

We added the abbreviation "(dSCD)" in line 180, it now reads "differential SCDs (dSCDs)", where it is first introduced.

TC15) L243- Use upper case for "Pacific".

We changed it accordingly

TC16) L435- Better to use the noun-phrase "Investigation of systematic…" instead of the continuous verb form "Quantifying the systematic…" in the title of a sub-section.

Many thanks for this suggestion! We prefer to use the noun-phrase "Quantification of the systematic effects", as the title "investigation of systematic effects" is already the title of the section (cf. line 234).

TC17) L472. Similar than previous comment.

Also here we prefer to use the noun-phrase "Quantification of the statistical uncertainties". The reasoning is the same as in TC16.

TC18) L577- Does the value indicated as typical for TROPOMI correspond to one or four standard deviations of SO2? And is this the standard deviation of the residual?

The value $2x10^{16}$ molecules cm$^{-2}$ corresponds to four times the std. dev. The std. deviation refers to the std. dev. of the $SO_2$ background distribution. We clarified this in the text by changing the sentence to:

"This is chosen in this study as four times the standard deviation of the $SO_2$ background distribution (for TROPOMI the standard deviation is typically in the range of $5x10^{15}$ molecules cm$^{-2}$, i.e. the threshold is in the range of $2x10^{16}$ molecules cm$^{-2}$ […]"

**Figures**

Fig. 1) Indicate which references cross sections were used before convolution (reference to authors), preferably as a legend or caption to the figure.

We added them in the caption, which now read as follows:

"Absorption cross-section of BrO (blue, based on Fleischmann et al., 2004) and HCHO (red, based on Meller and Moortgat, 2000) convolved with a typical TROPOMI instrument spectral response function."

Fig. 2) Add labels to y-axes.

We included them.

Fig. 3) Reference to Fig. 4 and Fig.5 in the figure description is not appropriate because one needs the three figures to understand the meaning. Better to explain briefly the reason of the two regions, e.g., with relation to the study of the cloud and stratospheric O3 interferences.

We changed the second sentence in the caption to:

"The areas in the equator and mid-latitude region (marked by red rectangles) are used as case studies to investigate and quantify the effect of clouds and O3 respectively (see section 4.1)."

Fig. 4 (and all following maps) Add units of "deg" to all axes showing lat and lon.

We changed it for all figures to "latitude [°N]" and "longitude [°E]" respectively.

Fig. 5) The O3 background varies very drastically with latitude. It would be good to discuss if the reason for this steep gradient can be found in terms of the general features of O3 dynamics (e.g., presenting a plot of typical O3 latitudinal gradients for comparison).

In the case presented here, the gradient in O3 originates from a change occurring on the boundary of the polar cell and the Ferrel cell, e.g. polar air masses north of 50°N and mid-latitude air masses south. Firstly, the polar jet stream defines this boundary. Secondly the tropopause height decreases in the polar cell.

The jet stream can be identified by a strong wind pattern in roughly 9km altitude. As can be seen in the wind speed at 9km (cf. fig. A1) such a strong wind pattern occurs directly on the edge of the O3 gradient, indicating that the gradient occurs at the location of the jet stream and hence at the boundary of polar and mid-latitudinal air-masses. This is strengthened when looking at the height where the potential vorticity is 2 potential vorticity units ($10\text{-}6\ m^2Ks^{-1}\ kg^{-1}$) – which is one

definition of the tropopause height. This height and hence the tropopause height is decreasing from 12 km south of 50°N to 8km north of 50°N in congruence with the O3 gradient (cf. Fig. A2).

We made the following changes within the manuscript:

In line 264 we write instead of "This follows the gradient line of O3 along the same latitude, indicating that this increase is most probably due to a higher stratospheric column." The following:

"This follows the gradient line of O3 along the same latitude (cf. Fig. 5c). Both BrO and O3 gradients coincide with the transition from mid-latitude to polar air masses indicated by the presence of the jet stream and a change in the tropopause height (cf. Fig B1), where O3 is higher and the stratospheric column is increasing."

We added the section 'High-latitude air-masses influence on BrO and O3' in the appendix including the following text and figure of the tropopause height and wind speed at 9km:

"In order to interpret the coinciding gradients of O3 and BrO in Fig. 5 with respect to meteorology, we look at the location of the jet stream estimated as band of strong wind-speed at 9 km altitude and the tropopause height estimated as the height where the potential vorticity is 2 potential vorticity units (PVU). The corresponding data are taken from ECMWF ERA-5 model data at 12 am UTC for the respective region and is depicted in Fig. B1."

We added the following figure in this section of the appendix:

(a)     (b)

[Figure]

Figure B1: Meteorological conditions for the high-latitude case (cf. fig. 5) on 1 October 2018, taken from ECMWF ERA-5 data at 18:00h UTC: (a) the wind speed at 9km indicating the presence of the jet stream and (b) the tropopause height indicated by the height where the potential vorticity is equal to 2 potential vorticity units.

Fig. 6) This sequence of figures, presented in the sensitivity studies, is difficult to understand. It would help to add text in the caption to guide the reader towards a conclusion. The pattern cannot be understood at first glance. What complicates matters is that the scales for VCD are different.

We added the following explanation to the caption:

"The almost linear influence of both cloud height and cloud fraction is clearly visible for tropical latitudes (middle row). This changes for mid- to high latitudes (upper and lower row), where other influences related to the stratospheric column independent of the cloud parameters can also be seen."

We realize that the different VCD scales complicate the matter and we thank the reviewer for addressing this point. We see the need that this should be addressed somehow. However, as the centers of the different colormaps deviate by $3.5 \times 10^{13}$ which is almost twice the range of the current color scale ($2 \times 10^{13}$), using a single color scale for all subplots would drastically reduce the visibility of the effects in question. We therefore decided against a universal color scale and added the further line to the caption:

"Note the different color scales for each subplot. They always encompass the same range of $2 \times 10^{13}$, but start at different BrO VCDs."

**Tables**

Table 1) "Shift and squeeze" and "ISFR" should be classified as instrumental corrections and not "pseudo absorbers". Define "ISFR" and its parameters in the table's description or as a footnote.

We moved the ISRF and shift & stretch to a new subsection named "Pseudo absorbers for instrumental effects" and defined ISRF and its parameters in two footnotes.

Table 3) It could be limited to include only new information not presented already in Table 1.

We removed this table, changed the reference in line 505 to Table 1 and added "considering the wavelength fit range of 323-360 nm and excluding the absorption cross-section of HCHO." in the same line after "The complete overview of the DOAS fit settings is listed in Table 3"

Furthermore, we changed Table 1, so that the "Species" and "Temperature" have a separate column each (as was done in Table 3).

Lastly, we highlighted the chosen fit wavelength range (323-360nm) in bold and added to the caption: "The proposed final wavelength fit range is highlighted in bold." Furthermore, we added to the footnote of HCHO "Not included in the proposed final fit settings."

---

## Author Comment (AC2)

We like to thank the reviewer for his/her review and the useful and constructive remarks. We are also thankful for the opportunity to revise our manuscript in light of the reviewers' comments. We are confident that we addressed all reviewers' comments and trust that the revised manuscript is can now be accepted for publication.

In the following, we give the answers to the anonymous referee #2 regarding the preprint manuscript Warnach et al.: Referee comments are reproduced in black, answers in red. Changes to the text in the revised manuscript are also given in red and are enclosed in quotes ("…").

Warnach et al. presented. A new scheme to retrieve BrO with improved precision and accuracy. This is made possible thanks to a combination of improved DOAS settings and sophisticated bias correction. This is an important study in view of the systematic investigation of volcanic BrO plumes measured by TROPOMI. The paper is well structured and a pleasure to read. The approach is scientifically sound. I recommend publication in AMT after addressing my (minor) comments below.

All line numbers in the following answers refer to the original (preprint) version of the manuscript.

**Section 2**

-In the section describing TROPOMI, there are sub-sections 2.1.1 and 2.1.2 which are very short. The author could consider removing the sub-sections structure (but keep the text in section 2.1).

We moved the paragraph where the O3 VCD calculation is described into Sect. 2.1.2 as suggested by Reviewer #1. Thus, Sect. 2.1.2 is now longer and we hope that this merits an own subsection. In addition, we feel that it would be better to keep the additional data in a separate section, so that the reader can directly find it.

-Section 2.1.2. Here FRESCO and MICRU are mentioned as available cloud products, but the author does not consider OCRA/ROCINN which is the S5P operational cloud product. Is there a reason for not considering OCRA/ROCINN. For the O3 VCD, why was the operational total ozone column product not used?

We appreciate the valuable comment. The primary reason for choosing FRESCO is that its cloud fraction is directly retrieved on the TROPOMI UV-VIS grid. Ergo no transformation from the SWIR grid was needed for this. The MICRU cloud fraction product was tested and in fact used within the first author's Ph.D project (but is not a readily available product), and we therefore found it relevant to mention it.

We acknowledge that the use of OCRA/ROCINN is another viable option for such kinds of studies as it is readily available on the TROPOMI UV-grid via for example the L2 SO2 product.

As OCRA/ROCINN is the operational product, it is worth to mention this, and we changed line 128 to:

> "However also other cloud products are available (a comparison of TROPOMI cloud products can be found in Latsch et al., 2022), such as the operational cloud product (OCRA/ROCINN, Loyola et al., 2018), and could in principle be used."

Concerning the selection of the O3 VCD:

The decision not to utilize the operational total ozone column product in our study was based on two factors. Firstly, from a practical standpoint, we found it more convenient to use the ozone column obtained from the BrO fit, as it was readily available and does not require, for example, additional data download.

Second, in contrast to our fit, the operational total ozone column product does not include the SO2 cross-section in the DOAS fit. The total ozone column fit is therefore more susceptible to spectral interference between SO2 and O3. This susceptibility becomes particularly critical when retrieving trace gases within a volcanic plume characterized by high SO2 columns. Consequently, the ozone columns derived from the total ozone fit will deviate within the volcanic plume. This inaccuracy, in turn, would lead to an incorrect correction term within the plume and an erroneous BrO column estimation

Therefore, considering both practicality and accuracy considerations, we made the decision to rely on the ozone column derived from the BrO fit rather than using the operational total ozone column product.

In order to reflect this in the manuscript, we add the following after line 135 (where we also moved the detailed explanation of the O3 VCD calculation following a comment from reviewer #1) after "The O3 VCD is derived directly from the BrO DOAS fit.":

> "We favour this O3 VCD over the operational O3 L2 product (1) because it is more practical and most importantly (2) because in difference to our fit, the operational O3 product does not include SO2 within the DOAS fit and is therefore affected stronger by SO2-O3 spectral interference leading to high inaccuracies within volcanic plumes."

We added the following reference:

Loyola, D. G., García, S. G., Lutz, R., Argyrouli, A., Romahn, F., Spurr, R. J. D., Pedergnana, M., Doicu, A., García, V. M., and Schüssler, O.: The operational cloud retrieval algorithms from TROPOMI on board Sentinel-5 Precursor, Atmospheric Measurement Techniques, 11, 409–427, https://doi.org/10.5194/amt-11-409-2018, 2018.

-Equation 1 does not include the broad-band terms. Please add those to the equation.

We thank the reviewer for alerting us of this omission. Indeed, we forgot to include the Polynomial term fitted in the DOAS algorithm to account for broad band absorption structures. We added the polynomial term "P(lambda)" in eq. 1 and added after eq. 1 prior to line 144:

"where sigma is the absorption cross-section and c the concentration of the trace gas i, while the polynomial term P(lambda) accounts for broad-band absorption and scattering processes, e. g. Rayleigh and Mie scattering."

-Equation 3 is incorrect as it should in principle imply the cosine of the angles (SZA, VZA). Please update the equation.

We thank the reviewer for noticing this small but significant omission and added the cosine to both parameters. The denominators now read "cos(SZA)" and "cos(VZA)".

-Section 2.3: it is mentioned that the full zonal band (from 20S to20N) is used as reference region. However, the author is not really justifying its choice. This is a very large region which covers many volcanoes (with potential contamination by strong plumes SO2 and BrO, and to some extend HCHO). Please clarify why it is an advantage to use such an extended reference region compared to a smaller region (e.g., in the equatorial Pacific).

As this comment overlaps with the General comment 1 (GC1) from referee #1, we deemed it beneficial for both referee comments to be address by a single answer, which encompasses all aspects of this topic.

We thank the reviewer for these positive remarks. Indeed, our approach is novel and this warrants some justification, as up to now, the region used to calculate the earthshine reference was chosen over the pacific only (Theys et al., 2017, Seo et al., 2018). There the assumption that no volcanic plume can be assumed is valid in almost all cases.

We chose a new approach using a reference spectrum stretching over the complete equatorial region for two reasons:

Firstly, and importantly, we chose a band stretching over the complete equatorial region to better account for variations in the spectral response over the day. We found that there are weak stripe features introduced over the day. Therefore, using a pacific reference spectrum, we observed cross-track stripes at locations at a different longitude (e.g. Europe). These vanished when using a reference spectrum obtained from the same orbit (e.g. over equatorial Africa). We found that using a reference spectrum of the complete equatorial band reduces this stripe pattern while best representing the equatorial reference spectrum of the complete day.

Secondly, using the complete band, (1) enhances the statistics for each detector and (2) ensures that each across-track detector has (nearly) the same number of pixels used for the calculation. This minimizes possible inconsistencies between the across-track detectors.

Nevertheless, it has to be ensured that influence of potential sources do not out-weight these advantages. Therefore, for comparison with the previous methods, we also applied a pacific earthshine reference spectrum fit calculated using (135-105°W, +-20°N). We included the results for October 1 in figure 2, and changed its caption to:

[Figure]

[Figure]

"Figure 2: (a) Distribution of the BrO VCD and (b) the rms uncertainty of the DOAS fit for the complete equatorial region [+-20°N, +-180°E] on 1 October 2018 employing three different reference spectra: An earthshine spectrum calculated using the complete equatorial region [+-20°N, +-180°E] (blue), an earthshine spectrum calculated using the equatorial pacific region [+-20°N, 135°W-105°W] (red), as well as an irradiance spectrum (yellow). For comparability with the earthshine results, the median BrO VCD (corresponding to the median stratospheric column) is subtracted for the Irradiance BrO VCDs."

We moved lines 191 – 196 after line 185, and changed them to:

"However, an earthshine spectrum using the complete equatorial latitude band might include influences from volcanoes as well as biogenic or anthropogenic influences. A comparison between the use of the new expanded area earthshine spectrum calculated from the complete equatorial region [+-20°N, +-180°E], the earthshine spectrum from the pacific equatorial region only [+-20°N, 105°-135°W], as well as using an irradiance spectrum is shown in Fig. 2 for measurements over the equatorial region [+-20°N, +-180°E] on 1 October 2018. It can be seen that the retrieved VCD distribution shows no difference or offset between all three fits (here the stratospheric influence in the irradiance data is eliminated for comparison by subtracting the median BrO VCD). The fit root-mean-square (RMS), however, is about 25% lower (at roughly $6 \times 10^{-3}$) for both earthshine fits compared to the irradiance fit. This RMS distribution is in very good agreement with RMS reported over a pacific equatorial region by Seo et al. (2019, Fig. 11b) who employed a DOAS earthshine fit based on a large pacific equator region (+-30°N, 150-240°E) independently from the fit presented in this study."

We added a section at the beginning of the appendix (Appendix A) to investigate the strength of a potential contamination of volcanic plumes:

"In order to quantify the influence of the presence of volcanic plumes within the equatorial reference spectrum region onto the BrO VCDs, we selected two example days: 2 October 2021, where only several, small plumes are present (cf. Fig. A1a, red areas), representative of normal conditions, and 30 July 2018, where a very large plume stretched over a large

portion of the equatorial region (cf. Fig. A2a, red areas), representative of exceptionally strong volcanic activity within the equatorial region. For both days we identified areas affected by a volcanic plume based on the SO2 signal (as done in Warnach, 2022, cf. Sect. 5.2). Lastly, we calculated the mean BrO VCD within the equatorial region independently for each across-track detector both including and excluding the affected volcanic areas. The difference between both should be equivalent to the contamination of the earthshine reference spectrum.

For the 2 October 2020, there is no difference in the BrO and for SO2 only a small difference for a few detectors (cf. Fig. A1c,d). Our interpretation is that typical signals are too weak to exceed the noise of the BrO retrieval (as the signal-to-noise is two orders of magnitudes larger than for SO2) and that typically only a very small fraction of pixels are affected by volcanic plumes. On the 30 July 2018, which is representative for an exceptionally strong volcanic plume, there is only a contamination of the SO2 SCD of $6 \times 10^{15}$ molecules cm$^{-2}$ is more than one order of magnitude lower than typical volcanic SO2 SCDs (which are on the order of $1 \times 10^{17}$ molecules cm$^{-2}$). For BrO, the difference is even smaller and less than $1 \times 10^{11}$ molecules cm$^{-2}$ which is also at least 1.5 orders of magnitude lower than typical volcanic signals (which are on the order of $1 \times 10^{13}$ molecules cm$^{-2}$)."

We added two plot for the 30 July 2018 and 2 October 2020 to showcase the differences:

[Figure]

"Figure A1: (a) Map of the SO$_2$ VCD for the equatorial region on 2 October 2021. The areas of enhanced volcanic signals are marked in red squares. (b) Map of the BrO VCD for the equatorial

region on 2 October 2021. (c) Mean SO$_2$ VCD for each across-track detector considering all pixel (blue) and excluding pixel with volcanic gas columns (red). (d) Same plot for the across-track dependent mean BrO VCD."

(a)

[Figure]

(b)

(c)                                                            (d)

"Figure A2: (a) Map of the SO$_2$ VCD for the equatorial region on 30 July 2018. The areas of enhanced volcanic signals are marked in red squares. (b) Map of the BrO VCD for the equatorial region on 30 July 2018. (c) Mean SO$_2$ VCD for each across-track detector considering all pixels (blue) and excluding pixels with volcanic gas columns (red). (d) Same plot for the across-track dependent mean BrO VCD."

In the main body of the manuscript, we added the following text after lines 191-196 (which are modified already above):

„In order to ensure that the inclusion of volcanic plumes within the reference spectrum will not introduce a noticeable contamination into the reference spectrum, we investigated the difference between including and excluding volcanic areas onto the retrieved mean SCD over the equator. This is done for two days in the appendix (cf. Sect. A): 2 October 2021, where only several, small plumes are present (cf. Fig. A1a, red areas), representative of normal conditions, and 30 July 2018, where a very large plume stretched over a large portion of the equatorial region (cf. Fig. A2a, red areas), representative of exceptionally strong volcanic activity within the equatorial region. For the normal conditions on 2 October 2021, excluding the volcanic areas only leads to negligible changes in the SO$_2$ SCD

(cf. Fig. A1c) and no detectable changes in the BrO SCD (cf. Fig. A1d). For the exceptional conditions on 30 July 2018, there is a difference of several $10^{11}$ molecules cm$^{-2}$ visible between including and excluding the volcanic areas for BrO SCDs (cf. Fig. A2d). However, this is 1.5 orders of magnitude below typical volcanic BrO columns ($1\times10^{13}$ molecules cm$^{-2}$) and therefore negligible. The same is the case for SO$_2$ SCD, where it is more pronounced, but still 1 order of magnitude below typical volcanic columns of $1\times10^{17}$ molecules cm$^{-2}$ (cf. Fig. A2c). Furthermore, the large plume on the 30 July 2018 stretches also over the pacific area typically used as a pacific reference region (e.g. 120°-160°W, as used for the operational SO2 product, Theys et al., 2017, or even more affected using 150°E – 120°W, Seo et al., 2019). Thus, in this exceptional case using a pacific reference sector will also not be free of volcanic influence. To the contrary, in this case the influence is most likely stronger using a pacific reference area, as the plume affects a relatively larger portion of pixels within the reference area compared to our reference area which spans the complete equatorial band. It should further be noted that a constant offset expanding over all across-track detectors would be removed efficiently by our background correction algorithm and would therefore be irrelevant to our approach."

**Section 4**

-Section 4.1. on the effect of clouds. From Fig4, it is not clear to me whether the observed effect of clouds is an artefact or not. For large CF and elevated clouds, the BrO VCDs are lower which is compatible with a possible cloud shielding of the tropospheric BrO column. I am not sure why this should be corrected.

We thank the reviewer for this comment. We agree, that we have maybe not sufficiently explained why these structures of non-volcanic origin should be removed. The possible cloud shielding constitutes a "background offset" which – while interesting for other applications – is a source of uncertainty for the estimation of the volcanic BrO column and therefore not desirable in this study. This is especially relevant as cloud edges can form a distinct gradient in BrO. Any gradient in the BrO background overlapping with the volcanic plume dispersion pattern can lead to a systematically false gradient in the estimation of the volcanic BrO column.

We add the following sentences to the end of section 3.1 (after line 234):

"As all gradients in the BrO column of non-volcanic origin constitute a potential systematic error source, they ideally are removed via a background correction (e.g. via a spatial polynomial, Hörmann et al., 2013). In order to further improve the accuracy of the BrO retrieval, we investigate a more sophisticated correction scheme in Sect. 4.2."

To also add some more information on the origin of the cloud-related BrO patterns, we added the following sentence prior to the aforementioned sentences:

"The reason for this cloud effect is not fully clear. The effect could be a spectroscopic artefact (e.g. via the Ring Spectrum) or a true shielding effect of a potential tropospheric BrO background column."

-Fig5: it is stated that:" the sign of the relation is inverted and high cloud fraction results in elevated BrO VCDs" but without the information on cloud height it is difficult to know whether there is a significant difference as for the effect of clouds, compared to the tropics. Please add the cloud height map and expand the discussion.

We thank the reviewer for noticing this. We added the FRESCO cloud height to fig. 5 and rearranged the subplots, the caption now reads:

"Maps of the (a) FRESCO cloud fraction, (b) FRESCO cloud height, (c) retrieved O3 VCD, (d) BrO VCD from fit SR 323 – 360 nm without any correction and (e) BrO VCD from fit SR 323 – 360 nm after applying the correction scheme. Data taken over the northern high latitude region [40° – 60° N, 110° – 50° W] on 1 October 2018."

After re-analyzing the situation, we agree that it is difficult to separate the effect of cloud fraction and O3 for low clouds in this figure. As this distinction is not necessary for the interpretation, we removed the following line 265: "These coincide with cloud cover (indicated by the cloud fraction). In contrast to the equatorial region, the sign of the relation is inverted and high cloud fraction results in elevated BrO VCDs." And replaced it with:

"These coincide with cloud cover (indicated by the cloud fraction) and show a positive BrO signal for high cloud cover."

Furthermore, we replaced lines 284-288 "While the general […] independent of cloud fraction. This dependency" with the following:

"While the general CH dependency of the BrO VCD (higher clouds, lower BrO VCD) is prevailing for high latitudes, the CF dependency appears to be changing its sign, so that a lower cloud fraction leads to a lower BrO VCD for 70° N – 90° N. However, there is an overlaying BrO signal for cloud heights between 1000 and 5000 m, independent of cloud fraction in this region, which complicates the interpretation. This feature"

From FigA1, it seems that the 323-360nm range is the one with the strongest cloud impact. I find it hard to justify that this range is the one retained for the final SCD retrievals.

While this is true and would mean that the 323-360nm fit range is not the best, after applying the cloud-ozone correction, this impact is removed and all fit ranges show a similarly negligible cloud impact. Therefore, regarding the cloud impact, there is no "better" or "worse" fit range after the correction is applied and therefore not considered for the fit selection. This is addressed in the text already in line 500: "The systematic influence of clouds and the stratospheric background can be well corrected by the correction scheme described in Sect 4.1 for all eight fits. Thus, there is no preference for a fit range or fit setting."

**Section 4.2.**

-for low CF, the retrieved cloud height is uncertain, not to say ill-defined. How do you manage this in your correction?

We thank the reviewer for this remark, and agree completely that this has to be considered carefully. We therefore employed a low order polynomial (order two) for the cloud height to prevent potential inconsistencies to affect the correction term.

-Regarding the lat-ozone correction, it is mentioned that 'the latitude band can be adjusted for each volcanic plume' (line 351). I guess this is a future implementation wish or is this really what is implemented. Also, it is not clear how frequent the correction parameters are updated. Is this done separately for each calendar day? Please clarify.

We thank the reviewer for raising this question. The latitude band is in fact chosen individually for each volcano already in the present study and this is mentioned in section 6.3. We realize that our wording in line 351 might suggest otherwise. Therefore, we changed line 352 to read:

"As the latitude band location can be adjusted freely, the latitude band will be chosen individually for each volcano in order to ensure that the volcano is located at the center of the latitude band and not on its edge (e.g. for Mt. Etna located at 37°N, the latitude band 25°-45° N is chosen for the background correction, cf. Sect. 6.3)."

-Figure 11 is an interesting plot but it is not clear from where the model factor of 6.22e12 is coming. Could you elaborate?  Also, in the evaluation of the systematic uncertainty of the retrievals (summarized in Table4), the authors assume that a good estimate of these systematic influences can be obtained from the SCD std over many pixels. The validity of this approach is not clear because random uncertainties are still present and contribute to the std (these are not reducing as the square root of the number of pixels, as they do for the estimated mean). Also, systematic uncertainties like the one related to the BrO cross-section uncertainty is not accounted for.

We thank the reviewer for this comment. The factor $6.22 \times 10^{12}$ originates from the mean std. deviation without any binning (e.g. single pixel base) at the equator and is used in Fig. 11a only to illustrate the difference. In Fig. 11b the respective model curves for each latitude bands are used. We realize this is not clear from the figure caption and therefore added to the caption of 11a: "For illustration purpose only the model curve for the equatorial region [10°S -10°N] is plotted.". Furthermore, we added to the caption of 11b: "For each latitude band an individual model curve based on the respective standard deviation for pixel binning factor of 1 is used."

We thank the reviewer for questioning the validity of our approach to estimate the systematic uncertainties. Indeed, the standard deviation does not depend on sample size. However, in our approach we bin the data spatially and use these "spatial means" as a new sample. This modified sample has then a lower standard deviation, following a 1/sqrt(N) dependency: a binning of 2 pixels results in a sample with ~0.71 times the original stdev (similarly the standard deviation of the TROPOMI pixels at the edge of the swath increase, because the binning size is reduced there). The data plotted in fig. 11a is therefore the standard deviation of different samples, who all have

the same mean value but are less affected by random noise for larger binning factors. We realize that this is not well elaborated in the manuscript. We therefore add change line 443 to :

"The statistical variation can be quantified by a Gaussian fit over the BrO VCD distribution and the statistical variation then estimated by the standard deviation. Statistical fluctuations can be reduced by spatial binning of neighbouring pixels, which reduces the standard deviation of the binned data by 1/square root of the number of pixel binned. Systematic effects, however, can generally not be eliminated by spatial averaging/binning. Thus, ideally, only the systematic effects remain for high binning factors and the deviation from the expected decrease can be used as an estimate for the order of magnitude of systematic effects. In order to estimate the systematic uncertainties, we employ a spatial binning of neighbouring pixels using binning factors of 1 to 100 in both spatial dimensions, corresponding to 1 to 10000 pixels binned respectively. For each binning factor as well as for each latitude band a Gaussian fit is done separately. The resulting standard deviations are plotted in Fig. 11."

The reviewer is correct in pointing out that systematic uncertainties of the BrO cross-sections are not addressed by this. However, a possible error of the assumed BrO cross-section would basically result in an overall scaling factor of the retrieved BrO SCDs, but would not affect the spatial gradients and regional enhancements over background which are the basis of our quantification of BrO emissions.

**Typos and suggestions**

-p4, l111: ultra-viole**tt** -> ultra-violet

We changed it accordingly.

-p4, l112: characteristic**a** -> characteristics

We changed it accordingly.

-Single sub-section like 3.1 is not necessary. Instead I would write just after l225: "To illustrate the fit performance, the global BrO VCD map for 1 October 2018…"

We thank the reviewer for this improvement in readability and changed it accordingly.

-Fig4c: it would be good to only show the cloud heights above a certain CF threshold, to better appreciate low/high clouds.

We acknowledge the notion of the reviewer, but would rather not mask any values, as we do not do this in our correction scheme.

-FigA1: it would be good to have the same color bar limits for all 8 subplots, otherwise it is difficult to compare the results.

We agree with the reviewer completely and thank for this remark. We changed the colorbar limits to [-1x10$^{13}$ 1x10$^{13}$] for all plots.

-line 310: '..the for the wavelength..' -> '..the wavelength..'

We changed it accordingly.

-line 445: '..increasing increasing..'->'..increasing..'

We changed it accordingly.

-Table 3 is redundant. I would propose to remove it. Table 1 could better highlight the preferred settings (e.g., with the corresponding text in bold).

We thank this reviewer for this comment and agree that Table 3 is redundant.

We removed Table 3, changed the reference in line 505 to Table 1 and added "considering the wavelength fit range of 323-360 nm and excluding the absorption cross-section of HCHO." in the same line after "The complete overview of the DOAS fit settings is listed in Table 3"

Furthermore, we changed Table 1, so that the "Species" and "Temperature" have a separate column each (as was done in Table 3).

Additionally, we highlighted the chosen fit wavelength range (323-360nm) in bold and added to the caption: "The proposed final wavelength fit range is highlighted in bold." Furthermore, we added to the footnote of HCHO "Not included in the proposed final fit settings."

-line 532: 'the latitudinal background correction is applied' here you mean the ozone-cloud correction, right?

Yes, we changed it to "ozone latitude correction".